# Chronic macrophage activation derails muscle repair by disrupting mannose-receptor-linked plasticity revealed by endogenous *irg1/acod1* tracking

Caroline G. Spencer [1,3], Matthew Hamilton [1,3], Ethan Bedsole[1], Yingshan N. Wei [1], Alison M. Rojas [1], Andrew Burciu[1], John Zhu[1], Keith Z. Sabin[1] & Celia E. Shiau [1,2] ✉

Macrophages are central drivers of chronic inflammation, yet how a sustained inflammatory state alters their function remains unclear. Using GFP knock-in zebrafish targeting *irg1/acod1* that marks macrophage activation, we track the dynamic transitions of macrophage states during acute muscle injury under homeostatic and chronically inflamed conditions, induced by genetic mutation of *nlrc3l*. In the chronic inflammation model, muscle repair is impaired and expression of the mannose receptor *mrc1b/cd206* is severely downregulated in a *myd88*-dependent manner. Two reparative macrophage subtypes, defined by their cellular behavior and single-cell transcriptomics profile, clustering and muscle-encasing, are lost. A chronic infection model recapitulates these defects, underscoring the link to macrophage *mrc1b* repression. Depleting either *mrc1b* or macrophages impairs muscle repair. Reinstating normal macrophage states by restoring macrophage *nlrc3l* expression or ablating *myd88*-mediated inflammatory pathways rescues muscle repair in *nlrc3l* mutants. Contrary to conventional discrete states, we identify hybrid M1/M2 macrophage states post-injury. While transient during normal injury response, a pro-inflammatory hybrid state persists during chronic activation, which restricts macrophage heterogeneity, represses *mrc1b*, and inhibits intracellular cathepsin K accumulation, a hallmark of reparative subtypes. Thus, our study provides mechanistic insight into the dynamics of macrophage activation during muscle injury and repair, and how these processes are modulated under chronic inflammation.

Immune dysregulation that causes persistent inflammatory cell infiltration can lead to chronic tissue damage, including secondary injury beyond the original insult[1–5]. Macrophages play a central role in this process, as inappropriate or prolonged activation drives chronic inflammation[1,3,4,6]. However, the impact of altered macrophage states on injury responses and tissue repair, particularly in the complex intact living organism, remains incompletely understood. Furthermore, the mechanisms by which macrophages balance the need to promote

[1]Department of Biology, University of North Carolina at Chapel Hill, Chapel Hill, NC, USA. [2]Department of Microbiology and Immunology, University of North Carolina at Chapel Hill, Chapel Hill, NC, USA. [3]These authors contributed equally: Caroline G. Spencer, Matthew Hamilton. ✉e-mail: shiauce@unc.edu

acute inflammation for tissue repair with preventing chronic inflammation and associated collateral damage remain elusive.

Skeletal muscle has been a prominent model for studying immune regulation of tissue injury and repair[3,6,7]. This model highlights the critical interactions among macrophages (the primary inflammatory cells), injured muscle cells, satellite cells (muscle-specific stem cells), and other cell types, all of which contribute to tissue recovery following injury[1,3,6,8]. Recent research underscores the essential and evolutionarily conserved role of acute inflammation in tissue repair, with evidence spanning from zebrafish to humans[7,9–11]. Although prior studies have provided insights into the role of macrophages in responding to and facilitating acute muscle injury repair[1,3,6,8], including via single-cell transcriptomics and trajectory analysis to infer temporal changes and cellular interactions[12,13], direct real-time in vivo imaging of macrophage behaviors and interactions throughout the injury response remains lacking. Time-lapse imaging, combined with a rigorous longitudinal microscopy approach, enabled us to track macrophages across distinct phases of muscle injury and repair within the same subjects. Such analyses are essential for fully and precisely understanding the progression from inflammation to resolution at both the cellular and organismal levels in an intact living system. Macrophages are believed to transition from a pro-inflammatory state to a pro-repair state at different phases of injury response[1,8]. However, there remains debate about whether these macrophage states are distinct and non-overlapping, or if they represent different subpopulations[3,14,15]. Understanding these transitions is critical, as macrophage dysregulation is implicated in various diseases, including diabetes, cancer, and neurodegenerative disorders, where chronic activation of macrophages or microglia contributes to pathology[1,5,15].

Metabolic reprogramming is a hallmark of macrophage activation and involves the upregulation of the mitochondrial enzyme aconitate decarboxylase (*acod1*), also known as immune-responsive gene 1 (*irg1*), in response to diverse immune triggers, including microbial infections and endogenous damage-associated molecular patterns (DAMPs)[16–19]. The transcriptional induction of *irg1* in macrophage activation has been well-characterized in both zebrafish and mammals[17,18,20–22]. To investigate macrophage state transitions during injury in real time within a localized and controlled environment, we generated a GFP knock-in allele at the zebrafish *irg1/acod1* locus. This genetically modified allele allows tracing and isolation of macrophages at different activation states based on differential *irg1* expression. Previous studies in zebrafish have shown that *irg1* induction is macrophage-specific following bacterial infection, as determined through co-localization with cell-specific markers and macrophage ablation approaches[17,22]. While an earlier study created a transposon-generated fluorescent reporter for *irg1*[22], we anticipated that the precise GFP knock-in allele would offer a more quantitative readout of endogenous *irg1* expression, ensuring reliable and consistent quantification across individuals and clutches while avoiding non-specific expression. Our direct comparison of this knock-in model with a conventional transposon-mediated *irg1* reporter for analysis of activated macrophages supported these expectations.

To explore the consequences of chronic macrophage activation in vivo, we employed the zebrafish GFP knock-in allele in a mutant background *st73* lacking the NOD-like receptor *nlrc3l*, a mutation known to cause chronic inflammation[23,24]. These mutants display constitutive macrophage activation in the absence of overt immune challenges[24], allowing consistent tracking of chronic macrophage activation without the need to experimentally induce immune responses, thereby minimizing variability. Through dynamic imaging and longitudinal analysis, we evaluated the chronic macrophage activation and demonstrated that it led to the depletion of two key functional macrophage subtypes required for muscle injury clearance and repair in vivo. A parallel model of chronic macrophage activation by persistent *E. coli* infection mirrored these phenotypes. Furthermore,

impaired tissue repair mimics the effects of macrophage depletion, and is linked to a significant reduction of the mannose receptor *mrc1b* (also known as CD206). Single-cell analysis revealed a profound shift in the transcriptional landscape of the chronically activated macrophages toward a dominating inflammatory state (sustained expression of inflammatory proteinase *mmp9*, *tnfa*, *cxcl11.1* among other inflammatory genes) while still activating reparative pathway genes. This resulted in reduced cellular heterogeneity and loss of reparative subtypes required for muscle repair. Thus, chronic macrophage activation impairs muscle injury repair not by enhancing or expanding inflammatory functions, but by disrupting reparative programs—particularly through *mrc1b* downregulation.

Taken together, this study focuses on the dynamic transitions and shifts in macrophage cell states by visualizing these processes in real time during an injury response. By forcing all macrophages toward chronic pro-inflammatory states using the *nlrc3l* mutants, we test how adaptable and reversible macrophage states are upon an injury perturbation within the intact living organism. Notably, chronic activation drives *myd88*-dependent repression of the mannose receptor *mrc1b/cd206* in macrophages in both the genetic and persistent infection model of chronic inflammation. These insights reveal a previously unrecognized mechanism linking chronic inflammation to a severe loss of mannose receptor as a cause for reduced macrophage plasticity and impaired tissue repair.

## Results

### Generation of GFP knock-in in zebrafish *irg1/acod1* to monitor macrophage activation

We employed a 48-bp short DNA homology-mediated end joining CRISPR-Cas9 strategy, termed GeneWeld[25,26], to facilitate the integration of a P2A-GFP reporter cassette into exon 2 of the zebrafish *irg1/acod1* locus (referred to as *irg1* herein), proximal to the coding sequence initiation (Fig. 1). Following the selection of *irg1* gRNA #2 based on its superior efficacy in inducing CRISPR-induced double-stranded DNA breaks, the genomic region encompassing the gRNA was sequenced from multiple wild-type zebrafish to identify the upstream and downstream 48 bp homology sequences for cloning into the gene targeting GeneWeld pGTag vector (Fig. 1a). Wild-type zebrafish harboring exact matches to the short homology sequences flanking the *irg1* gRNA target site were chosen as parents for single-cell zebrafish embryo injections for CRISPR/Cas9 targeting (Fig. 1a). Subsequent analysis revealed a germline transmission rate of less than 1% for precise GFP knock-in, validated by site-specific PCR assays with two correctly targeted founder fish (#7 and #31) in the F1 generation (Fig. 1b, c). The precise in-frame integration of P2A-GFP at the zebrafish *irg1* locus was confirmed via Sanger sequencing in both F1 and F2 generations (Fig. 1c). Notably, zebrafish carrying the GFP knock-in allele of *irg1*, designated as "*irg1-KI:GFP*," exhibited segregation consistent with Mendelian inheritance patterns expected for a single-site integration.

Furthermore, to test the function of the new knock-in reporter, we devised a dual labeling system utilizing *irg1-KI:GFP* to label activated macrophages coupled with *mpeg1:BFP*, which labels all macrophages with BFP (Fig. 1d). Upon LPS activation, macrophages in the brain, also known as microglia, robustly expressed GFP, with varying levels at the single-cell level, indicative of the capacity of the knock-in reporter to capture heterogeneity in activation levels compared to the uniformly low basal GFP expression observed in uninjected wild-type controls (Fig. 1d).

Since the GFP knock-in (KI) model effectively replaces the functional *irg1* locus, creating a knock-in knockout condition, we independently created an *irg1* knockout mutation *bcz13* using CRISPR/Cas9 that causes a premature stop codon in *irg1* (Supplementary Fig. 1). This mutation was studied to verify that heterozygous individuals did not exhibit a phenotype and were functionally equivalent to wild-type

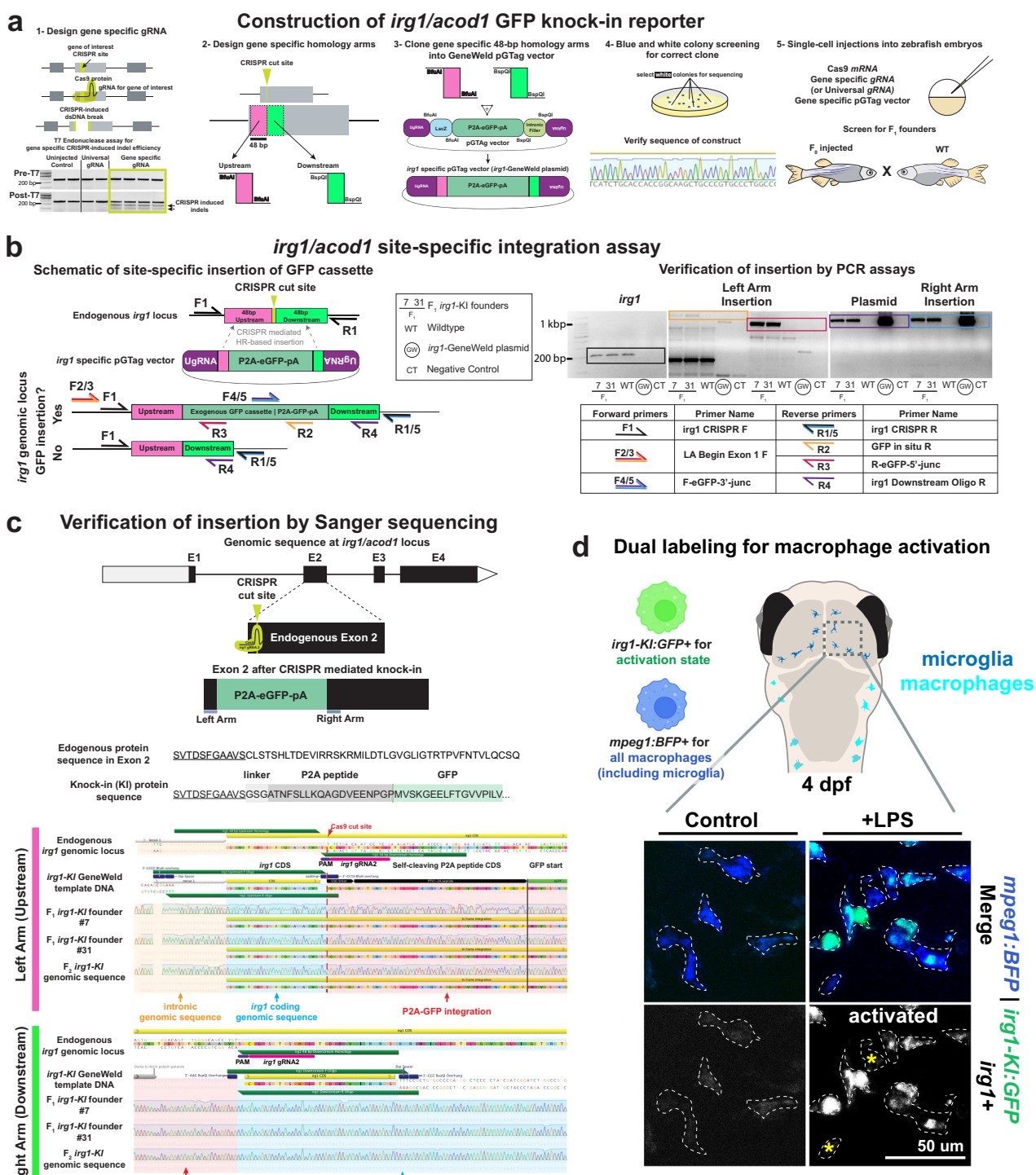

**Fig. 1 | Construction of zebrafish *irg1/acod1* GFP knock-in for tracking the dynamics of macrophage activation in vivo. a** Construction of vector and zebrafish line based on GeneWeld method[25,26]. **b** Site-specific insertion of GFP and verification of insertion by a suite of PCR assays. See Supplementary Data 1. **c** Schematic of the GFP knock-in protein sequence at the targeted locus as verified by Sanger DNA sequencing of F1 founders (#7 and #31) and a F2 progeny. **d** Dual labeling of activated brain-resident macrophages, also known as microglia, in the larval zebrafish brain after LPS stimulation demonstrates efficacy of the GFP knock-in reporter. Reporter labels LPS-activated microglia with high GFP and conveys heterogenous states as few cells (asterisks) show less activation based on very low GFP. Microglia are labeled by *mpeg1:BFP* and demarcated by dotted lines. Homeostatic microglia (no LPS) have low baseline GFP expression as expected.

(Supplementary Fig. 1). This validated the use of the *irg1-KI:GFP* as a reliable reporter for *irg1* expression without confounding effects from a knock-in knockout situation. Using a systemic challenge assay involving brain injection of zebrafish embryos[27], we analyzed gene expression changes post-*E. coli* infection to compare different *irg1*

genotypes for five genes involved in inflammatory signaling (*il1b, tnfa, irg1*) and immune cell increase (*mfap4, mpx*) (Supplementary Fig. 1). No distinction was observed between wild-type and heterozygous siblings, while homozygous *irg1*[bcz13] mutants showed significant increases in several genes, suggesting intact *irg1* function in

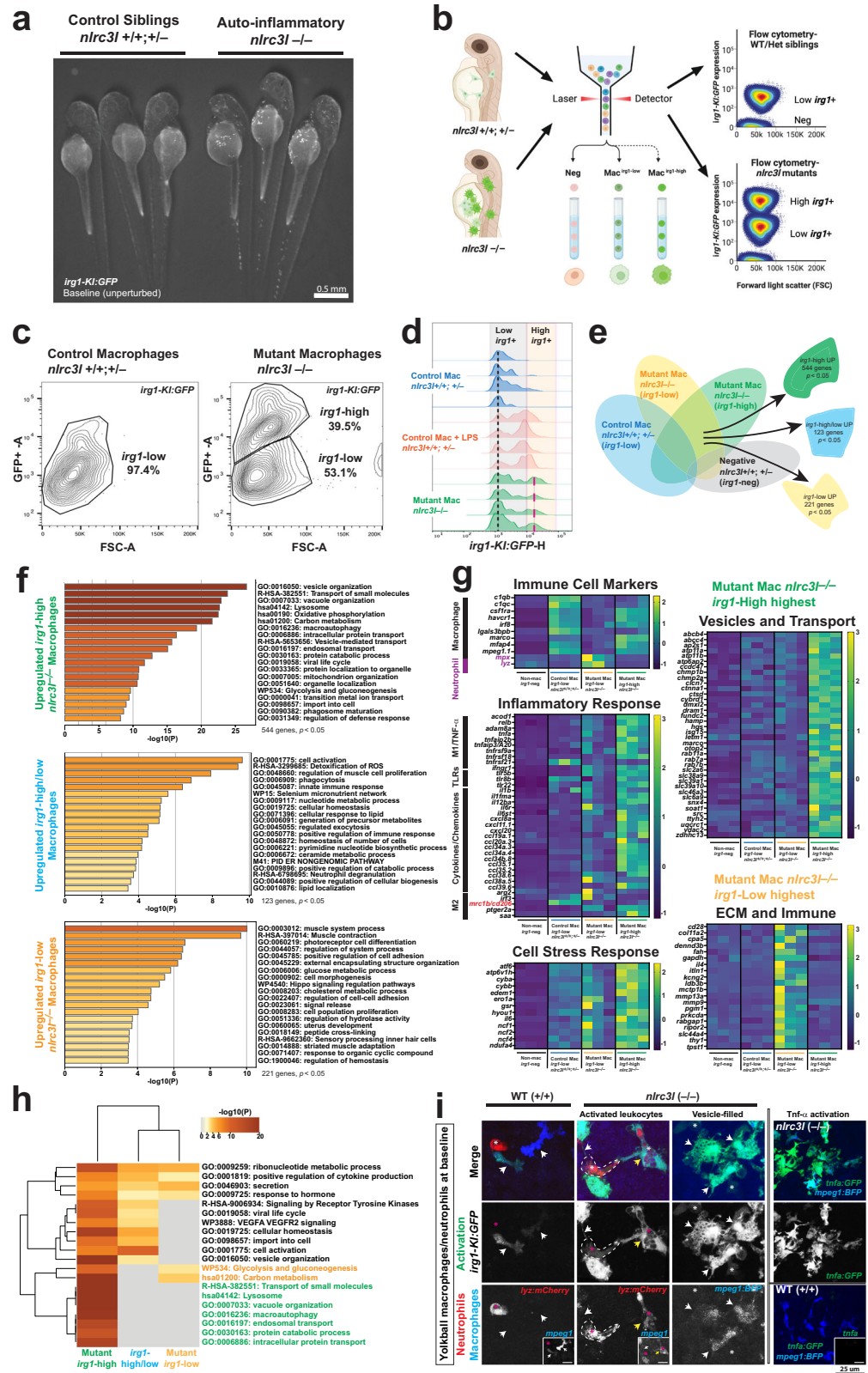

**Figure panels a–i** (Immune Cell Markers; Inflammatory Response; Cell Stress Response; Mutant Mac nlrc3l−/− irg1-High highest — Vesicles and Transport; Mutant Mac nlrc3l−/− irg1-Low highest — ECM and Immune)

heterozygous fish akin to wild type (Supplementary Fig. 1). We, therefore, used heterozygous GFP knock-in transgenic zebrafish, carrying only one copy of the transgene, to track normal endogenous *irg1* expression in all experiments.

Previous studies using whole-mount RNA in situ hybridization (WMISH) in zebrafish demonstrated a clear bimodal distribution of *irg1* gene expression in macrophages, with *irg1* specifically detected in

macrophages at high levels following infection or other immune challenges and normally no expression at baseline[17,24]. In contrast, an *irg1* fluorescent reporter using GFP, generated via Tol2 transposon-mediated transgenesis, showed expression in wild-type macrophages even at baseline, with a significant increase post-challenge in a macrophage-specific manner[22]. Considering the variability in number and location of GFP insertions by Tol2-mediated integration, we

**Fig. 2 | *irg1-KI:GFP* knock-in reporter enables isolation and characterization of macrophage subpopulations based on activation states. a** *irg1-KI:GFP* reporter brightly labels auto-inflammatory macrophages in *nlrc3l* mutants, enabling rapid and reliable sorting of live *nlrc3l* mutants from siblings using a low-magnification stereoscope. **b** Schematic for FACS sorting of macrophages from *nlrc3l* mutants and siblings with *irg1-KI:GFP*, isolating subpopulations based on GFP levels (high, low, negative). Created in BioRender. Shiau, C. (2026) https://BioRender.com/f3hmn0s. **c** Contour plots from FACS sorting showing the frequency of macrophage subpopulations with varying *irg1* levels (based on GFP expression). **d** Histogram depicting *irg1* expression levels inferred from GFP intensity in macrophages across different genotypes or conditions (showing four independent samples per category). Auto-inflammatory *nlrc3l* mutants exhibit higher *irg1* levels compared to LPS-activated controls. The *irg1*-low and *irg1*-high regions are color-coded. **e** Four-way Venn diagram comparing macrophage populations from bulk RNA-seq analysis to identify genes enriched in *nlrc3l* mutant subpopulations versus control macrophages. The color-coded shapes represent the gene sets that were compared to acquire the list of upregulated genes in each category. **f** Bar graphs of enriched biological pathways from Metascape analysis in *irg1*-high

mutant, *irg1*-low mutant, and control macrophage populations. **g** Heat maps show RNA-seq expression levels of selected genes and immune markers across macrophage populations. Mutant macrophages are enriched in inflammatory and stress response genes, while *irg1*-high and *irg1*-low mutant macrophages show differential expression in vesicle/transport and ECM/immune categories, respectively. *mrc1b/cd206* as highlighted in red text is especially downregulated in all mutant macrophages. **h** Heat map showing unsupervised hierarchical clustering analysis with NG-CHM, revealing shared and unique biological pathways significantly enriched in mutant macrophage subpopulations compared to control macrophages. **i** In vivo imaging of *irg1-KI:GFP* and *tnfa:GFP* in *nlrc3l* mutants and their control siblings at baseline reveals elevated *irg1* and induced *tnfa* expression in mutant macrophages (*mpeg1* +), which are morphologically altered and vesicle-filled, and accompanied by abnormal *irg1* expression in mutant neutrophils (*lyz* +) (demarcated by dotted lines and asterisks). The inset shows a single channel for *mpeg1:BFP* or *tnfa:GFP* as indicated. Arrows, indicator of macrophage; asterisks, indicator of neutrophils. All scale bars show 25 um. **f, h** DEGs analyzed for pathway enrichment were derived from DESeq2 analysis using default Wald test to obtain p-values corrected for multiple testing.

---

anticipated that the precise single-site GFP knock-in at the endogenous *irg1* locus would offer a more accurate reflection of *irg1* transcriptional levels. To directly address any differences between the GFP knock-in and Tol2-mediated GFP transgenesis, we generated a similar Tol2-mediated *irg1* reporter line, utilizing approximately 4.8 kb of regulatory sequence (Supplementary Fig. 2). Comparing the Tol2 and KI GFP reporters, we found that the KI reporter offered lower baseline GFP expression, a larger dynamic GFP range, and no off-target expression, while there was no difference in the reference macrophage BFP reporter used alongside the Tol2 or KI construct (Supplementary Fig. 3). These results also suggest fluorescent reporters are more sensitive in revealing very low levels of *irg1* that would be below detection by WMISH.

### Tracking macrophage activation in auto-inflammatory *nlrc3l* mutants reveals a MyD88 dependence

To systematically and continuously observe macrophage state changes in a living system without the need to provoke an immune response that is inherently variable from individual to individual, we utilized the *irg1-KI:GFP* reporter in *nlrc3l*[st73] mutants. These mutants exhibit auto-inflammatory macrophages at baseline in the absence of any overt immune challenges[23,24], but were previously distinguishable only by marker analysis in fixed samples, not in living individuals. Introducing a single copy of *irg1-KI:GFP* into *nlrc3l*[st73] heterozygous fish and crossing them with non-transgenic *nlrc3l*[st73] heterozygous individuals showed a Mendelian inheritance pattern, with about 50% of progeny inheriting the GFP knock-in reporter as expected for a single site GFP insertion. The *irg1-KI:GFP* reporter enabled rapid sorting of live *nlrc3l* homozygous mutants under a fluorescent stereoscope based on a strong GFP induction that is indicative of activated macrophages not observed in sibling and wild-type counterparts (Fig. 2a–d). These mutant macrophages also tend to aggregate on the yolk, thereby offering an ideal location to assay differential activation (Figs. 2a and 3a, b). A complete correspondence (100%) between the *irg1-KI:GFP* phenotype and *nlrc3l*[st73] genotype was confirmed by PCR-restriction-based genotyping after sorting.

To profile macrophage states in *nlrc3l* mutants, we used FACS to sort macrophages based on GFP levels (high, low, negative) in *irg1-KI:GFP* mutants and their heterozygous and wild-type siblings (Fig. 2b, c). Flow cytometry analysis revealed an *irg1*-high population in mutants surpassing LPS-activated control macrophages in *irg1* expression level (Fig. 2d). Bulk RNA-sequencing of sorted populations identified distinct gene sets upregulated in mutant macrophages, including immune activation, cell stress, and inflammatory response pathways (Fig. 2e–h). Sorted macrophages confirmed their identity with elevated expression of macrophage-specific markers (*c1qb, c1qc, csf1ra, havcr1,*

*irf8, lgals3bpb, marco, mfap4, mpeg1.1*)[28,29], although a few markers showed inconsistent upregulation in mutant macrophages (Fig. 2g). Functional annotation highlighted increased expression of vesicle- and transport-related genes in *irg1*-high mutant macrophages, consistent with abnormal vesicle-filled phenotype observed in mutant macrophages with high *irg1* expression (Fig. 2i), while *irg1*-low mutant macrophages showed enrichment in ECM and cell adhesion pathways (Fig. 2f–h). Although *irg1*-low mutant macrophages showed similarly low *irg1* levels as control sibling macrophages, they modestly expressed pro-repair M2-like genes (*arg2, irf3, ptger2a*)[28,29], and significantly downregulated the mannose receptor *mrc1b/cd206*, a M2-associated marker[30,31]. A substantial downregulation was also seen in *irg1*-high mutant macrophages (Fig. 2g). Additionally, *irg1*-low mutant cells displayed an enrichment for neutrophil-specific markers (*mpx, lyz*)[29] (Fig. 2g). To directly confirm these cellular changes in vivo, we imaged double transgenic zebrafish expressing both neutrophil and *irg1-KI:GFP* reporters in vivo, and observed weak *irg1* expression in mutant neutrophils in stark contrast to the absence of *irg1* in control sibling neutrophils (Fig. 2i), suggesting that some *irg1*-low mutant cells are likely neutrophils exhibiting altered states. Furthermore, in vivo imaging revealed an unusually strong induction of inflammatory cytokine *tnfa* expression at baseline in almost all macrophages using the *tnfa:GFP* transcriptional reporter[32], which is predominantly macrophage cell-autonomous as it can be reversed by restoring macrophage *nlrc3l* expression (Supplementary Fig. 4). Weak *tnfa* in select few neutrophils was also observed in *nlrc3l* mutants, while no *tnfa* was detected in control sibling counterparts (Fig. 2i and Supplementary Fig. 4). These findings highlight substantially altered macrophage and neutrophil states in *nlrc3l* mutants, and similarly low *irg1* immune cell expression in mutants and siblings can render significantly different cellular states.

Additionally, unsupervised hierarchical clustering[33] of cell-specific expression levels for a target gene set, followed by protein network analysis[34], identified significantly elevated TNF-alpha and non-canonical NF-κB pathways in *nlrc3l* mutant immune cells (Supplementary Fig. 5). This gene set comprised 55 genes known to be enriched in macrophages, neutrophils, or immune responses (Supplementary Fig. 5). Analysis was performed separately on sorted macrophages (high *irg1-KI:GFP*+ cells) and sorted neutrophils (*lyz:GFP*+ cells) from *nlrc3l* mutants compared to control sibling counterparts (Supplementary Fig. 5). These results revealed notable upregulation of inflammation-associated genes in *nlrc3l* mutant macrophages, particularly TNF-alpha and components of the non-canonical NF-κB pathway (*nfkb2, traf2b, nik/map3k14a, relb*) (Supplementary Fig. 5). Interestingly, mutant neutrophils exhibited expression of genes we found typically restricted to zebrafish macrophages (*mfap4, tnfa, tlr22,*

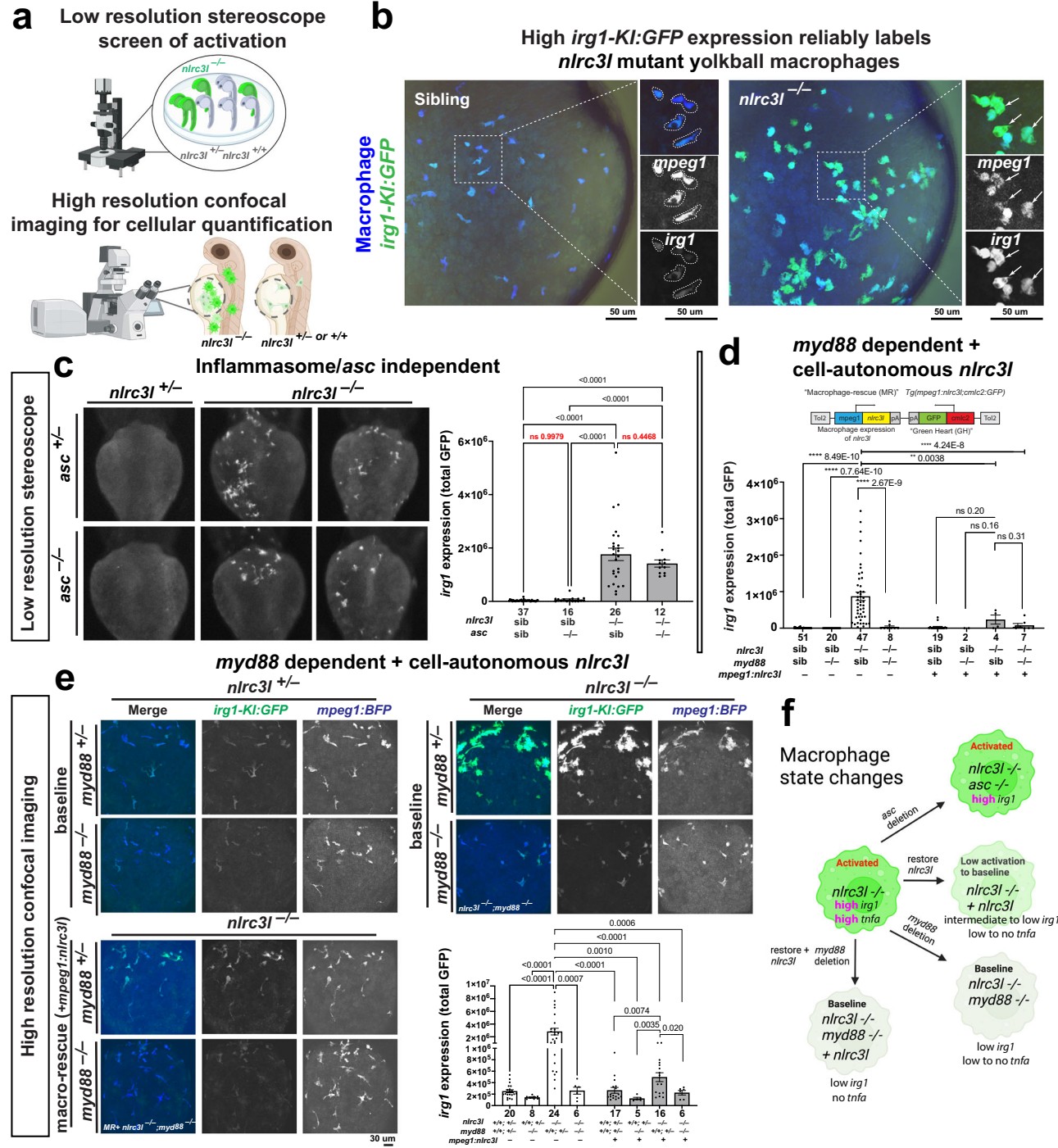

tnfaip8l2b, tnfrs1b/TNFR2, traf6), suggesting a phenotypic shift resembling activated macrophages (Supplementary Fig. 5).

Given significant upregulation of TNF-alpha and NF-kB pathway genes in mutant macrophages that may account for their chronic inflammatory state, we aimed to investigate the role of the central mediator MyD88 by analyzing nlrc3l/myd88 double mutants (Fig. 3a, b, d). MyD88 is crucial for TNF-alpha production and NF-kB pathway activation via Toll-like receptors (TLRs) and interleukin-1 receptors (IL-1Rs), and for non-canonical NF-kB signaling through TNF receptor-associated factors (TRAFs)[35–37]. In nlrc3l mutants, restoring macrophage nlrc3l expression, deleting myd88, or both, reversed the inappropriate macrophage activation, as evidenced by irg1 and tnfa downregulation, with the combination fully restoring a normal macrophage state on the yolk ball (Fig. 3d, e) and in the trunk

(Supplementary Figs. 4, 6). The increase in irg1 expression in activated macrophages corresponded with the induction of macrophage tnfa, while restoring a normal macrophage state in nlrc3l mutants reduced irg1 to baseline levels (Fig. 3d, e) and eliminated tnfa expression (Supplementary Figs. 4, 6), validating irg1 and tnfa as consistent markers of macrophage activation in zebrafish, in agreement with previous reports that examined each marker individually[17,38,39]. Furthermore, because several Nod-like receptors are known to interact with the adaptor protein called apoptosis-associated speck like protein containing a CARD (also known as ASC) to form cytosolic multiprotein complexes called inflammasomes, responsible for activation of inflammatory signaling, we explored whether the inappropriate macrophage activation in nlrc3l mutants was ASC-dependent. We generated nlrc3l/asc double mutants that showed no change from the nlrc3l

**Fig. 3 | Screening yolkball macrophages distinguishes macrophage states at scale using *irg1-KI:GFP* in zebrafish, enabling epistasis analysis that reveals a genetic interaction between *nlrc3l* and *myd88*. a** Schematic depicting the screening of mutants showing inappropriate macrophage activation as reflected by high *irg1* expression, indicated by intense GFP expression in macrophages. Observations can be made rapidly at low resolution using a fluorescent stereoscope or at high resolution using a confocal microscope for cellular resolution at 2 or 3 dpf. Created in BioRender. Shiau, C. (2026) https://BioRender.com/fmrpgzm. **b** Confocal imaging shows the dual labeling system of yolkball macrophages at cellular resolution in an *nlrc3l* sibling or mutant, using *irg1-KI:GFP* for immune activation and *mpeg1:BFP* as a pan-macrophage marker. High magnification images show sibling macrophages delineated by dotted lines, indicating baseline low *irg1-KI:GFP* expression characteristic of macrophages at homeostasis. **c** Epistasis analysis of single and double mutants (*nlrc3l* and *asc*) using embryo yolkball screening of *irg1* GFP knock-in expression under a stereoscope (left images). Quantification of *irg1* expression in *nlrc3l/asc* double mutants showed no modification from the single *nlrc3l* mutant phenotype of inappropriate macrophage activation using *irg1-KI:GFP*, suggesting no interaction between *nlrc3l* and *asc*. **d-e** Epistasis analysis of

single and double mutants (*nlrc3l* and *myd88*) with macrophage rescue ("macro-rescue", construct shown in cartoon), examined using stereoscope screening (**d**) or confocal imaging (**e**). *nlrc3l/myd88* double mutants reverse the single *nlrc3l* mutant macrophage activation phenotype, suggesting that *myd88* acts downstream of *nlrc3l* in opposing pathways. Confocal imaging, more sensitive to GFP, detects differences in *irg1-KI:GFP* expression between macrophage-rescued *nlrc3l* mutants and controls, sometimes visible by eye but not quantifiable in low-resolution stereoscope images. Both macrophage rescue and addition of *myd88* mutation reverse the single *nlrc3l* mutant phenotype, but macrophage rescue shows some variation, suggesting non-cell-autonomous factors or possible differences in the expression level of the rescue construct influencing the degree of the phenotypic rescue. **f** Overview cartoon of macrophage state changes in *nlrc3l* mutants due to different genetic manipulations. Created in BioRender. Shiau, C. (2026) https://BioRender.com/qyvz4k1. See also Supplementary Figs. 4, 6, and 9 for data related to *tnfa* expression. **c–e** Numbers below bar indicate *n*, number of embryos analyzed. Each experiment was repeated at least twice. Data are presented as mean values +/- SEM. Statistical significance was determined using one-way ANOVA, followed by multiple comparisons.

mutant activation phenotype or impaired muscle repair after injury, thereby suggesting *nlrc3l* functioned independently of *asc* in this context (Fig. 3c and Supplementary Fig. 7a–d). Given that MyD88 mediates TLR signaling responsible for bacterial recognition and response[35], we evaluated macrophage activation in *nlrc3l* mutants under germ-free versus conventionally raised conditions (Supplementary Fig. 7e–g). Our findings showed that while commensal microbes contributed, *nlrc3l* mutant macrophages themselves retained substantial intrinsic activation (Supplementary Fig. 7e–g). In agreement with this, full rescue of inappropriate macrophage activation in *nlrc3l* mutants required both macrophage rescue and *myd88* deletion, as neither intervention alone produced complete rescue consistently (Supplementary Fig. 6). This indicates contributions from both cell-intrinsic defects and likely non-cell-autonomous signals, supported by the need to delete *myd88*, a gene mediating TLR signaling. Taken together, these findings highlight the intricate nature of the inappropriate macrophage activation in *nlrc3l* mutants, where MyD88-dependent signaling, microbial cues, and internal cellular dysregulation collectively contribute to their chronic inflammatory activation state, in addition to the altered state of neutrophils. These results implicate both cell-autonomous and non-cell-autonomous mechanisms in the inappropriate immune activation.

## Chronic macrophage activation impairs response to acute skeletal muscle injury

We focused on tracking macrophage states during tissue injury due to the controlled and localized perturbation injury provides, which progresses through distinct phases from inflammation to resolution, with macrophages playing a central role[6,11,40]. This is particularly evident in skeletal muscle injury where macrophages are thought to transition from a pro-inflammatory to a pro-repair state in both mammals and zebrafish[1,6,41,42]. Despite this, the precise cellular mechanisms by which macrophages coordinate an effective muscle injury response and repair in vivo, especially their dynamic transformations into diverse subtypes, have remained elusive through static or single-cell transcriptome analysis in previous studies lacking in vivo dynamic imaging. To address this gap, we developed an acute skeletal muscle injury model in zebrafish involving a major tail amputation (Fig. 4a). We found this model to trigger robust immune responses, where both *irg1* (Fig. 4b and Supplementary Fig. 8) and *tnfa* (Supplementary Fig. 9) are highly induced in macrophages in wild-type zebrafish post-injury, and significant skeletal muscle repair (Fig. 5), distinct from previous zebrafish tail injury studies which focus on less disruptive tail fin truncation and regeneration without extensive muscle injury[11,40,43,44].

We used in vivo longitudinal and timelapse imaging to monitor individual zebrafish embryos, employing a dual labeling system that

marks all macrophages and tracks their activation status (Fig. 4b and Supplementary Fig. 10). This approach revealed a consistent macrophage response to injury in wild-type zebrafish (Fig. 4b, c). Initially, macrophages are recruited to the injury site, peaking during the inflammatory phase between 24–30 h post-amputation (hpa) (Fig. 4b, c). Concurrently, there is robust expression of *irg1* (Fig. 4b, c and Supplementary Fig. 8) and significant *tnfa* induction (Supplementary Fig. 9) in a subset of macrophages at the cut site. Timelapse imaging captured dynamic *irg1* induction in macrophages from 5 to 14 h post-injury, both in those migrating to and already present at the cut site (Supplementary Fig. 10 and Supplementary Movie 1). Macrophage recruitment primarily occurred via migration outside circulation: dorsally along the dorsal longitudinal anastomotic vessel (DLAV), ventrally through the caudal vein plexus (CVP), or through the vasculature-free midline (Supplementary Fig. 10 and Supplementary Movie 1). Most macrophages at the cut site within 24 hpa exhibited intermediate to high GFP expression, indicating robust activation with elevated *irg1* expression (Fig. 4b). Subsequently, macrophages migrated away from the injury site, redistributing throughout the body during immune deactivation, coinciding with tissue repair and regeneration from 48 hpa onwards (Fig. 4b and Supplementary Fig. 8). Timelapse imaging showed macrophages and neutrophils engaging in reverse migration away from the injury site, and dynamically surveying the repairing tissue, accompanied by a significant macrophage *irg1* downregulation (Supplementary Movie 2). By 96 hpa, complete tissue repair was evident with the tail muscle fully healed, though regeneration to original length did not occur (Fig. 4b).

In stark contrast, chronically activated mutant macrophages maintained consistently high levels of *irg1* and *tnfa* expression both at homeostasis and post-injury (Fig. 4c and Supplementary Figs. 9, 11, 12), yet were deficient from the initiation of the inflammatory response to the acute muscle injury (Fig. 4d–g). Although mutants had comparable macrophage numbers to siblings at baseline, they showed no significant increase post-injury, unlike controls which about tripled in number (Fig. 4d). Mutants also had elevated neutrophil counts at baseline similar to post-injury levels seen in controls (Fig. 4d). Moreover, these mutants did not form the characteristic large and dense macrophage cluster at the cut site observed at ~24 hpa; instead, they exhibited a void in this region where removal of injured and dying cells, and regeneration of myocytes were expected (Fig. 4e, Supplementary Figs. 11, 12, 13). These mutant macrophages interacted as smaller groups of cells that persisted even through the expected resolution phase at 72 hpa and 96 hpa (Supplementary Fig. 12), as depicted in the graphical plots comparing wild-type and mutant macrophage responses (Fig. 4c). Furthermore, while *tnfa* expression was restricted to macrophages at the cut site in control animals (~31%

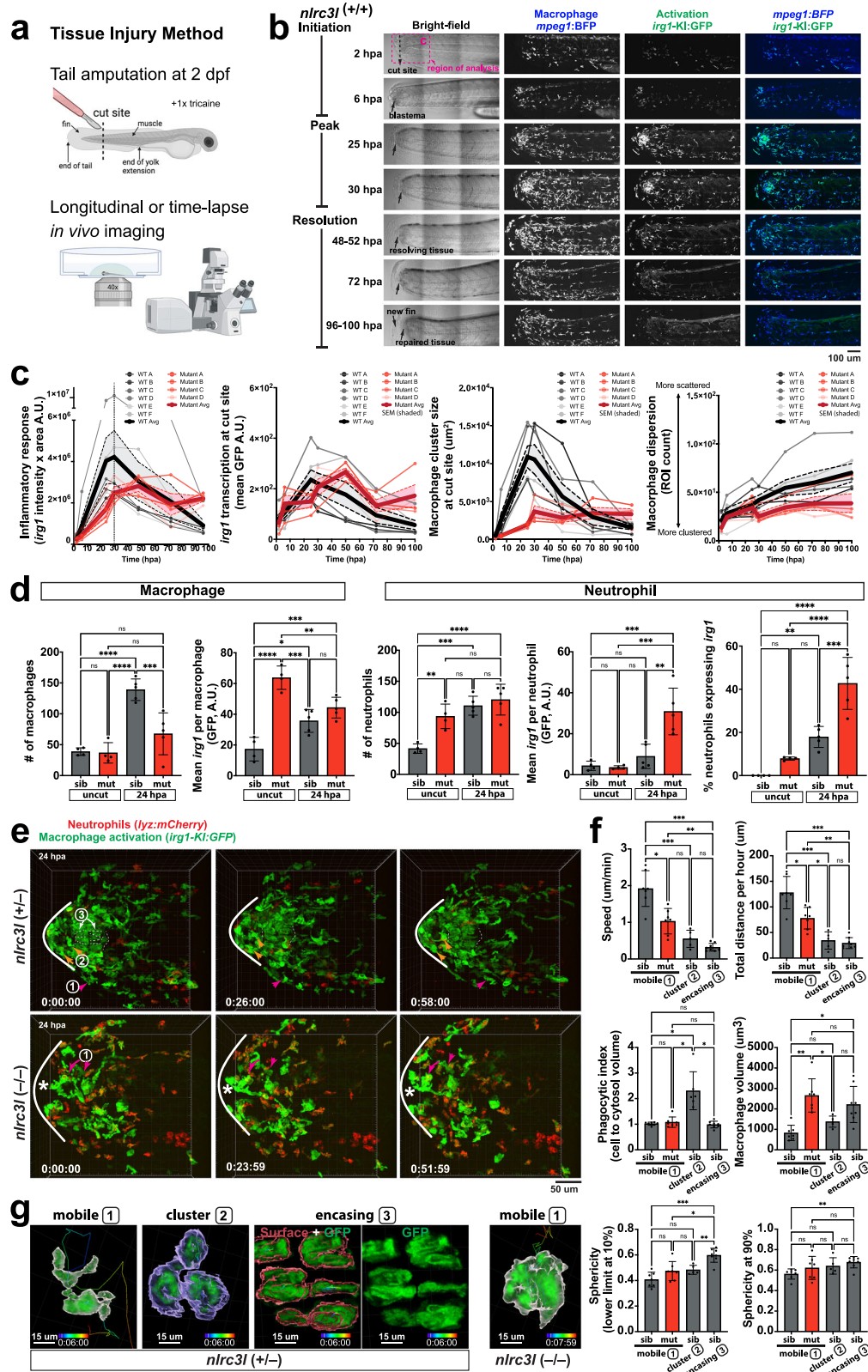

of macrophages post-injury and nearly none at baseline), nearly all macrophages in the mutants, including those in the caudal hematopoietic tissue (CHT), showed uniformly high *tnfa* expression (~90% at baseline and >95% post-injury) (Supplementary Fig. 9). Single cell *irg1* profiling reveals distinct differences in the patterns of immune cell activation between mutant and control macrophages and neutrophils (Supplementary Fig. 11). Mutant macrophages at baseline and post-

injury consistently exhibit high *irg1* expression levels comparable to or generally higher than activated control macrophages post-injury (Fig. 4d and Supplementary Fig. 11). Additionally, a small fraction of neutrophils in mutants show an atypical induction of *irg1* and *tnfa* expressions at baseline compared with control neutrophils which express neither gene at baseline (Fig. 2i and Supplementary Figs. 4, 11). After injury, *irg1* expression is detectable in a small number of

**Fig. 4 | In vivo longitudinal and time-lapse imaging tracks macrophage activation during acute injury response, revealing severe deficiencies in auto-inflammatory *nlrc3l* mutant. a** Schematic of injury model involving tail amputation at 2 dpf, where the cut is positioned ~ tenth somite down from the end of yolk extension (or the midpoint between yolk extension and tail end). Created in BioRender. Shiau, C. (2026) https://BioRender.com/rxau1o9. **b** In vivo confocal images from a longitudinal study of wild-type embryos (*n* = 6) post-injury, representative data from one individual. **c** Longitudinal analysis plots comparing wild-type (*n* = 6) and *nlrc3l* mutants (*n* = 4) during injury from initiation to resolution. Wild-type individuals are plotted in shades of black, and *nlrc3l* mutants in shades of red. The dotted box in **b** highlights the quantified region. Thicker plot lines indicate group averages, with SEM shown by shaded regions. **d** Quantification of macrophage and neutrophil responses between *nlrc3l* control siblings (sib, heterozygous and wild-type, *n* = 4 uncut and 5 cut) and homozygous mutants (mut, *n* = 4 uncut and 5 cut) at baseline and peak inflammation ( ~ 24 hpa). Each data point represents

an individual animal. **e** Representative static time series from time-lapse confocal imaging of double transgenic zebrafish labeling macrophages and neutrophils at 24 hpa. Images were captured every 2 min using a 40x objective. Three distinct macrophage subtypes (1, 2, 3) are labeled by numbers and arrows, with mutants lacking subtypes 2 (cluster) and 3 (muscle-encasing). For more details, see Supplementary Movies 3 and 5 (timelapse imaging) and Supplementary Movies 4 and 6 (rendered movies with subtype annotations). **f** Quantification of macrophage cell behavior and morphology along with representative microscopy images of the three subtypes (1: mobile, white; *n* = 7 sib cells; *n* = 7 mut cells; 2: cluster, blue; *n* = 5 sib cells; 3: muscle-encasing, red, *n* = 8 sib cells)(shown in **g**), shows differences between *nlrc3l* mutants and control siblings. See also Supplementary Fig. 12 for supporting data. Statistical significance for all scatter bar plots was determined using a one-way ANOVA followed by multiple comparisons: *, *p* < 0.05, **, *p* < 0.01; ***, *p* < 0.001; ****, *p* < 0.0001; ns, not significant. A.U., arbitrary units. Each experiment was repeated at least twice. Data are presented as mean values +/- SD in **d** and **f**.

neutrophils—on average 7.6% in siblings and 10.7% in *nlrc3l* mutants at 20 hpa (and 4.3% at baseline in mutants, none in siblings) (Supplementary Fig. 11c). These data show that most macrophages are bright *irg1-Kl:GFP+* after injury and in baseline mutants, while neutrophils rarely cross the detectable threshold at 30 A.U. (Supplementary Fig. 11b, c). Thus, despite low-level expression in a few neutrophils, *irg1-Kl:GFP* remains a reliable reporter for tracking macrophage-specific responses. Overall, when detectable in injured controls or in mutants, whether at baseline or post-injury, neutrophil *irg1* or *tnfa* expression is significantly weaker and limited to a select number of neutrophils, in contrast to the broad and strong induction of these genes in activated macrophages (Fig. 4d and Supplementary Figs. 9, 11). This data is also consistent with the previously discussed bulk RNA-seq and imaging results in Fig. 2g, i where *irg1*-low cells from *nlrc3l* mutants include that of the neutrophil cell type besides macrophages.

Most notably, during the inflammatory phase at ~ 24 hpa, time-lapse imaging revealed distinct cellular behavioral differences among activated macrophages at the cut site in wild-type and control siblings post-injury (Fig. 4e–g and Supplementary Movies 3 and 4). Macrophages at the injury site that displayed dynamically irregular and migratory cell shapes with high mobility (increased speed and distance traveled, Fig. 4f) are categorized as subtype 1 "mobile". Other macrophages in close proximity to the repairing tail fin with increased vesicular content and higher phagocytic index (cell-to-cytosol volume ratio), indicating large intracellular vesicles such as phagosomes or endosomes (Fig. 4f) are categorized as subtype 2 "cluster". The final macrophage population at the injury site exhibited largely stationary behavior with an elongated and columnar morphology oriented along the muscle fibers that are aligned with the body anterior-posterior axis is referred to as subtype 3 "encasing" herein (Fig. 4e–g, Supplementary Fig. 13). Subtype 3 macrophages encircle myocytes and take on a muscle-cell-like shape, showing the largest cell size and greater overall sphericity, particularly in the lowest 10th percentile of the population distribution, indicating less variability in cell shape than the other subtypes (Fig. 4f, g, Supplementary Fig. 13). In mutants, our dynamic tracking of the injury response indicated an apparent absence of subtypes 2 and 3 macrophages, which were consistently observed in all control animals (*n* = 4 for each genotype, Fig. 4e–g and Supplementary Movies 3–6). These observations underscore the behavioral and morphological cellular diversity that may define distinct macrophage functions post-injury, which are compromised in chronically activated mutants.

**Chronic macrophage activation linked with severe downregulation of mannose receptor causes unresolved muscle injury, reversible by reinstating a normal state**

To delve into the functionally altered macrophages, we explored whether injured skeletal muscle in *nlrc3l* mutants could effectively

repair and regenerate. To visualize muscle tissues, we used muscle actin antibody or actin-binding phalloidin staining, enabling co-localization of macrophages with injured or repairing myocytes (Fig. 5 and Supplementary Fig. 13). Detailed examination of macrophage-myocyte interactions identified three functional macrophage subtypes corresponding to those previously discussed: 1) mobile surveying cells, 2) clustering cells that appear likely phagocytic for clearing injured and dying cells, and 3) elongated stationary cells encasing injured or repairing muscle cells (Supplementary Fig. 13 and Supplementary Movies 3 and 4). Immediately after tail amputation (within 1 hpa), muscle injury was comparable across all genotypes (Supplementary Fig. 13). However, by 24 hpa, mutant macrophages inadequately infiltrated the cut site, although recruitment of neutrophils appeared normal, lacking both clustering subtype 2 and encasing subtype 3 (Fig. 5a, Supplementary Fig. 13 and Supplementary Movie 4). In contrast, sibling controls displayed macrophages that formed clusters appearing likely phagocytic (subtype 2) in the most severely damaged region of the injury site, and encasement of seemingly intact muscle fibers by macrophages (subtype 3) in the adjacent area (Supplementary Fig. 13). By 48 hpa, control siblings had mostly cleared injured cells and debris, resulting in a clean-cut tail end with a well-defined tissue boundary, and only occasional, if any, non-intact muscle fibers remaining (Fig. 5c, d and Supplementary 13). In *nlrc3l* mutants, although more macrophages were recruited to the injury site, they exhibited limited engulfment of damaged muscle cells, leaving significant cellular damage and debris compared to controls at the injury site (Fig. 5c–f). By 96 hpa, control siblings showed fully repaired muscle with regrowth at the tail tip, forming a pointed end (Fig. 5d-f). In contrast, mutants retained unresolved tissue and failed to form the expected muscle tip at the tail end (Fig. 5c–f). This coincided with deficient activation of satellite cells[12,45] critical for regenerating myocytes during active muscle repair[1,46], as evidenced by a significant reduction in the emergence of Pax7+ satellite cells[46,47] at the injury site observed at both 24 hpa and 48 hpa in *nlrc3l* mutants (Fig. 5a). These findings underscore that the impaired initial inflammatory response by macrophages in these mutants significantly hindered the ability of macrophages to phagocytose and support injured muscle repair and regeneration.

The loss of essential macrophage functional subtypes 2 and 3, critical for clearance of cellular damage and muscle support, in the chronically activated mutant macrophages prompted us to investigate whether similar outcomes could result from macrophage ablation. To address this, we utilized *irf8* mutants[48], known to lack macrophages in steady-state zebrafish embryos at stages concurrent with our injury study. Interestingly, although initially devoid of macrophages, *irf8* mutants after injury showed the presence of macrophages (Supplementary Fig. 14), likely due to injury-triggered hematopoiesis that is *irf8* independent. Despite this partial macrophage recovery, muscle

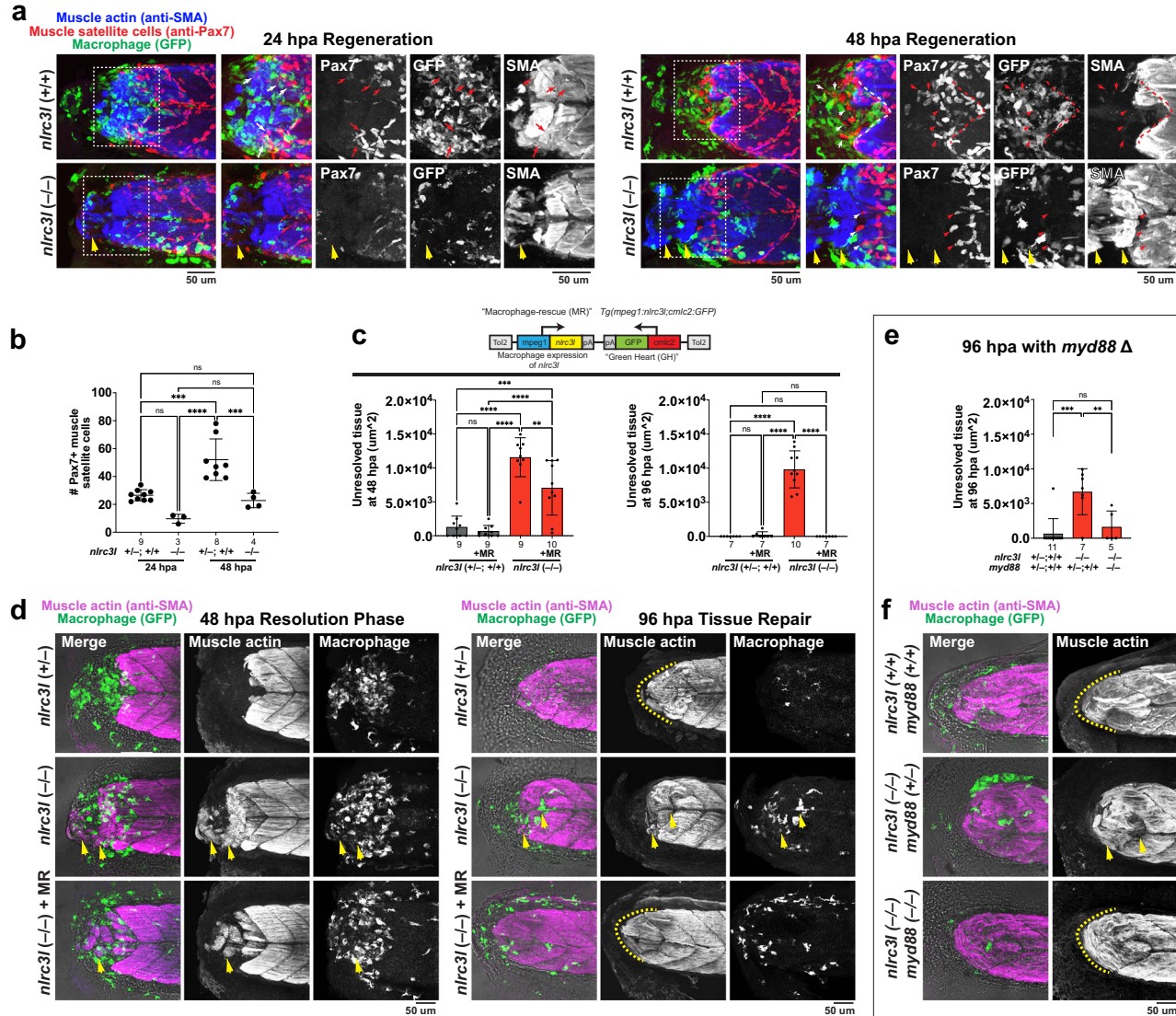

**Fig. 5 | Chronic inflammatory activation of macrophages impairs acute muscle injury repair, reversible by restoring wild-type macrophages or eliminating *myd88*. a** Whole mount immunohistochemistry (IHC) analysis of injured sibling and mutant embryos reveals significantly fewer Pax7+ satellite cells (white or red arrows) in *nlrc3l* mutants at both 24 hpa and 48 hpa, indicating impaired muscle regeneration. Dotted box show region magnified in panels on the right showing merged and corresponding single-channel images. Yellow arrows mark unresolved injured muscle in *nlrc3l* mutants. **b** Quantification of Pax7+ satellite cells shows a significant reduction in *nlrc3l* mutants at 48 hpa, a critical period when satellite cell emergence and muscle regeneration are most active. **c** Scatter bar charts corresponding to data images in **d** show the area of unresolved tissue (yellow arrows) during the resolution phase at 48 hpa and 96 hpa. The yellow dotted line marks the repaired muscle tip at the end of the tail. *nlrc3l* mutants with the macrophage-rescue construct show a significant reversal of impaired muscle regeneration.

**d** Whole mount IHC analysis at 48 hpa and 96 hpa compares muscle injury repair in *nlrc3l* mutants and their siblings, with and without the macrophage-rescue (MR) construct. Merged channels are overlayed on brightfield images, alongside corresponding single-channel images. **e** Quantification of unresolved tissue area at 96 hpa compares tissue repair between single *nlrc3l* mutants, double *nlrc3l/myd88* mutants, and their control siblings. **f** Whole mount IHC analysis at 96 hpa shows that the addition of a *myd88* deletion reverses injury impairment in *nlrc3l* mutants. Double *nlrc3l/myd88* mutants exhibit muscle repair comparable to that of their control siblings. For all plots (**b**, **c**, **e**), each data point represents an individual embryo, and data are presented as mean values +/- SD. Statistical significance was determined using a one-way ANOVA followed by multiple comparisons: **, *p* < 0.01; ***, *p* < 0.001; ****, *p* < 0.0001. *n*, number below bar indicates number of embryos analyzed.

injury in macrophage-depleted *irf8* mutants exhibited persistent impairment in repair and regeneration from 48 to 96 h post-amputation, resembling the phenotype of chronically activated *nlrc3l* mutants (Supplementary Fig. 14). These findings underscore that both chronic activation and depletion of macrophages lead to a similar impaired muscle repair outcome, suggesting a comparable loss of macrophage function during acute injury. To directly assess whether the impaired muscle injury repair was due to altered mutant macrophages, we restored normal macrophages devoid of inflammatory signatures by rescuing macrophage wild-type *nlrc3l* expression or deleting *myd88*

signaling. Both approaches effectively reinstated a normal macrophage state, reducing or eliminating *irg1* and *tnfa* inductions (Fig. 3, and Supplementary Figs. 4, 6), and significantly facilitated recovery of injured muscle by 96 hpa to a level indistinguishable to controls (Fig. 5c–f). By contrast, genetic deletion of an inflammasome adaptor protein *asc* failed to restore a normal macrophage state or muscle injury repair in *nlrc3l* mutants (Fig. 3c, and Supplementary Fig. 7b–d), supporting that their chronic activation is *asc*-independent.

To investigate mechanisms underlying functional deficiencies of chronically activated mutant macrophages post-injury, we performed

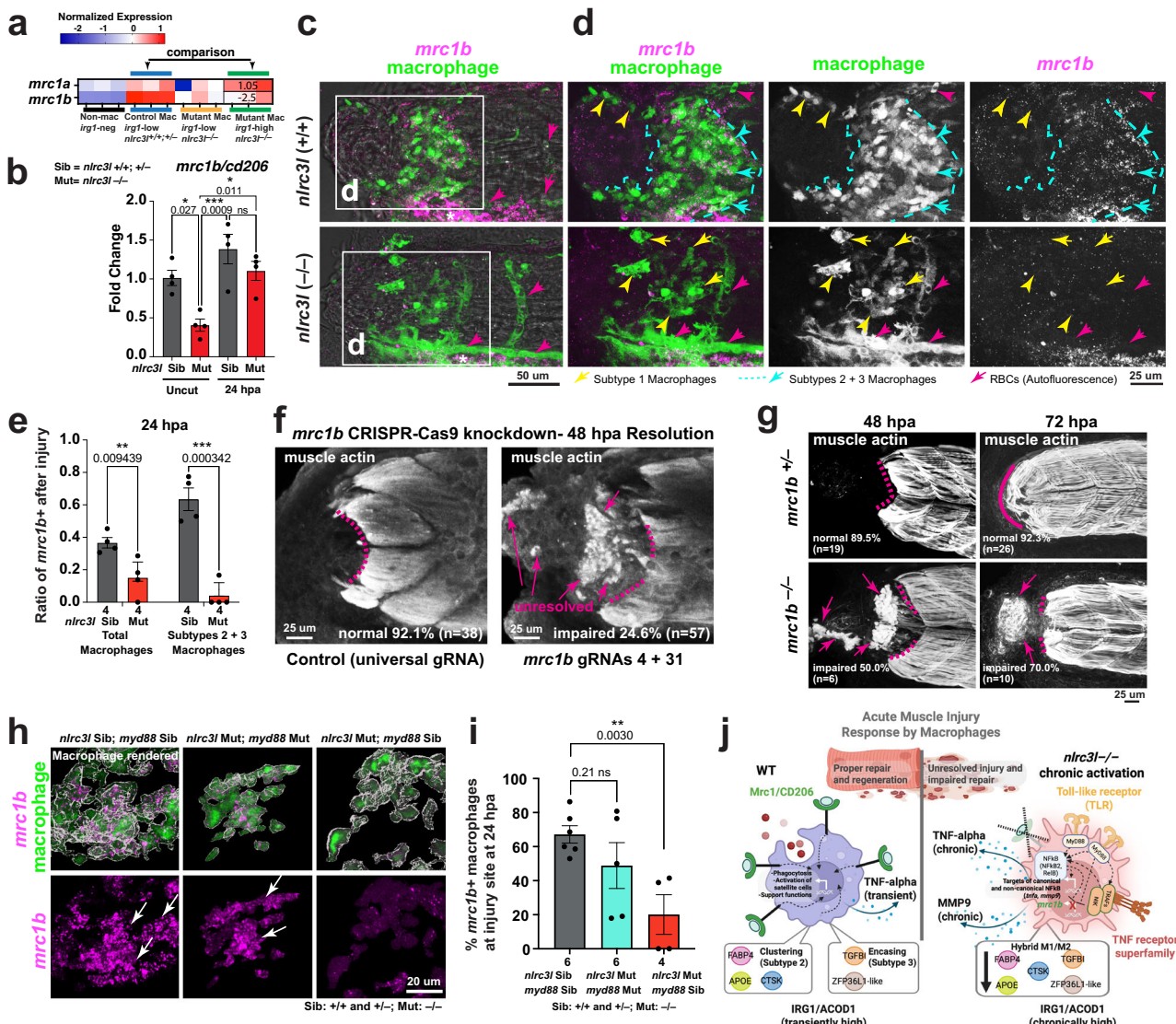

**Fig. 6 | Macrophage subtype loss and impaired muscle repair in chronically activated *nlrc3l* mutants are linked to *mrc1b* downregulation, reversible by *myd88* deletion. a** Heatmap of zebrafish Mrc1 from the same RNA-seq dataset as in Fig. 2. *mrc1a* expression unchanged (p-adjusted = 0.9), while *mrc1b* shows a significant 2.5-fold-reduction (p-adjusted = 0.0023) in *nlrc3l* mutant macrophages relative to controls. Statistics are from DESeq2 Wald test with multiple testing. **b** qPCR analysis of whole-embryo transcription of *mrc1b*, also shown in Supplementary Fig. 15, shows mean values +/- SEM and significance from one-way ANOVA test with multiple comparisons. **c** Whole-mount *mrc1b* HCR in situ hybridization in wild-type embryos expressing *irg1-KI:GFP* (used as a macrophage marker) reveals dense *mrc1b* expression in subtypes 2 and 3 (blue arrows and dotted blue region). **d** Higher magnification of c demonstrate lack of *mrc1b* in *nlrc3l* mutant macrophages. Yellow arrows indicate subtype 1, which have little to undetectable *mrc1b*. Magenta arrows show auto-fluorescent red blood cells in the GFP channel. Asterisk, caudal hematopoietic tissue. **e** Quantification of *mrc1b*+ macrophages (4 animals per genotype) shows significant reduction particularly in subtypes 2 and 3 in *nlrc3l* mutants. Data shows mean values +/- SEM; unpaired t-tests with FDR approach.

**f** CRISPR-Cas9 knockdown of *mrc1b* phenocopies *nlrc3l* mutants. Lower right, percentage of animals (*n*) and total *n* analyzed showing normal repair for control (universal gRNA + Cas9 or uninjected), or deficient repair for *mrc1b* gRNAs + Cas9 injected at 48 hpa and 96 hpa. Magenta dotted lines, border of intact muscle fibers. Arrows, unresolved muscle. **g** *mrc1b⁻ᐟ⁻* mutants (sa18640) (see also Supplementary Fig. 16) show impaired muscle repair (arrows). **h** Whole-mount *mrc1b* HCR in situ hybridization (magenta) counterstained with anti-GFP for saturation-level detection of *irg1-KI:GFP* used as a macrophage marker. Double *nlrc3l; myd88* mutants show recovery of *mrc1b* (bottom row, magenta). Top row, merged fluorescent channels of macrophages (GFP +) and *mrc1b* signals (magenta) with IMARIS rendering (white surfaces) at the cut site. **i** Quantification of *mrc1b*+ macrophages at cut site. Data shows mean values +/- SEM; two-tailed student T-tests. **j** Proposed mechanism of macrophage response to acute muscle injury. Chronically activated macrophages predominantly adopt a hybrid M1/M2 state with reduced expression of *mrc1b* and reparative genes that define macrophage subtypes 2 and 3. Created in BioRender. Shiau, C. (2026) https://BioRender.com/yvxoz4x.

---

bulk RNA-seq on FACS-sorted macrophages from uninjured and 24 hpa zebrafish embryos. We identified several injury-induced upregulated immune cell genes, including significantly elevated matrix metalloproteinase-9 (*mmp9*), known as an inflammatory proteinase upregulated in various diseased and inflamed tissues that can facilitate tissue remodeling, cell migration and inflammatory signaling through cleaving and activating cytokines and other secreted proteins[49–51], and

*relb*, a non-canonical NF-kB pathway transcription factor[52] (Supplementary Fig. 15). qPCR analysis of additional animals confirmed excessive *mmp9* expression in *nlrc3l* mutants at baseline and post-injury, implicating dysregulated remodeling activities that may potentially hinder macrophage ability to respond and phagocytose injured cells (Supplementary Fig. 15). Additionally, *relb* was significantly increased in *nlrc3l* mutants at baseline and even higher post-

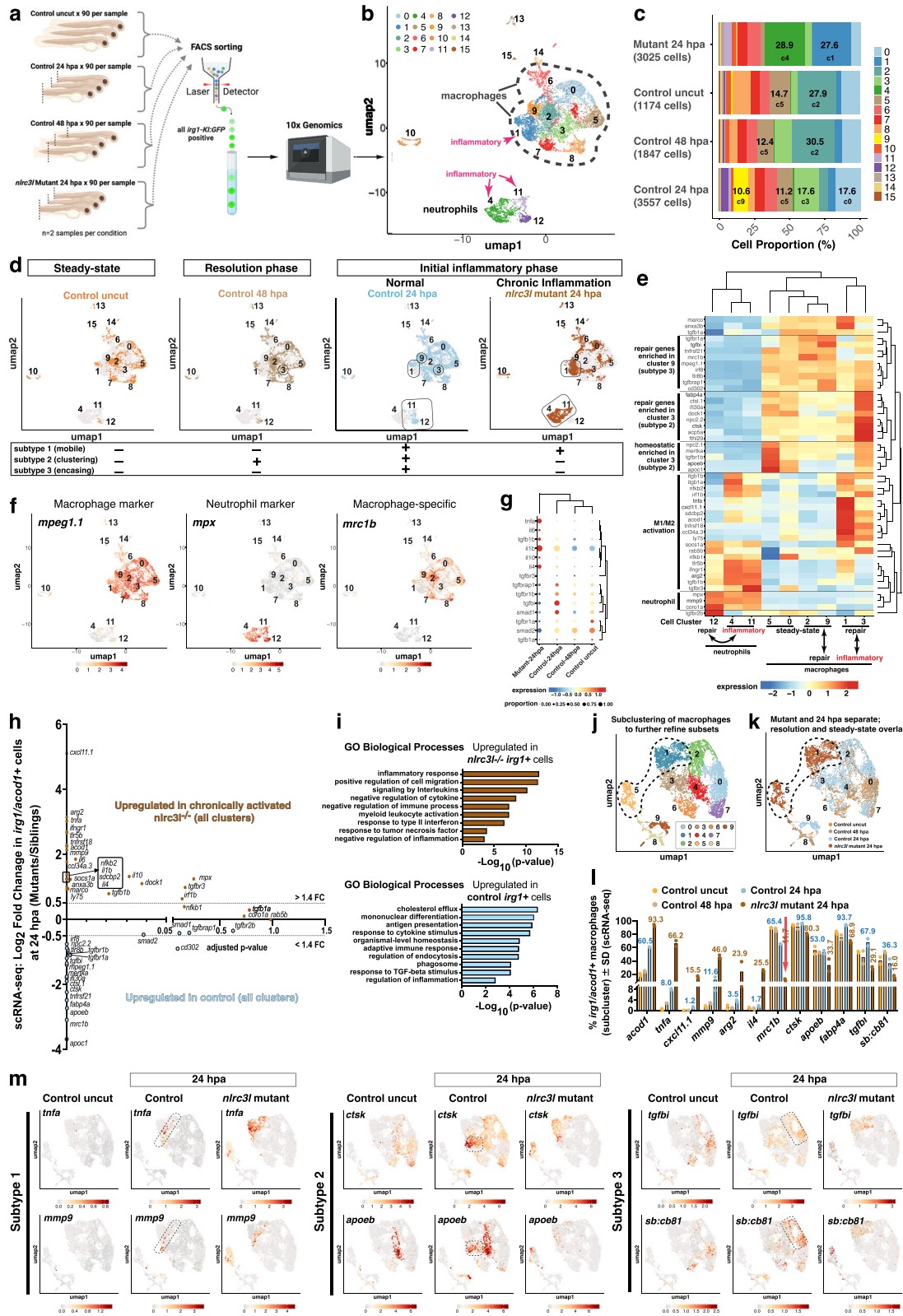

injury as shown by the qPCR analysis, suggesting sustained NF-kB activation that may compromise macrophage function in injury response compared to controls (Supplementary Fig. 15).

Furthermore, *mrc1b*, a gene encoding the mannose receptor (also known as CD206) of the C-type lectin receptor family crucial for phagocytosis, was shown earlier by RNA-seq analysis as significantly downregulated in baseline mutant macrophages (Fig. 2g), but not its

paralog *mrc1a* (Fig. 6a). To assess the normal post-injury expression of *mrc1b*, we performed qPCR analysis and HCR in situ hybridization, revealing strong expression in injury-responsive macrophages clustered at the cut site, corresponding to subtypes 2 and 3 in control siblings, and in the ventral region appearing in the caudal hemato-poietic tissue (CHT)(Fig. 6b–e). qPCR and HCR validation confirmed downregulation of *mrc1b* in *nlrc3l* mutant macrophages at both

**Fig. 7 | Single-cell profiling of *irg1*+ cells reveals that chronic inflammatory conditioning amplified macrophage divergence post-injury, causing subtype loss, restricted repair programs, *mrc1b* downregulation, and altered neutrophil states in *nlrc3l* mutants. a** Schematic of *irg1*+ cell isolation for single-cell RNA-seq (scRNA-seq). Created in BioRender. Shiau, C. (2025) https://BioRender. com/8472ma7. **b** UMAP using 30 principal components and SNN-graph-based clustering at resolution 0.4 shows majority of cells in the macrophage-identified region (as demarcated) based on macrophage markers (e.g., *mpeg1*, *csf1ra*). A separate smaller neutrophil population is also detected. **c** Stacked bar plots show cluster composition by condition (percentage of cells). Total cell counts sequenced per condition are shown in parenthesis. Major enriched clusters are annotated as a % of cells and a "c#". **d** UMAPs split by condition show cell distribution across the landscape. Outlined cluster numbers mark condition-enriched subsets distinct from uncut baseline cells. Neutrophils are almost exclusively associated with injury. **e** Heatmap of DEGs in 24 hpa cells across clusters provide cell type and state of each cluster. Scaled expression is shown. **f** UMAPs show macrophage and neutrophil markers used for cell-type annotation. *mrc1b* expression is restricted to the

macrophage domain. **g** Bubble plot of key M1/M2 cytokines and TGF-β pathway genes across conditions. Mutants show notable downregulation of several TGF-β pathway components. **h** Scatter plot of $\log_2$ fold changes for DEGs shown in (**e**). **i** Bar plots of enriched biological pathways (Metascape) based on DEGs from (**h**) with adjusted $p < 0.1$. **j–k** UMAPs of subclustered macrophage-domain cells (from panel **b**). (**j**) Reclustering macrophage-specific cells reveals subclusters (0–9). (**k**) Condition-specific distribution shows *nlrc3l* mutant macrophages confined to subclusters 1 and 5 (outlined). **l** Bar plots from macrophage subclustering show the percentage of *irg1*+/*acod1*+ macrophages expressing selected M1/M2 markers and reparative genes (*mrc1b*, *ctsk*, *apoeb*, *fabp4a*, *tgfbi*, *sb:cb81*). *mrc1b* is the most downregulated in mutant macrophages, red arrow. Numbers on bars indicate cell percentage expressing each gene. **m** UMAPs from macrophage subclustering of key genes define major macrophage subsets (demarcated by dotted lines in control 24 hpa cells) during the inflammatory phase. Subset 1 is mutant-cell-enriched; subset 2 is control 24/48 hpa cell-enriched; and subset 3 is control 24 hpa-specific marked by *tgfbi* and *sb:cb81*. See also Supplementary Fig. 19 for additional data.

baseline and post-injury (Fig. 6b–e). This aligns with the loss of functional subtypes 2 and 3, linked to phagocytosis and muscle enclosure, respectively (Supplementary Fig. 13).

The excessive upregulation of *mmp9* and *relb*, combined with a severe loss of *mrc1b/cd206* in *nlrc3l* mutant macrophages at baseline and post-injury, elucidates persistent molecular changes that likely impede critical macrophage functions for facilitating tissue repair processes. In support of this, knockdown of *mrc1b* using transient CRISPR gRNA injections with Cas9 was validated using T7 assay and sequencing (Supplementary Fig. 16); this resulted in similar reduction in cluster and muscle-encasing subtypes (2 and 3) (Supplementary Fig. 16) and repair defect as seen in chronically activated *nlrc3l* mutants, suggesting an essential role for *mrc1b* in muscle repair (Fig. 6f). The muscle repair defect was also confirmed in the stable *mrc1b*^sa18640^ mutant zebrafish (Fig. 6g and Supplementary Fig. 16). These mechanistic alterations may be consequences of the chronic inflammatory activation state of *nlrc3l* mutant macrophages that hinders their ability to perform essential tissue repair functions.

Furthermore, *myd88* knockout in *nlrc3l* mutants using double *nlrc3l/myd88* mutants led to recovery of *mrc1b* expression in macrophages post-injury (Fig. 6h, i), thereby restoring a more homeostatic phenotype. These results are consistent with *myd88*-dependent signaling contributing to dysregulated macrophage state and significant macrophage *mrc1b* downregulation in *nlrc3l* mutants. To further explore the generalizability to additional paradigms of chronic macrophage activation, we developed a model based on persistent systemic *E. coli* infection (Supplementary Fig. 17a–c). We showed that similarly to *nlrc3l* mutants, the chronic macrophage activation caused by infection impaired normal skeletal muscle repair (Supplementary Fig. 17d, e), and was associated with a *mrc1b* downregulation and reduction of both clustering and encasing macrophage subtypes (Supplementary Fig. 17f–k).

## Chronic macrophage activation erodes cellular heterogeneity, driving a universal pro-inflammatory program while still expressing repair-promoting genes

To better define the molecular impact of chronic macrophage activation and distinguish macrophage subtypes during muscle injury repair, we performed single-cell RNA-seq (scRNA-seq) on ~ 10,000 *irg1*+ cells from zebrafish at baseline (uncut), inflammatory phase (24 hpa), and resolution phase (48 hpa), as well as from *nlrc3l* mutants at 24 hpa (Fig. 7a and Supplementary Fig. 18). Seurat clustering of quality-filtered cells ($n = 9603$ comprised of 1174 cells (uncut), 3557 cells (24 hpa), 1847 cells (48 hpa), and 3025 *nlrc3l* mutant cells (24 hpa)) revealed 16 clusters spanning macrophages and neutrophils, with the vast majority of cells expressing canonical macrophage markers (*mpeg1.1*, *csf1ra*,

*mpeg1:BFP*)(Fig. 7b–f, Supplementary Fig. 18). Comparing cell distributions across conditions, baseline and resolution (uncut and 48 hpa) cells showed substantial overlap in UMAP space, indicating similar cell states (Fig. 7d). In contrast, cells from 24 hpa (control and *nlrc3l* mutant) were more divergent−both from other conditions and from each other−with *nlrc3l* mutant cells preferentially occupying distinct but confined clusters (notably 1, 4, and 11)(Fig. 7d and Supplementary Fig. 18, 19). Neutrophils were identified as clusters 4, 11, and 12, marked by *mpx* and *lyz* (Fig. 7f and Supplementary Fig. 20), confirming prior observations of *irg1* expression in some neutrophils only after injury especially in *nlrc3l* mutants (Fig. 2i and Supplementary Fig. 11). The remaining clusters (10, 13–15) were sparse and uncharacterized. The skewed clustering pattern in mutant cells, compared to the broader distribution of control cells, suggests a loss of macrophage heterogeneity under chronic activation (Fig. 7d and Supplementary Fig. 18).

Using prior knowledge from in vivo imaging of macrophages and select markers, we identified Seurat clusters corresponding to previously characterized macrophage subtypes: mobile (subtype 1), clustering (subtype 2), and muscle-encasing (subtype 3)(Figs. 4e–g and 7d and Supplementary Fig. 19). Subtype 1 cells, known to express high levels of *irg1* and *tnfa*, were enriched in cluster 1−predominantly composed of *nlrc3l* mutant and 24 hpa control cells (Fig. 7d and Supplementary Fig. 18d, e). Subtype 2, which appears only in 24 and 48 hpa control macrophages with strong *irg1* expression, was identified in cluster 3 (Fig. 7d and Supplementary Fig. 18d, e). Subtype 3, found exclusively in 24 hpa controls with muscle-encasing macrophages, was mapped to cluster 9, distinguished by high *mrc1b* expression and predominance of 24 hpa control cells (Fig. 7d and Supplementary Fig. 18d, e). To molecularly profile these subtypes, we performed pseudobulk analysis comparing 24 hpa *nlrc3l* mutant and control macrophages. Clusters of interest (macrophage clusters 0, 1, 3, 4, 5, 9; neutrophil clusters 4, 11, 12) revealed key transcriptional signatures (Fig. 7e–i). Cluster 3 (subtype 2) showed enrichment for repair and homeostatic genes, particularly those involved in lipid metabolism (*fabp4a*, *apoeb*, *apoc1*) and phagocytosis/ECM remodeling (*mertka*, *npc2.1*, *npc2.2*, *ctsk*, *ctsl.1*, *ifi30a*) with moderate levels of pro-inflammatory genes (including *tnfa* and *nfkb2*). Cluster 9 (subtype 3) was characterized by upregulation of TGF-beta signaling (*tgfbr1a*, *tgfbi*, *tgfbrap1*) and endocytic pathways (*cd302*)(Fig. 7e). Overall, chronically activated *nlrc3l* mutant cells upregulated pro-inflammatory genes and negative regulators of inflammation− including TNF-alpha, interleukins, cytokines, type II interferon and NF-kB components (*tnfa*, *ifngr1*, *il6*, *tgfb1b*, *il10ra*, *il4*, *nfkb2*, *traf2b*, *nik/map3k14a*), while downregulating TGF-beta pathway genes that were normally upregulated in 24 hpa control macrophages (Fig. 7g–i). This shift points to a dysregulated and uniformly pro-inflammatory state in chronically

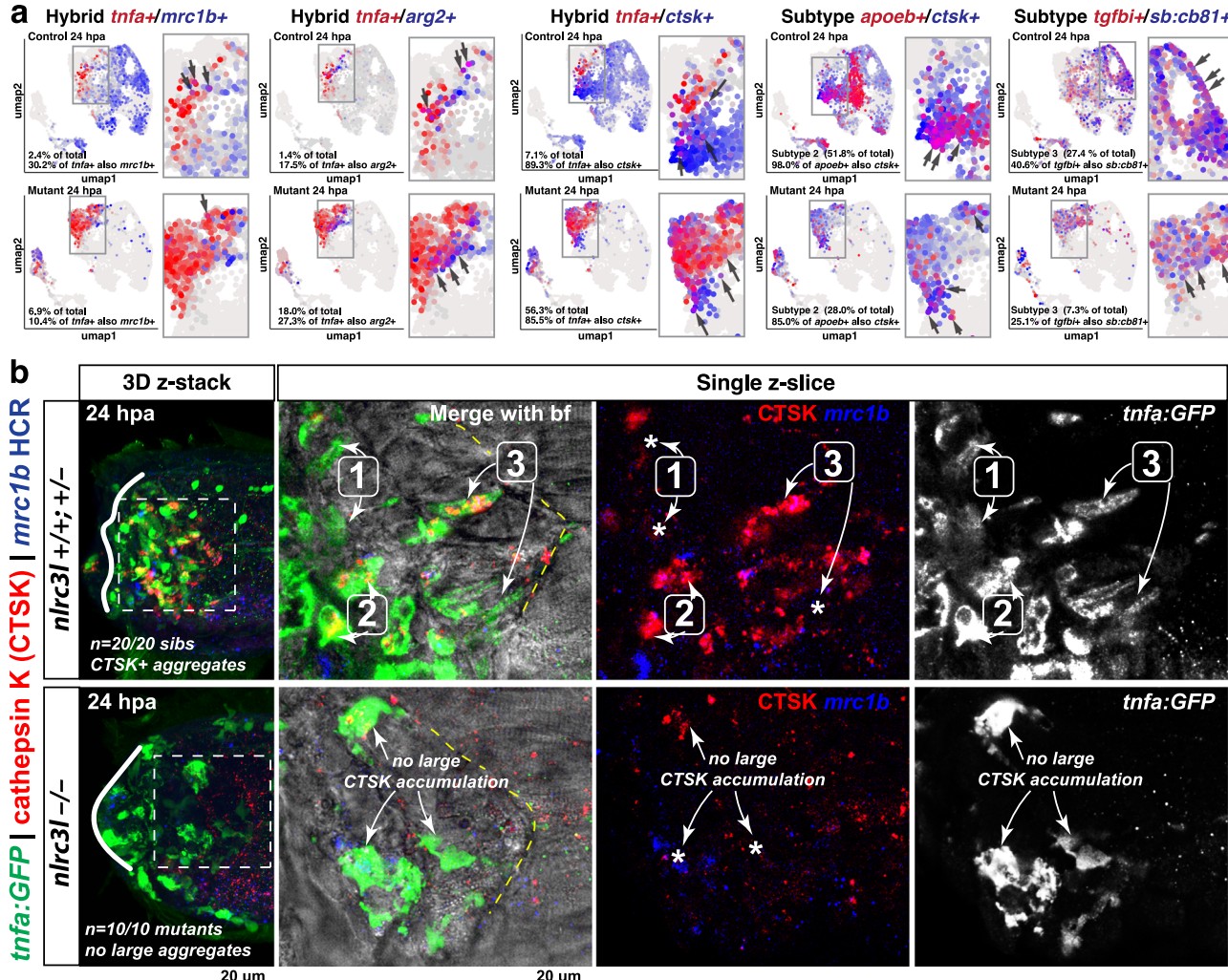

**Fig. 8 | Chronic activation of macrophages in *nlrc3l* mutants restricts macrophage heterogeneity, favoring hybrid M1/M2 states, and impairs subtype specification and cathepsin K accumulation. a** UMAP projections of scRNA-seq data show co-expression of selected M1/M2 or subtype gene pairs from subclustered macrophage identity populations (see Supplementary Fig. 19). Shades of purple indicate cells co-expressing both genes based on red/blue intensity ratios. Boxed region in the UMAP highlights hybrid M1/M2 populations or subtype 2/3 cells at 24 hpa across genotypes. Side panels show magnified views of the boxed areas. **b** Confocal imaging of macrophages in *nlrc3l* mutant and control siblings at 24 hpa showing co-localization of *tnfa* reporter (GFP), CTSK protein, and *mrc1b* mRNA using HCR labeling. The first column presents 3D volumetric views of

merged fluorescence channels; adjacent panels show a representative single z-slice from the dotted box region, followed by individual channels: CTSK/*mrc1b* merged and GFP (*tnfa*) alone. Brightfield (bf) overlay delineates tissue morphology. Yellow dotted lines demarcate injured (left) versus uninjured (right) muscle. Numbered arrows identify representative macrophage subtypes (1, 2, 3). In *nlrc3l* mutants, macrophages consistently lack prominent, densely packed CTSK accumulations that typically occupy much of the cell volume in a subset of control macrophages of the reparative subtypes 2 and 3. Asterisks denote macrophages with absent accumulation of CTSK protein. *n* indicates number of animals analyzed across two independent experiments.

activated mutant macrophages, consistent with and extending insights from bulk RNA-seq results (Fig. 2g).

To isolate macrophage-specific changes, we further subclustered the dataset using cells in clusters 0-3 and 5-9 (Fig. 7b), excluding neutrophils (Fig. 7j–m, Supplementary Fig. 19). UMAP projections of macrophage-only data revealed that *nlrc3l* mutant macrophages were concentrated in specific subclusters (notably 1 and 5), while control macrophages at various timepoints were broadly distributed (Fig. 7j, k). Baseline (control uncut) and 48 hpa control macrophages again showed close overlap, reinforcing their shared transcriptional identity. Despite high levels of pro-inflammatory genes (*acod1, tnfa, cxcl11.1, mmp9*), chronically activated mutant macrophages still express many repair-associated genes (*mrc1b, ctsk, apoeb, fabp4a, tgfbi, sb:cb81* also known as ZFP36L1-like) and alternative activation markers (*arg2, il4*), albeit at lower levels (Fig. 7l, m). Among these, *mrc1b* was the most strongly downregulated gene in mutant macrophages at 24 hpa (Fig. 7l

and Supplementary Fig. 19g). Although thought to support phagocytosis, *mrc1b* was unexpectedly most highly expressed in homeostatic and resolving macrophages (Fig. 7l, Supplementary Fig. 19g), rather than during the peak phagocytic period in the initial inflammatory phase of the injury response. Its strong suppression in chronically activated macrophages suggests that macrophage activation may negatively regulate *mrc1b* transcription and that its functional role in macrophage biology is more complex and not fully understood.

Moreover, a substantial proportion of mutant macrophages (ranging from ~10% to over 50%) co-expressed classical M1 and M2 markers, a phenomenon we termed hybrid M1/M2 states (Fig. 8a, Supplementary Fig. 19). These hybrid cells were less than 10% of the control 24 hpa macrophages, indicating that while hybrid states occur under normal conditions, chronic activation significantly enhances this mixed inflammatory/repair phenotype (Fig. 8a, Supplementary Fig. 19). Although gene markers corresponding to subtypes 2 (based

on cluster 3 markers, *apoeb + /ctsk + )* and 3 (based on cluster 9 markers *tgfbi + /sb:cb81 + )* were present in *nlrc3l* mutant macrophages, they were significantly underrepresented and localized to regions with high expression of pro-inflammatory activation genes like *irg1* and *tnfa* (Figs. 7l, m and 8a). This further supports the conclusion that chronic macrophage activation skews the transcriptional landscape toward a uniform, hybrid inflammatory state while eroding the normal cellular diversity required for balanced muscle repair.

Although some *irg1*⁺ cells post-injury were identified as neutrophils (clusters 4,11, and 12)(Fig. 7f), especially in *nlrc3l* mutants, their *EGFP* transcript counts were tens to hundreds times lower than in macrophages (Supplementary Fig. 19h), consistent with low reporter levels seen by in vivo imaging (Fig. 2i and Supplementary Fig. 11). Subclustering of neutrophils from scRNA-seq showed that most of these cells were from 24 hpa *nlrc3l* mutants and exhibited altered states, expressing both M1 and M2 markers (e.g., *acod1, tnfa, mmp9, arg2*)(Supplementary Fig. 20), consistent with bulk RNA-seq results showing phenotypic shift resembling activated macrophages (Fig. 2g and Supplementary Fig. 15). Overall, *irg1-KI:GFP* reliably tracks macrophages at baseline, when neutrophils do not express the reporter, and during activation, when macrophages express the reporter at a much higher intensity and abundance, clearly distinguishing them from neutrophils under perturbation.

Our scRNA-seq analysis identified an endolysosomal protease cathepsin K (*ctsk*)[53] as a key marker highly upregulated in clustering subtype 2 macrophages during the initial inflammatory phase of a muscle injury response at 24 hpa (Fig. 7m). To validate this, we conducted in vivo experiments showing that cathepsin K (CTSK) protein was not only abundant in clustering macrophages, as predicted, but enriched in muscle-encasing macrophages, forming large dense cytosolic accumulations (Fig. 8b), suggesting intracellular collagen and protein degradation[53,54] as a critical function shared by these subtypes at 24 hpa. Strikingly, these densely packed CTSK⁺ domains were entirely absent in chronically activated *nlrc3l* mutant macrophages post-injury (Fig. 8b), indicating a profound loss of a hallmark protease. During injury response, CTSK⁺ cells co-expressed *tnfa*, consistent with a hybrid reparative-inflammatory state in wild-type and control macrophages as suggested by the scRNA-seq data (Fig. 8a and Supplementary Figs. 18, 19). While mutant macrophages co-expressed *tnfa* and *ctsk* at the transcript level, *ctsk* expression was markedly reduced transcriptionally, and CTSK protein was detectably low (Fig. 8b and Supplementary Figs. 18, 19).

Overall, single-cell transcriptomic analysis revealed that chronic macrophage activation due to *nlrc3l* deletion reduced cellular heterogeneity, reinforced a dominant pro-inflammatory transcriptional program, and promoted hybrid M1/M2 states co-expressing inflammatory and repair-associated markers. These hybrid states, less frequent in control macrophages (< 10% at 24 hpa), were significantly enriched in mutants. At least three macrophage subpopulations were identified, including the pro-inflammatory subtype 1 marked by high *irg1* and *tnfa*, and two pro-repair subtypes: subtype 2, characterized by cholesterol efflux (*fabp4a, apoeb, apoc1*) and phagocytosis-related genes (*mertka, npc2.2, ctsk, ctsl.1, ifi30a*), and subtype 3, marked by TGF-β signaling (*tgfbr1a, tgfbr1b, tgfbi*). Both reparative subtypes were significantly depleted in mutants. Thus, chronic macrophage activation impairs muscle injury repair not by enhancing or diversifying inflammatory functions, but by disrupting reparative programs. Furthermore, pro-inflammatory gene expression underlies the transcriptome of these macrophages, constraining their plasticity despite showing expression of pro-repair pathways, albeit at a reduced level. Together, these results provide new mechanistic insights into how chronic inflammation disrupts macrophage-mediated repair by downregulating *mrc1b/cd206*, blocking cathepsin-K-rich degradative compartments, and promoting functionally deficient hybrid states,

and also establish cathepsin K as a validated marker of reparative macrophage identity in zebrafish.

## Discussion

Acute inflammation is essential for repairing skeletal muscle damage from trauma, infections, or toxins[1,2]; however, the duration and intensity of the inflammatory response after the injury significantly affect the outcome. Chronic inflammation, characterized by a prolonged response, is associated with impaired muscle regeneration in mammals that can lead to fibrosis, myofiber degeneration, and persistent inflammation[1,2,6]. Although the immune system, particularly macrophages, largely determines the nature of the inflammatory response to injury, the mechanisms by which macrophages regulate that process remain unclear.

In this study, we show that a deleterious *nlrc3l* mutation in zebrafish leads to chronic inflammatory activation of macrophages throughout the injury time course, marked by persistently high expression of a metabolic marker of macrophage activation *irg1* and the pro-inflammatory cytokine *tnfa* in addition to a significantly upregulated *mmp9* expression, a known secreted inflammatory proteinase upregulated in various chronic inflammatory diseases[49,50,55]. The significant upregulation of *mmp9* in *nlrc3l* mutant macrophages at baseline and after major tail cut injury (Supplementary Figs. 15 and 19) may reflect its role in sustaining the inflammatory environment, as it has been proposed to activate and regulate levels of cytokines, adhesion molecules, and growth factors, possibly activating TNF-alpha and IL-1B proteins[49–51]. This contrasts sharply with the significant reduction or elimination of immune activation marked by high *irg1* expression after the acute inflammatory phase by 72 hpa in wild-type macrophages (Fig. 4b). Using an endogenous reporter of macrophage activation, based on a GFP knock-in allele at the *irg1* locus, we demonstrate that chronically activated macrophages impair muscle repair following tail amputation. Restoring normal macrophage function, either by reintroducing wild-type *nlrc3l* or inhibiting *myd88*-dependent inflammatory signaling, rescues muscle repair defects. These findings highlight the importance of properly controlled acute inflammation during injury response to prevent collateral and long-term muscle damage. This requirement for a balanced inflammatory response appears to be evolutionarily conserved, as we show in this study that prolonged excessive inflammation hinders skeletal muscle injury repair in zebrafish, mirroring the findings in mammals from both in vitro and in vivo methods[42,56,57].

The impact of prolonged inflammatory activation on macrophage heterogeneity and function during muscle repair remains poorly defined. Previous studies in rodents and human cells have shown that pro-inflammatory M1 macrophages promote satellite cell activation and myoblast proliferation, but inhibit differentiation, while pro-repair or anti-inflammatory M2 macrophages facilitate myoblast differentiation[7,8]. Consistent with macrophages having an essential role in muscle repair, we show that depletion of macrophages in zebrafish, irrespective of state, impairs muscle repair. Similarly, depletion of mammalian monocytes/macrophages, such as by *Ccr2* deletion, irradiation, diphtheria toxin, clodronate, or pharmacological inhibitors of migration, also impede normal muscle repair[8,58–60], highlighting a conserved and essential role for macrophages in muscle injury repair. While pro-inflammatory macrophages have been shown to promote myoblast expansion in vitro[8,61], our data suggest that chronically activated macrophages inhibit satellite cell activation (Fig. 5a), thereby preventing enough myoblasts to form and facilitate an effective muscle regeneration. Chronic activation may cause a significant imbalance in macrophage states, skewing heavily toward a chronically pro-inflammatory profile, but yet not functionally equivalent to the classic pro-inflammatory state. Since inflammatory myopathies can involve persistent M1 macrophages and elevated pro-inflammatory cytokines[62,63], our *nlrc3l* mutant provides a useful paradigm for

exploring relevant mechanisms and potential therapeutic strategies. Results from this study raise the possibility that non-functional macrophages may be a driver of these disorders.

Through dynamic in vivo imaging of normal macrophage behavior following acute muscle injury, we observed that during the peak inflammatory phase (around timepoint 24 hpa), most macrophages at the injury site expressed high levels of M1 markers such as *irg1* and *tnfa*. Interestingly, a subset of these macrophages concurrently expressed *mrc1b/cd206*, a marker typically associated with M2 macrophages[64], and this marker was most enriched in macrophages located in the muscle region where muscle cells are phagocytosed or encased by macrophages to facilitate muscle regeneration (Fig. 6a–d). This suggests that injury-responding macrophages initially categorized as pro-inflammatory based on markers expression can simultaneously adopt pro-repair functions, such as subtype 2 (phagocytosis) and subtype 3 (muscle-encasing), as shown in Fig. 8 and Supplementary Fig. 19. We found that macrophages display substantial functional plasticity, showing a dynamic mix of M1- and M2-like behaviors early on during the initial inflammatory phase of the injury response. They express markers and perform functions characteristic of both pro-inflammatory (M1-like) and pro-repair (M2-like) states, suggesting a broader overlap of activation states and challenging the conventional view of distinct macrophage states (Figs. 7 and 8). Single-cell analysis of normal control and chronically activated *nlrc3l* mutant macrophages indicate a small percentage (<10%) of normal macrophages during initial inflammatory phase of the injury response co-express alternative activation or pro-repair genes (such as *mrc1b* and *arg2*) with pro-inflammatory M1 marker *tnfa*, while this is significantly increased to as much as 50% in chronically activated *nlrc3l* mutant macrophages (Fig. 8a and Supplementary Fig. 19). Notably, the two functional subtypes 2 and 3 were largely absent in auto-inflammatory *nlrc3l* mutants based on multiple modes of analyses (in vivo imaging (Fig. 4 and Supplementary Fig. 13), bulk and single-cell RNAseq (Figs. 2, 7, and Supplementary Figs. 18, 19), where macrophages appear locked in a persistent M1-like pro-inflammatory state, losing the ability to concurrently adopt M2-like pro-repair functions although expressing low levels of reparative program genes.

An important question is how the functional macrophage subtypes 2 and 3, categorized as "clustering" and "muscle-encasing", respectively, can be distinguished at the molecular level and by their specific roles during injury. Addressing this will be crucial for elucidating the mechanisms that enable macrophages to perform the diverse array of essential functions in muscle injury repair. Insights from our scRNA-seq analysis highlighted distinct pathways elevated in the two pro-repair subtypes, whereby cholesterol efflux (*fabp4a*, *apoeb*, *apoc1*) and phagocytosis-related genes (*mertka*, *npc2.2*, *ctsk*, *ctsl.1*, *ifi30a*) primarily marked the clustering subtype 2 macrophages, and response to TGF-β signaling (*tgfbr1a*, *tgfbr1b*, *tgfbi*) is most highly induced in muscle-encasing subtype 3 macrophages (Fig. 7m and Supplementary Fig. 19). In vivo analysis of the subtype 2 marker *ctsk*, encoding the protease cathepsin K (CTSK), revealed a marked difference between control and *nlrc3l* mutant macrophages. In controls, a subset of both clustering and muscle-encasing macrophages showed high intracellular accumulation of the CTSK protein during the inflammatory phase of injury (Fig. 8b), a feature absent in chronically activated *nlrc3l* mutants. This CTSK accumulation, consistent across all wild-type and control animals analyzed, marks normal reparative macrophage activation and suggests an active macrophage role in collagen and matrix degradation. *ctsk* deficiency is known to impair breakdown of cholesterol and extracellular matrix, leading to foam-like macrophages in atherosclerosis[65], which the vesicle-engorged *nlrc3l* mutant macrophages may phenotypically mimic given their shared *ctsk* loss. While *ctsk* is best known as a cysteine protease involved in collagen degradation during osteoclast-mediated bone resorption, it is also upregulated under pathological and inflammatory

conditions[53,66], including in epithelioid cells and multinucleated giant cells of macrophage origin[67], tumor-associated M2 macrophages[68], and human atherosclerotic plaques[65,69]. Here, by contrast, we demonstrate that upregulation and accumulation of cathepsin K is part of a normal process in macrophage activation during tissue injury response, one that fails to occur in chronically activated *nlrc3l* mutants. This loss of functional plasticity mirrors the impaired muscle repair seen with macrophage depletion.

Our zebrafish model of acute muscle injury, induced by major tail amputation at 2 days post-fertilization, enables the targeted study of recruited macrophages and their interactions prior to the establishment of muscle-resident macrophages, offering relevant insights into mammalian inflammatory monocyte mechanisms[1,6,70]. Although neutrophils are generally first to be recruited, macrophages rapidly appear at the cut site by 5 hpa, becoming the primary source of *tnfa* expression and necrotic cell clearance (Fig. 4). This suggests zebrafish macrophages implement functions typically attributed to mammalian neutrophils and monocytes recruited to the injury site. In *nlrc3l* mutants, neutrophils exhibit unexpected changes, including increased cell numbers and expression of the transcriptional indicators of inflammation (*irg1*, *tnfa*) typically confined to macrophages in zebrafish[17,39,40], indicating possible compensatory mechanisms when macrophages are dysfunctional. Interestingly, a previous study has described muscle lesions shielded by macrophages to protect them from further damage due to inflammatory neutrophils[71]. This phenomenon may resemble the muscle-encasing macrophages (subtype 3) we observed in zebrafish, although at a different spatial scale. In zebrafish post-injury, individual macrophages are seen wrapping around single myocytes rather than a cluster of macrophages enveloping an area of injured myocytes. The role and significance of neutrophil-macrophage interplay during the various phases of muscle injury response and resolution remain areas of ongoing investigation.

The significant loss of the mannose receptor *mrc1b* (an ortholog of human *Mrc1*) in chronically activated macrophages of *nlrc3l* mutants aligns with the observed loss of phagocytic and engulfment capabilities in these cells post-injury, offering a possible molecular explanation for the unresolved muscle injury. Furthermore, our study shows that chronic activation disrupts macrophage subtype dynamics, leading to the loss of *mrc1b*-expressing reparative macrophage subtypes during injury response in both genetic and persistent infection models, highlighting a broad link between chronic macrophage activation and inappropriate Mrc1 downregulation, with important implications for understanding the mechanism impeding inflammatory macrophages from supporting tissue repair. Consistent with this, a prior scRNA-seq study showed that *Toxoplasma gondii* infection-induced chronic inflammation impeded muscle repair following cardiotoxin injury in mice, which coincided with a loss of *Mrc1*+ macrophages, a subset typically enriched post-injury and suggested to facilitate repair[42]. The scRNA-seq analysis also identified macrophages co-expressing pro-repair *Mrc1* and pro-inflammatory markers early in injury[42], but their functions remain unclear, and validation of the co-expression of markers was lacking in vivo. Together, these findings indicate a possible conserved mechanism in fish and mammals by which chronic inflammation downregulates the mannose *Mrc1* receptor expression in macrophages and thereby causes impaired muscle repair.

Mrc1 (also known as CD206) is a transmembrane glycoprotein belonging to the C-type lectin family, primarily expressed by tissue-resident macrophages, dendritic cells, and some lymphatic and endothelial cells[64,72]. It is also upregulated in tumor-associated macrophages and during inflammatory responses[64]. Mrc1 functions as a scavenger receptor on macrophages, especially within M2-like pro-repair subsets, binding to endogenous molecules (including lysosomal hydrolases and collagens), and microbial components[64,72]. In zebrafish, *mrc1a* and mrc*1b* are orthologs of human *Mrc1*, with mrc*1a* mainly

expressed in lymphatic endothelial cells[73,74] and weakly in some microglia[75,76], but mrc1b is much less characterized. We show that *mrc1b* is highly expressed in macrophages clustered at the muscle injury site corresponding to the clustering (likely phagocytic) and muscle-encasing macrophage subtypes 2 and 3, and its knockdown impairs muscle repair similarly to macrophage depletion (in *irf8* mutants) or chronic activation (in *nlrc3l* mutants), indicating its essential role in regeneration. Our findings suggest possible sub-functionalization between the two paralogs, with the role of mammalian Mrc1 in macrophages predominantly fulfilled by *mrc1b* in zebrafish. This interpretation is also supported by previous scRNA-seq data, which identified specifically *mrc1b* expression in a subset of tissue-resident macrophages in adult zebrafish[28].

While previous studies have associated Mrc1/CD206 positive macrophages with wound repair[77,78], our single-cell transcriptomic analysis from isolating macrophages at different timepoints of the injury response suggests that *mrc1b* dynamically marks macrophage functional states, and its unexpected high expression at baseline rather than during peak phagocytic activity of the injury response provides evidence that its in vivo function is still incompletely understood (Fig. 7l, Supplementary Fig. 19g). Therefore, our findings reveal a role of *mrc1b* in distinguishing macrophage subtypes during injury response and tissue repair in zebrafish, where it may mediate macrophage functions beyond clearance and repair of injured tissue, such as differentiation of reparative and inflammatory macrophage subtypes (schematized in Fig. 6j). While the exact molecular mechanism regulating *mrc1b* remains unclear, our findings provide a foundation for future studies into how chronic inflammatory activation alters *mrc1b* expression in zebrafish. Moreover, depletion of MyD88 signaling partially restores *mrc1b* levels in *nlrc3l* mutants post-injury, suggesting that target genes of the NF-kB pathway downstream of MyD88 may repress *mrc1b* transcription directly or indirectly in zebrafish.

Integrating a zebrafish GFP knock-in *irg1* reporter for tracking activated macrophages with acute muscle injury analysis in the *nlrc3l* mutant background offers a unique genetic paradigm amenable to in vivo single-cell and whole-body imaging for interrogating how chronic inflammation affects tissue injury and repair. While existing reporters like *tnfa:GFP*[32] label macrophage activation, they reflect distinct pathways. *tnfa*, encoding a pro-inflammatory cytokine, marks inflammatory signaling, while *irg1*, a mitochondrial enzyme, tracks metabolic reprogramming. Both show partial overlap during peak inflammation ( ~ 24 hpa; Fig. 7l–m, Supplementary Figs. 18 and 19), but *irg1* expression persists into resolution, indicating broader, sustained activation. Consistent with this, our scRNA-seq data reveal that *irg1* is more broadly expressed post-injury than *tnfa* (Fig. 7, Supplementary Figs. 18 and 19). *irg1-KI:GFP*, generated via targeted knock-in, ensures consistent, quantifiable expression across animals. In contrast, the expression of *tnfa:GFP*, inserted via transposon, varies with copy number and integration site. Moreover, *tnfa:GFP* is expressed in non-immune cells, including CNS and PNS neurons[79] (Supplementary Figs. 4, 9), complicating immune-specific analysis, such as by FACS and transcriptomics. *irg1-KI:GFP*, by contrast, is predominantly macrophage-specific, where its upregulation marks a shift in activation state without detectable expression in non-immune cells. Thus, *irg1-KI:GFP* offers a stable, specific, and quantitative in vivo tool for tracking immune activation and metabolic state unmatched by existing reporters.

We show that chronic inflammation diminishes macrophage functional plasticity, inhibiting their ability to clear damaged tissue and promote regeneration (Fig. 6j). These findings highlight the importance of maintaining macrophage plasticity for effective tissue repair that is associated with a normal level of Mrc1. While Mrc1 has been a potential therapeutic target for delivering small molecules and mannosylated cargo into macrophages to treat cancer, infectious diseases, and other conditions[64,80], this study implicates that

reconstituting Mrc1 in macrophages may be equally useful for chronic inflammatory conditions associated with deficient or non-functional macrophages. Our study demonstrates that chronic activation restricts macrophage cell heterogeneity and disrupts subtype dynamics, leading to loss of *mrc1b*-expressing clustering and encasing macrophage subtypes during injury responses in both genetic and infection models of chronic inflammation. It also results in the loss of high-density intracellular cathepsin K accumulation typically seen in reparative macrophages, suggesting deficiencies in intracellular protein degradation. These findings reveal that chronic activation suppresses *mrc1b* expression in a *myd88*-dependent manner and inhibits cathepsin K from accumulating at a high concentration, offering new insights into how chronic inflammation reprograms macrophages to impair their ability to support tissue repair.

## Methods

### Zebrafish transgenic and mutant lines
Embryos from wild-type, mutant, and transgenic backgrounds were derived from: *nlrc3*[st73][23], *irf8*[st95][48], *myd88*[b1358][81], *irg1/acod1*[bcz13] (this study), *asc*[bcz82/nc303cs] (this study), *mrc1b*[sa18640] (this study), *irg1-KI:GFP* (knock-in, this study), *irg1:GFP* (tol2-based, this study), *tnfa:GFP*[pd1028] [32], "macro-rescue" *mpeg1:nlrc3l*[24], *lyz:mCherry*[82], *mpeg1:GFP*[61], and *mpeg1:B*FP[82] and raised at 28.5°C to stage for analysis. Genotyping primers and assays are described in Supplementary Data 1. This study was carried out in accordance with the approval of UNC-Chapel Hill Institutional Animal Care and Use Committee (protocols 19-132 and 22-103).

### Generating GFP knock-in at the *irg1/acod1* locus using GeneWeld platform
GFP knock-in followed protocol as previously described[25,26]. High quality genomic DNA was used to identify the 48-bp upstream and downstream homologous arm sequences for cloning using Type IIS restriction enzymes BfuAI and BspQI for ligation into the GeneWeld (GW) plasmid, pGTag vector[25,26]. The following primers were used to verify the correct insertions: 5'_pgtag_seq: 5'-GCATGGATGTTTTCC-CAGTC-3' and 3'_pgtag_seq: 5'-ATGGCTCATAACACCCCTTG-3'. Single-cell injections into zebrafish were conducted using a mix containing the universal gRNA, *irg1* gRNA-2, Cas9 mRNA, and the *irg1*-targeting GW vector, and also without the *irg1*-targeting GW plasmid as a negative control. A total of 15 injected $F_0$ adults were outcrossed to generate stable $F_1$ founders. Progeny from only 1 out of 15 $F_0$ fish gave a GFP validation by PCR. Assessment of 49 adult $F_1$s yielded 2 fish (#7 and 31) carrying GFP that was localized to the *irg1* locus (yielding a 0.3% success rate) as verified by PCR and Sanger sequencing using multiple primer sets as well as by experimental evidence of a Mendelian segregation of the reporter expression. This new fish line is named "*irg1-KI:GFP*".

### Cloning and creating transgenic TOL2-based *irg1:GFP* reporter
Approximately 5 kb of regulatory sequence upstream of the *irg1/acod1* coding sequence for the start codon was cloned from the BAC DKEY-57A22, which contained the zebrafish chromosomal region encompassing the *irg1* region in chromosome 9, using the *irg1p* −4794 F and *irg1p* + 18 R primers as listed in Supplementary Data 1. This represents a longer fragment than that used in the previously published TOL2-based *irg1* reporter by Sander et al.[22]. The ~5 kb regulatory sequence of *irg1* was cloned into a Gateway-compatible p5E vector backbone to be combined with a pME-GFP coding sequence to create the *irg1:GFP* plasmid. Tol2-mediated transgenesis was used to integrate this reporter cassette into the zebrafish genome to create the new *irg1:GFP* stable transgenic line, as shown in Supplementary Fig. 2.

### Tail amputation
Zebrafish embryos at 2 dpf were anesthetized in 0.02% MS-222 (tricaine) supplemented with 0.003% PTU. Embryos were amputated at

the halfway point between the end of the yolk extension and tail fin end (~0.5 mm measured from tip of tail fin equivalent to ~8-10 somites from end of yolk extension) in a Sylgard lined petri dish using a 5 mm straight microsurgical blade with a 15 degree angle (Sharpoint Microsurgical Knives, 72-1551). After the operation, embryos were immediately transferred to fresh water with PTU and monitored for recovery. Most experiments focus on the initial inflammatory phase of the injury response, so embryos were typically collected 18-24 h after amputation for the 24 hpa timepoint.

### In vivo time-lapse and static confocal imaging and analysis

All time-lapse and static z-stack imaging were performed using a Nikon A1R+ hybrid galvano and resonant scanning confocal system equipped with an ultra-high speed A1-SHR scan head and controller. Images were obtained using an apochromat lambda 40x water immersion objective (NA 1.15) or a plan apochromat lambda 20x objective (NA 0.75). Z-steps at 1–3 μm were taken at 40x and 3–5 μm at 20x. Different stages of zebrafish were mounted on glass-bottom dishes using 1.5% low-melting agarose and submerged in fish water supplemented with 0.003% PTU to inhibit pigmentation. All image acquisition parameters were kept constant within an experiment (including laser power, gain, speed, and resolution). Image analysis was performed using ImageJ Fiji version 2.9.0 for injury analysis at the tissue or cell population level, or Imaris 10.1.1.software (Bitplane Oxford) for dynamic imaging or large-scale single cell quantifications (fluorescence, numbers). Traced ROIs were measured for area and fluorescence levels from original, unadjusted raw data. For tail cut images, "cut site" is defined as the region within the two intact somites from the edge of the injured tail. The largest ROI within one intact somite from the edge of the injured tail and injured tissue is defined as the "cut site cell cluster" for analysis. Imaris analysis was used to generate 3-dimensional surfaces of cells of interest using Machine Learning Pixel Classification with the Fiji/ImageJ Plugin Labkit, followed by manual curation to ensure accuracy. Quantification of fluorescence levels and other measurements were taken from surfaces created for the different cells using Imaris. For movie analyses, macrophage surfaces were generated using customized algorithms, where surfaces were tracked over time, and machine learning segmentation was used on the GFP channel for macrophages with repeated training. Slicer extended section was set at 3 μm, the size of each Z-slice, and any ROIs less than 10 voxels were filtered out. Tracking was conducted using the autoregressive motion algorithm with a max distance of 20 μm. Repeated manual editing was required to separate clumps and cells in close contact using the cut surface tool, or to connect pieces that belonged to one cell using the unify tool at every timepoint. Tracks of each cell ROI were then aligned which required manual checks and editing to disconnect or reconnect to assemble a correctly integrated track, which was repeated for each ROI at every timepoint.

### Fluorescent stereoscope imaging and analysis

Leica M165 FC microscope (Leica Microsystems, Germany) equipped with a Leica DFC9000 GT camera was used to acquire fluorescent images of live whole embryos. Embryos were mounted in petri dishes with 1% low-melting agarose and covered with water supplemented with 0.003% PTU, and imaged individually with consistent laser power and magnification settings within each experiment. Yolkball analysis was performed using ImageJ Fiji. Any visible fluorescent cells within the boundary of the yolkball were captured as an ROI. ROIs were created automatically by using the threshold tool to generate a mask of fluorescent cells and selecting the "Analyze Particles" option to create the ROI set. A freehand selection tool for manual tracing was used to generate additional ROIs of any missed cells from the automated threshold setting. ROIs were then used to measure area and fluorescence levels from original unadjusted images.

### Whole mount immunohistochemistry

Embryos were fixed in 4% paraformaldehyde (PFA) for 2 h at room temperature or overnight at 4°C, followed by a series of PBST washes (phosphate-buffered saline with 0.2% Tween20) and passage through 100% methanol at −20°C until proceeding with immunostaining. Pax7 antibody staining required a shorter fixation protocol using 2% PFA fix for 20 min at room temperature. Blocking solution was PBST with 5% normal goat serum, followed by overnight incubation with the primary antibody at 4°C, then washed before proceeding to an overnight incubation with the appropriate secondary antibody at 4°C. The following primary antibodies were used in blocking solution: rabbit anti-smooth muscle actin (GTX100034, GeneTex) at 1:500, chicken anti-GFP (ab13970, Abcam) at 1:500, mouse anti-Pax7 (AB_528428, DSHB) at 1:100, and rabbit anti-CTSK (cathepsin K) at 1:500 (E7U5N, Cell Signaling Technology) followed by incubation with the appropriate secondary antibodies used at 1:500 to 1:2000. Use of antibodies were validated based on pattern of staining in control normal samples that matched description of reagent provided by the manufacturer and previously published data. DAPI (Roche) may be used in the last wash at 5 μg/mL for an hour. See Supplementary Data 1 for details of reagents used.

### Phalloidin staining for muscle actin

Embryos were fixed in 4% paraformaldehyde (PFA) for 2 h at room temperature or overnight at 4°C, followed by several 10-min PBST washes and incubation with Phalloidin-iFluor 647 (Abcam) at 1:500 dilution in PBST overnight at 4°C. After staining, embryos were washed in PBST and mounted in 1% low-melting agarose for imaging.

### Whole mount hybridization chain reaction (HCR)

Embryos, either injured or normal, were fixed in freshly made 4% PFA overnight at 4°C and stored in 100% methanol. To proceed with HCR, embryos were rehydrated, permeabilized using proteinase K, and processed through a series of reagents as per protocols previously published and provided by the manufacturer (Molecular Instruments). Probe targets and amplifiers were designed and made by Molecular Instruments as listed in Supplementary Data 1.

### Isolation of macrophages for bulk transcriptome analysis by FACS

Embryos from *nlrc3*[st73] heterozygous incrosses were sorted as siblings or mutants based on a high induction of *irg1-KI:GFP* expression at 3 dpf. Around 20-50 embryos were dissociated per sample on ice in FACSmax Cell Dissociation Solution (Amsbio) in a microcentrifuge tube using a motorized pellet pestle (Fisherbrand) until the sample is completely homogenized. The cell suspension was filtered through a sterile 35 μm nylon mesh (Falcon, 352235) and another passage through a sterile 20 μm mesh strainer (EASYstrainer, 542120) with an additional 500 μL FACSmax solution run through. The dissociated cell suspension was then analyzed and sorted directly into RNA lysis buffer (Ambion) on a BD FACSMelody Cell Sorter. General gating was applied using forward and side scatter to distinguish populations for single live cells that represented macrophages expressing high and low GFP expression from the *irg1-KI:GFP* transgene. See gating strategy in Supplementary Fig. 21.

### RNA isolation and qPCR

RNA was isolated using the RNAqueous-Micro kit RNA Isolation Procedure (Ambion). Whole larvae were lysed in 100−300 uL RNA lysis buffer. cDNA was made from 150 or 200 ng of total RNA using oligo (dT) primer with SuperScript IV reverse transcriptase (Invitrogen). qPCR was performed on the QuantStudio 3 Real-Time PCR System (Applied Biosystems) using SYBR Green. The delta-delta CT method was used to determine the relative levels of mRNA expression between experimental samples and controls. *ef1α* was used as the reference

gene for the determination of the relative expression of all target genes. Primer sequences for qPCR analysis are listed in Supplementary Data 1.

## Bulk RNA-seq analysis

Paired-end RNA-Sequencing was performed on FACS-sorted macrophages derived from a pool of 20-50 embryos at 3 dpf either at baseline or 24 hpa. Three independent biological replicates for each condition were used for sequencing on an Illumina NovaSeq6000 S2 or NovaSeq6000 SP using 2×50 base pairs, each replicate represents a different cohort of embryos from which macrophages were isolated by FACS. The mutants were pre-sorted by spontaneous induction of the *irg1-KI:GFP*, and the genotypes of all RNA samples were also verified by PCR-restriction enzyme assay-based genotyping prior to library preparation using the SMARTr HT+ Nextera XT for low-input samples. 100% correspondence was found between the sorted phenotypes based on the presence or absence *of irg1* induction and the expected *nlrc3l* mutant or sibling genotypes, respectively. RNA-seq analysis followed the previously used protocol[24], which consists of modules and packages including Trimmomatic for data trimming, MultiQC for quality control, and BBMap for sequence alignment with the zebrafish reference genome (GRCz11). The differential expression analysis was performed using DESeq2. Gene ontology analysis was performed on the differentially expressed genes with Metascape[34] and clustered heat map analysis with NG-CHM[33].

## CRISPR-Cas9 targeted mutagenesis of *irg1*/acod1, *asc* and *mrc1b*

CRISPR gRNA designs, and synthesis of Cas9 mRNA and gRNAs for targeting zebrafish *irg1/acod1* (NCBI accession: NM_001126456.2; Gene ID: 795305), *asc* (NCBI accession: NM_131495.3; Gene ID: 57923), and *mrc1b* (NCBI accession: NM_001423726.1; Gene ID: 559502) followed previously described methods[82]. Co-injection of Cas9 mRNA and guide RNAs (gRNAs) was conducted in 1-cell stage zebrafish embryos. gRNA target sequences and genotyping primers are provided in Supplementary Data 1. To ensure high mutagenesis rate and large deletion mutations, two or more gRNAs were simultaneously injected with Cas9 mRNA for each gene. Injected clutches of embryos were validated to contain CRISPR-mediated mutagenesis by a T7 endonuclease assay.

## E. coli infection model combined with tail amputation

To induce systemic bacterial infection, wild-type transgenic zebrafish carrying *irg1-Kl:GFP; mpeg1:BFP* at 2 dpf were anesthetized in 0.02% Tricaine (MS-222) for direct microinjection into the brain ventricle either with 1 nl of ~1000 CFU/nL of a common laboratory *E. coli* strain (MG1655 [CGSC6300]) or 1 nl of phosphate-buffered saline (PBS) as negative controls alongside uninjected animals for each experiment. Previous studies showed brain ventricle microinjection in the zebrafish embryo rapidly spreads systemically throughout the body[82]. Persistence of macrophage activation based on *irg1* expression was assessed using the *irg1-KI:GFP* reporter with a single injection (4–6 h before amputation) or double injections (day-of and ~24 hpa), either method caused a sustained *irg1* induction and activation of macrophages. All data reflect a single brain injection to induce chronic inflammatory macrophage activation prior to muscle injury that lasted past the resolution phase at 72 hpa (Supplementary Fig. 17). After validation of a bacterial response 4-6 h post-injection based on a strong induction of the *irg1-KI:GFP* expression, the embryos were amputated to induce skeletal muscle injury as described previously. Zebrafish following tail amputation were placed back into the 28.5 °C incubator to allow recovery until embryo collection at 20-24 hpa for downstream analysis.

## scRNA-seq cell preparations and sequencing

Transgenic zebrafish carrying the GFP knock-in reporter *irg1-KI:GFP* generated in our lab were used to isolate macrophages at distinct phases of acute muscle injury response. Samples were collected at 24 hpa, representing the initial inflammatory phase, and at 48 hpa, marking the onset of tissue repair following debris clearance. Uninjured ("uncut") animals served as baseline controls. Responses were compared between wild-type (or control sibling) zebrafish and mutants with a deleterious *nlrc3l* mutation that causes chronic macrophage activation. To induce acute muscle injury, large tail amputations were performed on 2 dpf zebrafish embryos. Activated macrophages upregulate *irg1*, displaying strong GFP fluorescence, while steady-state macrophages express minimal *irg1*, resulting in faint GFP signal. All *irg1-KI:GFP*+ cells expressing any detectable level of GFP were isolated using FACS on a BD FACSMelody across four conditions: control uncut, control 24 hpa, control 48 hpa, and *nlrc3l* mutant 24 hpa. Two biological replicates were collected per condition, resulting in a total of eight samples, each consisting of pooled cells from 90 embryos. To maintain cell viability and quality, embryos were dissociated and sorted in batches of 30, with three such batches pooled to constitute a single sample. Sample collection was interleaved to evenly distribute conditions across the processing timeline. Embryo dissociation was performed in a buffer containing ROCK inhibitor, protease, and DNase with gentle mechanical grinding, and a holding solution composed of HBSS (without phenol red), 1% BSA, and ROCK inhibitor. Cells were kept on ice until downstream processing using the 10x Genomics Chromium X Controller and the GEM-X Universal 3′ Gene Expression v4 4-plex kit (PN-1000779) following the manufacturer's instructions. Cells were first verified for viability, concentration, and singleness using acridine orange and propidium iodide on the LUNA-FX7 Dual Fluorescence Cell Counter (Logos Biosystems) prior to 10x Genomics preparations. Single-cell libraries were prepared using the GEM-X OCM 3′ Chip, where samples were uniquely indexed and pooled in groups of four into two different GEMs during loading via on-chip multiplexing (OCM) technology. Reverse transcription was performed using a C1000 thermal cycler (Bio-Rad) to create cDNA libraries tagged with a cell barcode and unique molecular index (UMI). Dynabeads MyOne SILANE beads (Invitrogen) were used to purify broken GEMs prior to cDNA amplification, and final libraries with Illumina-compatible adapters were purified with SPRIselect magnetic beads (Beckam Coulter) and quantified using an Agilent Bioanalyzer High Sensitivity DNA chip (Agilent). Libraries were pooled in an equimolar ratio and sequenced on one lane of a NovaSeq X Plus 10B flow cell (Illumina) in paired-end format at Admera Health, LLC (Read 1: 28 cycles, i7 index: 10 cycles, i5 Index: 10 cycles, Read 2: 90 cycles).

## scRNA-seq analysis

Eight samples spanning four conditions were processed across two Cell Ranger Multi runs. Custom transgenes (EGFP and TagBFP) were appended to the zebrafish reference genome (GRCz11, from 10x Genomics). Gene expression matrices were generated using Cell Ranger Multi with sample demultiplexing guided by configuration CSVs specifying sample IDs, reference paths, and OCM barcodes. Downstream analysis was performed in R using Seurat (v5). Cells were filtered using miQC with a posterior cutoff of 0.80, mitochondrial content column set as "percent.mt", and nFeature_RNA as the gene feature input. Cells marked as "discard" by miQC were removed from the dataset. SCTransform was used for normalization and variance stabilization (SCT assay), followed by dimensionality reduction via PCA using the first 30 principal components. UMAPs were computed from these PCs, and clustering was performed using shared nearest neighbor (SNN) graph-based clustering across multiple resolutions, and 0.4 was selected for downstream analyses based on biological interpretability. Cells with expression >0 were classified as positive for each gene. For statistical comparison between conditions, pairwise t-tests with FDR correction were applied to both gene-positive fractions and average expression values, stratified by cluster. Differential expression analysis between conditions was performed using a pseudobulk

DESeq2 approach. For each cluster and condition comparison, raw RNA counts were aggregated by sample and filtered to retain genes with at least 10 counts in more than one sample. Only samples with ≥5 cells were included, and comparisons were restricted to condition pairs with at least two biological replicates. DESeq2 was run with fit-Type = "parametric", and estimateSizeFactors() was used with type = "poscounts". All scripts were executed on the UNC-CH's Longleaf computing cluster, and outputs were organized into directories for reference creation, quality control, clustering, visualization, and differential expression.

## Statistics and reproducibility

For pairwise comparisons, unpaired two-tailed t-tests were performed unless otherwise noted. F test was used to compare variances. For unequal variances, Welch's correction was used on the two-tailed t-test. For multiple comparisons of 3 or more groups, one-way ANOVA test was applied followed by multiple pair-wise comparisons to determine the pair(s) showing significant differences using FDR-adjusted p-values. Experiments were repeated at least twice independently with similar results. All graphical plots and statistical tests were generated using GraphPad Prism 10 unless otherwise noted.

## Reporting summary

Further information on research design is available in the Nature Portfolio Reporting Summary linked to this article.

## Data availability

Data generated in this study are provided in the publication, Supplementary Information and Source Data File (Supplementary Data 2). RNA sequencing data is freely available at the GEO repository as GSE283438 for bulk RNA-seq, and GSE301079 for scRNA-seq.

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

## Acknowledgements

We thank Maura McGrail and Jeff Essner for the generous sharing of GeneWeld plasmids and protocols, Michel Bagnat for the *tnfa:GFP* line, Brian Conlon and Anika Rueppell for the *E. coli* MG1655 support, and Peng Huang for reagent support. We are grateful to Mike Vernon and staff of UNC High-Throughput Sequencing Facility (supported by P30-CA016086 and P30-ES010126) for library preparations and RNA-sequencing, Gabrielle Cannon of UNC Advanced Analytics Core (UNC CGIBD, P30 DK034987) for scRNA-seq library preparations, researchers of UNC Bioinformatics and Analytics Research Collabrative (BARC): Tara Brennan, Matt Niederhuber, Ismael Gomez, and Austin Hepperla for processing and analzying scRNA-seq data, and Michelle Altemara and staff of UNC Zebrafish Aquaculture Core Facility for zebrafish housing and care. This work was supported by NIH NIGMS grant R35GM124719 to C.E.S.

## Author contributions

C.G.S.: data analysis and visualization, experimentation (injury model, time course analysis, epistasis experiments, multiple mutant phenotyping, timelapse and static injury imaging, CRISPR), writing; M.H.: data analysis and visualization, experimentation (bulk and single-cell RNA-seq analysis, time course analysis, imaging, injury analysis, cloning, qPCR, HCR, scRNA-seq, IHC), writing; E.B.: data analysis and visualization, experimentation (KI versus TOL2 imaging, CRISPR, injury analysis, scRNA-seq, infection); Y.N.W.: data analysis and visualization, experimentation (subtype and muscle imaging, genetic experiments, infection); A.M.R.: experimentation (generation of GFP knock-in and TOL2 lines, germ-free experimentation); A.B.: experimentation (FACS sorting and protocol development, bulk RNA-seq preparations); JZ: experimentation (KI versus TOL2 analysis); K.S.: experimentation (initial IHC analysis of injury); C.E.S.: conceptualization and experimental designs, data analysis and visualization, supervision, funding acquisition, writing- original draft and all revisions. All authors reviewed and edited the manuscript.

## Competing interests

The authors declare no competing interests.
