## [Transparent Peer Review file · Nature Communications]

Chronic macrophage activation derails muscle repair by disrupting mannose-receptor-linked plasticity revealed by endogenous *irg1/acod1* tracking

Corresponding Author: Professor Celia Shiau

Version 0:

Reviewer comments:

Reviewer #1

(Remarks to the Author)

This study investigates distinct macrophage populations during the inflammatory response and their significance in muscle repair. To address this, the authors generated a KI *irg1*:GFP reporter zebrafish line and combined it with *nlr3* deficiency, which is known to promote chronic inflammation. Using RNA-seq analysis, they revealed the dynamic expression of *mrc1b* across macrophage polarization states and highlighted its importance in muscle repair. The manuscript is well written, the experiments are carefully designed and executed, and most conclusions are supported by the data. However, additional experiments are necessary to further validate the role of the three macrophage populations in tissue repair and the contribution of *mrc1b*.

Major Points

1. Figure 4 and Supplementary Movies 3 & 4:

o The three macrophage subtypes should be quantified. The movies, as presented, do not allow a clear appreciation of the subtypes, and playback speed is too fast. Slower playback and subtype tracking would provide crucial insights.

2. Figures 5, 6, and Supplementary Figure 13:

o Phagocytosis events are difficult to discern and should be quantified.

o The use of the *irg1*:GFP KI line to track macrophages may not be appropriate since *irg1* expression is altered in *nlr3*-deficient larvae. Alternative macrophage markers (*mepg1*, *mfap4*) should be employed for proper identification and tracking.

3. Supplementary Figure 15:

o Bulk RNA-seq on sorted macrophages does not adequately resolve the different populations. Were *irg1*-low and *irg1*-high macrophages pooled? How was the purity of sorted populations verified using additional markers (*mpeg1*, *mfap4*, *mpx*, *lyz*)?

o Validation by RT-qPCR was performed on whole larvae rather than sorted macrophages. For example, *mmp9* is predominantly expressed by neutrophils and keratinocytes, which could confound the results. Additionally, as *irg1*-positive cells were sorted, *irg1*-positive neutrophils may have contaminated the macrophage population.

o Single-cell RNA-seq would provide a more granular characterization of the three macrophage populations and significantly strengthen the conclusions.

4. Figure 6:

o Quantification of the three macrophage subtypes and the effect of *mrc1b* knockdown (KD) is necessary. Additionally, the editing efficiency of *mrc1b* should be determined and reported.

Minor Points

1. Figure 1:

o This figure could be moved to the supplementary information section.

2. Lines 155–171:

o The paragraph needs revision, as certain sentences are repeated.

3. Figure 4c:

o The graphs are overly complex. Presenting the data as mean \pm SEM would enhance clarity.

4. Figure 3C:

o The term "*asc*-/-" is incorrect. Instead, use "crispant," "deficient," or "knockdown" (KD) as appropriate. Furthermore, these results need validation, including *asc* editing efficiency and/or *cas1* activity.

5. Supplementary Figure 14:

o *irf8* deficiency results in macrophage depletion and neutrophilia. Since neutrophils also express *irg1*, the *irg1*-positive cells

identified in *irf8*-deficient larvae may include neutrophils. This point requires clarification.

6. Discussion:

o The advantages of the *irg1*:GFP KI reporter line over the well-characterized *tnfa*:GFP reporter line should be discussed. The results suggest that the two reporters yield similar findings, which raises questions about the added value of the *irg1*:GFP line.

Reviewer #2

(Remarks to the Author)

Chronic inflammation or immune disorders that impair tissue regeneration represent a significant clinical challenge, with the underlying mechanisms still not fully understood. Leveraging zebrafish, an excellent model organism for studying tissue regeneration, Spencer et al. employed an *irg1/acod1* reporter line to reveal significant heterogeneity in macrophage behavior during muscle regeneration in wild-type zebrafish. Interestingly, they found that in *nlrc3l* mutants, which the authors previously showed to induce chronic inflammation and macrophage developmental defect, phagocytic and muscle-encasing macrophages were absent. They further suggested that a deficiency in the mannose receptor contributes to macrophage dysfunction under chronic activation. This study provides a unique system for observing the dynamic *in vivo* behavior of macrophages in tissue regeneration. However, several critical issues must be addressed before the study is suitable for publication in Nature Communications.

1. The *nlrc3l* mutant is not an ideal model for studying the impact of chronic inflammation on tissue regeneration. As the authors previously demonstrated, it not only exhibits an automatic pro-inflammatory response but also causes defects in macrophage development. This raises the concern that the observed phenotypes are primarily due to developmental abnormalities in macrophages rather than chronic inflammation *per se*. To increase the generalizability and relevance of their findings, the authors should explore additional chronic inflammation models to demonstrate broader significance beyond the *nlrc3l* mutant.

2. Observational data on macrophage behavior require deeper mechanistic insights.

The authors employed the *irg1* transgenic zebrafish to observe macrophage behavior *in vivo* during acute muscle injury, confirming that macrophages adopt distinct subtypes during muscle regeneration. While this behavioral data is valuable, it remains largely observational.

Regarding mechanistic exploration, the study links *nlrc3* and *myd88* as potential pathways influencing chronic inflammation and muscle regeneration. However, given that *myd88* is a well-established master regulator of inflammation/infection, these findings are somewhat predictable. The authors should aim to identify more precise regulatory steps within the *nlrc3*-*myd88* pathway.

Similarly, the study attempts to establish CD206 as a critical regulator of macrophages in muscle repair. However, since tissue repair involves extensive cell death and macrophage phagocytosis, it is expected that CD206, a key marker of phagocytosis, would be upregulated. A more in-depth investigation is needed to uncover why certain macrophage subtypes (subtypes 2 and 3, specifically those capable of phagocytosis) are absent in chronic inflammation. Sorting and analyzing these subtypes individually, or employing single-cell sequencing to identify differential pathways, followed by functional validation, could provide meaningful insights. The current comparison of injured versus uninjured macrophages is too simplistic to pinpoint these critical pathways.

3. The authors state: "In *nlrc3l* mutants, restoring macrophage *nlrc3l* expression, deleting *myd88*, or both reversed the inappropriate macrophage activation, as evidenced by *irg1* and *tnfa* downregulation, with the combination fully restoring a normal macrophage state (Fig. 3 and Supplementary Figs. 4, 6). These results implicate both cell-autonomous and non-cell-autonomous mechanisms in the inappropriate immune activation."

However, the manuscript does not provide direct evidence for the involvement of non-cell-autonomous mechanisms. The authors need to clarify or provide experimental data to support this conclusion.

4. The data on *irf8* mutants following tissue regeneration does not offer significant insights. Any mutation affecting macrophage function or development would be expected to exhibit tissue regeneration defects. Further exploration is necessary to demonstrate the unique relevance of *irf8* in this context.

Reviewer #3

(Remarks to the Author)

Spencer et al. first produced *irg1/acod1*-KI zebrafish for tracing activated macrophages. They compared the KI zebrafish with existing models and KO lines and demonstrated the effective and non-functional impairment of the newly established line. Using this line, next, the authors investigated the properties of activated macrophages in a chronic inflammatory state induced by the loss of *nlrc3*, which could be suppressed by the loss of *Myd88* and the restoration of *nlrc3* expression. Furthermore, using a muscle regeneration model, the authors verified the regeneration process including macrophage dynamics under control and *nlrc3*-deficient conditions, and demonstrated that regeneration does not occur when activated macrophages continue to exist, as evidenced by rescue experiments. Finally, the authors focused on the decreased expression of CD206 in the chronically activated macrophages and concluded that CD206 deficiency causes muscle regeneration failure and that CD206 in macrophages is responsible for regeneration failure under persistent inflammation. Overall, the experiments are well-conducted, and the data are convincing. However, this reviewer has some concerns about the model and results of CD206-KD. In particular, it is questionable whether the conceptual advance of this study meets the criteria of Nature Communications.

Major comments

1) The chronic model used here is an artificial model. It is unclear whether this model is applied to naturally occurring disease models. Based on the results found in this study, is it possible to verify the efficacy of the treatment by targeting a conceptually new molecule or pathway?

2) Given the role of CD206 in macrophage-mediated phagocytosis, it is unsurprising that CD206 knockdown (CD206-KD) results in impaired muscle regeneration. However, the relationship between CD206-KD and MyD88-null experiments remains unclear. Does the loss of MyD88 in NLRC3-null conditions restore CD206 expression? If so, how is CD206 expression regulated under chronic conditions?

3) Lines 65-66 ; dynamic in vivo and longitudinal analyses of cellular interactions across distinct phases of the injury and repair process are lacking

Some studies conducted dynamic and longitudinal analyses using scRNA-seq. For instance, the following studies conducted dynamic, longitudinal, interactome analyses using scRNA-seq during muscle regeneration in detail. This reviewer recommends citing these papers and re-rewrite more precisely.

iScience. 2020 Apr 24;23(4):100993. doi: 10.1016/j.isci.2020.100993.

Cell Rep. 2020 Mar 10;30(10):3583-3595.e5. doi: 10.1016/j.celrep.2020.02.067.

4) Lines 204; It is difficult to find Mrc1b/cd206 in the figure.

5) Line 244 ;

What is ASC? Please add a description of the relationship between ASC and nlrc3l.

6) Lines 451-452; The precise role of macrophages in skeletal muscle injury and repair remains incompletely understood.

Please describe what the precise role is not understood. In addition, the authors need to underscore what the present study reveals in the role of macrophages in muscle regeneration. This reviewer thinks that mechanisms inducing chronic inflammatory conditions in muscle regeneration are unexplored. However, the disruption of muscle regeneration by chronic inflammation is not conceptually new.

7) Lines 358-361, 461

The authors described that chronically activated macrophages inhibit satellite cell activation. There is no evidence that chronic macrophages directly inhibit satellite cell activation. Is it possible that chronically activated macrophages inhibit not the activation, but the proliferation? And is it possible that the chronic macrophages disrupt the balance of pro-inflammatory macrophages inducing satellite cell activation or proliferation?

Version 1:

Reviewer comments:

Reviewer #1

(Remarks to the Author)

The authors have satisfactorily addressed all my concerns in the revised manuscript. Importantly, they have now included a comprehensive scRNA-seq analysis, which provides high-resolution validation of the bulk RNA-seq data presented in the original version. These new data not only strengthen the conclusions but also significantly expand the impact of the work.

The exhaustive characterization of macrophages and neutrophils in the context of chronic inflammation is exceptional and provides information of broad relevance to the field, beyond the scope of this article. Of particular interest is the identification of a hybrid M1/M2 macrophage state that predominates under chronic inflammation, as well as the delineation of numerous markers of macrophage polarization, such as ctsk.

Overall, the manuscript is now excellent in all respects. The study represents a major advance in our understanding of immune cell heterogeneity and plasticity during chronic inflammation and tissue repair.

I strongly recommend acceptance of this article.

Reviewer #2

(Remarks to the Author)

The overall quality of the manuscript has improved significantly after revision, and some of my previous concerns have been addressed. However, several issues remain that the authors must further clarify and revise.

Major points

1. I acknowledge and agree that zebrafish Nlrc3l may represent a unique NLR without a clear mammalian ortholog. Nevertheless, its mechanism of action is expected to be evolutionarily conserved. Currently, there is insufficient evidence to

support that zebrafish Nlrc3l functions through a completely distinct mechanism from mammals. As the authors noted, zebrafish nlr3l lacks the LRR domain. In mammals, the only reported NLR lacking an LRR is NLRP10 (Kanneganti TD et al., *Immunity* 2007). Previous studies have demonstrated that NLRP10 can induce ASC speck formation and participate in anti-inflammatory regulation in human keratinocytes and mouse intestinal epithelial cells (Próchnicki T et al., *Nat Immunol* 2023; Zheng D et al., *Nat Immunol* 2023). However, the authors' current conclusion that Nlrc3l is independent of Asc is based solely on low-resolution stereomicroscope imaging in Fig. 3C, which seems premature. Zebrafish are three-dimensional organisms, where signals from the opposite side are easily missed, and out-of-focus GFP signals are inherently attenuated. More reliable approaches should include qRT-PCR or quantitative GFP Western blot. Moreover, the number of irg1⁺ cells shown in Fig. 3C appears to be slightly reduced in Nlrc3l/Asc double mutants. Even within zebrafish studies, evidence already suggests that Nlrc3l regulates microglial cell death in part through Asc (Chang MX et al., *Dev Comp Immunol* 2021; Wang T et al., *J Genet Genomics* 2019). Taken together with mammalian and zebrafish findings, as well as potential misinterpretation of the current data, it remains premature to exclude the possibility that Nlrc3l phenotypes are ASC-related. Additional experimental data are required.

2. The authors attempt to explain the phenotype of Nlrc3l deficiency via the “master regulator” Myd88, but this level of mechanistic interpretation is insufficient for a Nature Communications-level study. This is somewhat analogous to showing that an anti-inflammatory drug partially rescues the phenotype—an expected observation, but not true mechanistic insight. Furthermore, the irg1 used in the Myd88-dependent assays is expressed only in myeloid cells, so the conclusion applies only to this lineage, not to the entire organism. Similarly, the irg1 reporter line and mpeg1-nlr3l rescue address immune cell-specific phenotypes, but cannot rule out contributions from non-immune cells. In Fig. 3D, the difference between the last two groups exceeds threefold, yet the small sample size ($n \approx 5$ per group) makes the lack of statistical significance unconvincing. The subsequent germ-free model experiments further underscore the presence of non-cell-autonomous mechanisms, reinforcing this concern. In such models, commensal bacteria primarily interact with the skin and intestinal epithelium, where NLRP10 has been shown to be highly expressed. Given that the tissue injury model used by the authors inherently damages epithelial cells, a more plausible explanation is that Nlrc3l deficiency triggers commensal-driven mucosal systemic inflammation, which in turn disrupts macrophage function. This hypothesis may actually represent a more valuable research direction. I encourage the authors to integrate findings from NLRP10-related studies to explore this potential mechanistic conservation between zebrafish Nlrc3l and mammalian NLRP10.

3. The association of CD206 with tissue regeneration has been well documented (Shook B et al., *J Invest Dermatol* 2016; Honda A et al., *EMBO Rep* 2025). While not necessarily specific to skeletal muscle regeneration, the underlying logic remains the same. The current findings resemble more of a “me-too” type of study. The authors should sufficiently reference and acknowledge existing literature and refine the novelty of their study in light of prior research.

Minor points

1. In Fig. S17, the bacterial injection site is not the brain ventricle.

Reviewer #3

(Remarks to the Author)

In the first round of review, this reviewer raised three major concerns: (1) the generalization of the findings in the Nlrc1 mutant model, (2) the expression in Nlrc3l/Myd88 double mutant mice, and (3) the conceptual advance of this study. The authors have sincerely addressed these concerns with new data and descriptions. In addition, the authors performed scRNA-seq analyses, which strengthen their findings on the alteration of macrophage subtypes in chronic environments. Please address the following comments to further improve the manuscript.

- There are several instances in the main text where the specific figure panels are not indicated. Please provide this information consistently. In particular, the descriptions of Figures 3, 5, 8, and supplemental figures do not mention the corresponding panels.
- Related to the above comments, it is unclear which figure corresponds to the description of the scRNA-seq results. The relationship between subtypes 1–3 and the scRNA-seq clusters represents a crucial result in this manuscript. The reviewer recommends that all genes used to analyze the relationship between imaging subtypes and scRNA-seq clusters be presented in the main figure, so that readers can readily recognize them.
- Figure 6, Figure 4, Supplemental Figure 11, 15 contain errors in their preparation.
- The meaning of the sample name in Supplemental Figure 19f is unclear. What does 3d_1 signify?

Version 2:

Reviewer comments:

Reviewer #2

(Remarks to the Author)

All of my concerns have been addressed in this version.

Reviewer #3

(Remarks to the Author)

The authors have sincerely addressed all remaining concerns raised by this reviewer. I am satisfied with the revisions and have no further requests.

Point-by-point response to the reviewers' comments is shown in *italicized text*.

Reviewer #1 (Remarks to the Author):

This study investigates distinct macrophage populations during the inflammatory response and their significance in muscle repair. To address this, the authors generated a KI *irg1*:GFP reporter zebrafish line and combined it with *nlrc3* deficiency, which is known to promote chronic inflammation. Using RNA-seq analysis, they revealed the dynamic expression of *mrc1b* across macrophage polarization states and highlighted its importance in muscle repair. The manuscript is well written, the experiments are carefully designed and executed, and most conclusions are supported by the data. However, additional experiments are necessary to further validate the role of the three macrophage populations in tissue repair and the contribution of *mrc1b*.

Major Points

1. Figure 4 and Supplementary Movies 3 & 4:

o The three macrophage subtypes should be quantified. The movies, as presented, do not allow a clear appreciation of the subtypes, and playback speed is too fast. Slower playback and subtype tracking would provide crucial insights.

Thank you for your helpful feedback. In response, we have added new Supplementary Movie files corresponding to the original Movies 3 and 4, now renamed Supplementary Movies 3 and 5 (timelapse imaging of control sibling and mutant, respectively). These are accompanied by Supplementary Movies 4 and 6, which show 3D renderings of the same datasets with macrophage subtypes annotated and presented at slower playback speed to highlight cell behavioral and morphological features for quantitative analyses.

Supporting the additional movie files (Supplementary Movies 4 and 6), we have included new quantitative analyses characterizing the three macrophage subtypes during the initial injury response at 24 hpa, now presented in Figure 4f. Examples of individual macrophages from each subtype are also shown in Figure 4g, using Imaris to digitally isolate fluorescent data to show individually rendered macrophage cellular outlines for each subtype.

We define the three subtypes—subtype 1 (“mobile”), subtype 2 (“cluster”), and subtype 3 (“encasing”)—based on distinct behavioral and morphological features:

-Subtype 1 (mobile): Highly motile cells, with increased speed and distance traveled (Fig. 4f).

-Subtype 2 (cluster): Characterized by increased vesicular content, with a higher phagocytic index (cell-to-cytosol volume ratio), indicating large intracellular vesicles such as phagosomes or endosomes (Fig. 4f).

-Subtype 3 (encasing): Largely stationary cells with elongated, columnar morphology. These cells encircle myocytes and adopt a muscle-cell-like shape, showing the largest cell volume and greater overall sphericity, particularly evident in the lowest 10th percentile of the population distribution (Fig. 4f–g, Supplementary Fig. 12).

Our data also show that mutants lack subtypes 2 and 3, highlighting subtype-specific alterations in macrophage behavior. Relevant text describing these subtypes and their associated quantifications has been added to the Results section and to Figure 4 legend.

2. Figures 5, 6, and Supplementary Figure 13:

o Phagocytosis events are difficult to discern and should be quantified.

We have renamed subtype 2 as “clustering” to better reflect its defining characteristic, namely the tendency of these macrophages to coalesce and adopt a rounded morphology with visibly increased

vesicular content. To support this, we now include a phagocytic index quantification based on morphological features, specifically the ratio of total cellular to cytosolic volume, which is indicative of vesicular content and thereby phagocytic activity.

Our analysis shows that subtype 2 cells exhibit more than double the phagocytic index compared to the other macrophage subtypes, suggesting that they contain, on average, more than twice the vesicular volume (Fig. 4f).

Please note that the previous Supplementary Fig. 13 is now Supplementary Fig. 12.

o The use of the *irg1*:GFP KI line to track macrophages may not be appropriate since *irg1* expression is altered in *nlrc3*-deficient larvae. Alternative macrophage markers (*mpeg1*, *mfap4*) should be employed for proper identification and tracking.

Thank you for this insightful question. We agree that validating the *irg1*-KI:GFP reporter is important. To address this, we performed dual labeling with the established *mpeg1*:BFP line wherever possible. Our results demonstrate that *irg1*-KI:GFP reliably marks macrophages with high specificity and intensity, including after tail-cut injury.

At baseline, *irg1*-KI:GFP fully overlaps with *mpeg1*:BFP in macrophages (Fig. 3, Supp. Figs. 3, 8), confirming its specificity. After injury, all cells clustering or encasing muscle at the wound site remain *mpeg1*⁺ (Fig. 4b, Supp. Fig. 13), indicating that *irg1*-KI:GFP continues to label macrophages specifically. Although a small number of neutrophils show weak knock-in reporter expression post-injury (7.6% in controls, 10.7% in mutants; none in controls or 4.3% in mutants at baseline), these cells are rare and express significantly lower GFP levels than macrophages. We now include quantitative thresholds (Supp. Fig. 11b; 10 A.U. detection, 30 A.U. bright GFP) showing that macrophages consistently exhibit strong expression, while neutrophils rarely reach the bright threshold.

This is further substantiated by our single-cell RNA-seq data: while a portion of the *irg1*-KI:GFP⁺ cells sequenced were neutrophils, more prevalent in *nlrc3l* mutants, the levels of GFP expression in the neutrophil clusters (clusters 4/11/12) were significantly lower than the macrophage clusters (Supp. Fig. 18d). Moreover, quantification of the eGFP transcripts in the sequenced cells clearly show a large difference where neutrophils range from 0 to less than 30 normalized counts per sample while macrophages range is at least 10 to 100 times more counts at 420 to 2,700 in activated macrophages (Supp. Fig. 19h). We now include this analysis and quantification in the manuscript to highlight the large expression gap.

Importantly, while *mpeg1*:BFP served as a valuable control, its very low fluorescence precluded use in live animal sorting on a fluorescent stereoscope and cell sorting. The *irg1*-KI:GFP reporter was therefore indispensable for identifying live *nlrc3l* mutants without requiring post-fix genotyping, thereby enabling the core time course and live-cell experiments presented in this study.

In summary, although low-level knock-in reporter expression can occur in a small number of neutrophils under specific conditions such as inflammation or injury, *irg1*-KI:GFP remains a robust tool for tracking macrophages—both at baseline, when neutrophils do not express the reporter, and upon activation, when macrophages express the reporter at high intensity and in much greater abundance than neutrophils. The reporter makes macrophages readily distinguishable even under perturbation and has been essential for the experiments in this study.

3. Supplementary Figure 15:

o Bulk RNA-seq on sorted macrophages does not adequately resolve the different populations. Were *irg1*-low and *irg1*-high macrophages pooled? How was the purity of sorted populations verified using additional markers (*mpeg1*, *mfap4*, *mpx*, *lyz*)?

We thank the reviewer for raising this important point. The goal of our bulk RNA-seq was to characterize cells expressing irg1-KI:GFP at baseline. For cell sorting, we used stringent GFP gating to distinguish “high” vs. “low” GFP-expressing populations, based on thresholds set using non-transgenic negative controls (see Supplementary Fig. 21). Specifically, irg1-high cells (found only in mutants) exhibited GFP levels exceeding those of controls, while irg1-low cells (present in both groups) had lower GFP intensity. FACS was performed under high-purity settings with low event rates to ensure collection accuracy.

To further validate and resolve the identity of these populations, we have now included single-cell RNA-seq analyses (Figs. 7, 8 and Supplementary Figs. 18-20). These data show that irg1-high cells consistently represent macrophage cell identities, while irg1-low cells include a minority neutrophil cell population—primarily in mutant animals—alongside macrophage cells. This aligns with the bulk RNA-seq data, where irg1-high cells exclusively expressed macrophage markers, and irg1-low cells from mutants expressed both macrophage and neutrophil markers (Fig. 2).

Together, the combination of stringent FACS gating and new single-cell analysis provides a clear resolution of cell identities and supports the specificity of our macrophage-related conclusions.

o Validation by RT-qPCR was performed on whole larvae rather than sorted macrophages. For example, mmp9 is predominantly expressed by neutrophils and keratinocytes, which could confound the results. Additionally, as irg1-positive cells were sorted, irg1-positive neutrophils may have contaminated the macrophage population.

Single-cell RNA-seq would provide a more granular characterization of the three macrophage populations and significantly strengthen the conclusions.

We agree with the reviewer that single-cell analysis would help clarify the complexity of the macrophage populations and distinguish the transcriptome changes between macrophages and neutrophils. As discussed, we have now performed scRNA-seq, which provides the necessary clarity between macrophages and neutrophils, and further characterization of macrophage subsets. These new data, presented in Figs. 7, 8 and Supplementary Figs. 18 and 19, reveal macrophage-specific responses to acute tail-cut injury, as well as mutation-induced alterations driven by chronic inflammation.

Importantly, the single-cell analysis conveys differences in the macrophage population during an injury response, including hybrid states in the initial inflammatory phase at 24 hpa, marked by co-expression of M1 and M2 markers in individual macrophages in controls and nlr3l mutants, albeit even more pronounced in mutants (Fig. 8 and Supplementary Fig. 19e). Moreover, we observe a significant loss of subtypes 2 and 3 corresponding to repair-associated macrophages expressing ctsk/fabp4 and tgfb/sb:cb81 (or ZFP36L1-like), respectively, in nlr3l mutants.

To isolate macrophage-specific profiles, we used scRNA-seq data to subcluster macrophage-identity cells to analyze only the cells making up the macrophages (clusters 0-9, except 4), excluding neutrophil populations (clusters 4, 11, and 12), which is particularly important for genes like mmp9, which are expressed in neutrophils (Supplementary Fig. 18e). The macrophage-specific scRNA-seq analysis revealed and confirmed significant mmp9 upregulation at 24 hpa in both control and mutant macrophages post-injury, with the highest expression in mutant macrophages, while mmp9 expression remained low or undetectable in macrophages at baseline and resolution (Fig. 7i, Supplementary Figs. 18e and 19g).

These findings not only validate but also extend our bulk RNA-seq findings, offering high-resolution insight into both cell identity and the functional consequences of chronic inflammation on macrophage heterogeneity, which is clearly diminished in the chronically activated nlr3l mutants.

4. Figure 6:

o Quantification of the three macrophage subtypes and the effect of *mrc1b* knockdown (KD) is necessary. Additionally, the editing efficiency of *mrc1b* should be determined and reported.

*We agree and appreciate the reviewer's suggestion and have now addressed this by presenting data in Supplementary Fig. 16 demonstrating the efficiency of *mrc1b* editing using T7 endonuclease assay and DNA sequencing on embryos injected with Cas9 mRNA with *mrc1b*-targeting gRNAs (#4 and #31) compared with universal gRNA/Cas9 injected negative controls in Supplementary Fig. 16. T7 assay shows ~96% CRISPR efficiency, with Sanger sequencing confirming *mrc1b*-targeting induced indels causing frameshifts and premature stops. Additionally, sequencing *mrc1b*-targeted embryos suggests CRISPR often mutates only one allele, resulting in heterozygous, non-deleterious outcomes, which is consistent with the partial phenotypes observed (Fig. 6).*

*Furthermore, we analyzed the macrophage subtypes at 24 hpa in *mrc1b* knockdown animals that indicated a similar trend of reduction, although not statistically significant, in subtypes 2 (clustering) and 3 (muscle-encasing) with those observed in *nlr3l* mutants with chronic inflammation (Supplementary Fig. 16d), supporting the involvement of *mrc1b* in driving the observed mutant phenotype. Due to the variable degree of gene knockdown from transient CRISPR, we decided to inject CRISPR reagents into zebrafish that were heterozygous for a stable *mrc1b* knockout to increase the rate of gene knockdown. Separating the heterozygous fish from sibling WT showed a more clear trend of reduction particularly in the muscle-encasing macrophage subpopulation (Supplementary Fig. 16d).*

*In addition, to strengthen the function of *mrc1b* in acute muscle injury repair, we obtained a putative *mrc1b* knockout allele (*sa18640*) from ZIRC, derived from a large-scale mutagenesis in the UK Sanger Center Zebrafish Project. We confirmed the *sa18640* mutation by sequencing and demonstrated Mendelian segregation in progeny. This mutation introduces a G>T point substitution resulting in a premature stop codon in *mrc1b*. To our knowledge, this novel allele has not been previously validated or reported in any study. Data verifying the mutation and its associated phenotype are also included in the newly added Supplementary Fig. 16 and Fig. 6g, respectively. As generating new transgenic *sa18640* mutant zebrafish would require several additional months—beyond the scope of this study—we focused on using this mutation for muscle actin staining to assess injury repair following *sa18640* deletion rather than also macrophage subtyping. These results were consistent to that found in *mrc1b* knockdown animals by CRISPR.*

Minor Points

1. Figure 1:

o This figure could be moved to the supplementary information section.

Thank you for your comment. We chose to include this figure in the main manuscript to highlight the reporter system, which was technically challenging to generate and had a low success rate. We believe its inclusion is important to emphasize the novelty and technical significance of this tool, which underpins several key findings in the study. That said, if space constraints are a concern, we are open to streamlining the figure and relocating detailed components to the supplementary section while retaining a simplified version in the main text for clarity and emphasis.

2. Lines 155–171:

o The paragraph needs revision, as certain sentences are repeated.

*This paragraph describes the rationale and the results of the experiments comparing transposon versus knock-in based GFP reporters for *irg1*. We have refined and shortened the text.*

3. Figure 4c:

o The graphs are overly complex. Presenting the data as mean \pm SEM would enhance clarity.

We appreciate the suggestion regarding the plot format. Because this was a longitudinal study carefully designed and tediously conducted to track macrophage cell dynamics in the same individual animals over multiple timepoints, this plot was chosen to preserve that continuity. A bar graph, for example, would not effectively convey the temporal progression of an individual, or the variation between individual animals. To present the data comprehensively, we have made each line in the plot represent the measurement of a specific parameter (y-axis) per timepoint for a single animal, allowing visualization of within-subject changes.

To improve visualization of the plots for clarity, we have revised the graphs to show **bolder** black and red lines to indicate the group means for control and mutant animals, respectively, with shaded areas representing the standard error of means (SEM), which we hope helps with enhancing clarity.

4. Figure 3C:

o The term "asc-/-" is incorrect. Instead, use "crispant," "deficient," or "knockdown" (KD) as appropriate. Furthermore, these results need validation, including asc editing efficiency and/or casp1 activity.

We are using a stable knockout of asc that we generated using CRISPR, and the mutant line has been stably maintained for several generations as allele bcz82 (also known as nc303cs); so the denotation of "asc -/-" is correct. Thank you for pointing out the unintended omission of the supporting data in the initial manuscript version; we have now provided Sanger sequencing validation of the missense mutation due to a 4-bp deletion in the asc (or pycard) coding region in Supplementary Fig. 7a. Mutants segregate according to Mendelian ratios from a heterozygous incross, and genotyping using the primers in the Resources Table has provided a reliable method to identify the genotypes for this asc mutant bcz82 allele.

5. Supplementary Figure 14:

o irf8 deficiency results in macrophage depletion and neutrophilia. Since neutrophils also express irg1, the irg1-positive cells identified in irf8-deficient larvae may include neutrophils. This point requires clarification.

There was an error in the original figure legend, which incorrectly stated that macrophages were labeled using the irg1-KI:GFP reporter. We have corrected the figure and legend to reflect that, in the injury experiments, both mutant and sibling fish carried the macrophage-specific mpeg1:GFP reporter. Importantly, the irf8 mutant fish were available only with the mpeg1:GFP reporter background.

6. Discussion:

o The advantages of the irg1:GFP KI reporter line over the well-characterized tnfa:GFP reporter line should be discussed. The results suggest that the two reporters yield similar findings, which raises questions about the added value of the irg1:GFP line.

We appreciate the opportunity to clarify the unique value of the irg1-KI:GFP reporter line in comparison to the existing tnfa:GFP reporter.

We have added to Discussion the following text to address the raised question:

"While existing reporters like tnfa:GFP label macrophage activation, they reflect distinct pathways. tnfa, encoding a pro-inflammatory cytokine, marks inflammatory signaling, while irg1, a mitochondrial enzyme, tracks metabolic reprogramming. Both show partial overlap during peak inflammation (~24 hpa; Fig. 7j, Supplementary Fig. 17), but irg1 expression persists into resolution, indicating broader, sustained activation. scRNA-seq data support this, showing irg1 is more broadly expressed post-injury than tnfa (Fig. 7, Supplementary Fig. 17). irg1-KI:GFP, generated via targeted knock-in, ensures consistent, quantifiable expression across animals. In contrast, expression of tnfa:GFP, inserted via transposon, varies with copy number and integration site. tnfa:GFP also shows expression in non-immune cells, including CNS and PNS neurons (Smith et al., 2017 Plos Genetics PMC5397050;

Supplementary Figs. 4, 9), complicating immune-specific analysis, such as by FACS and transcriptomics. *irg1-KI:GFP*, by contrast, is predominantly macrophage-specific, where its upregulation marks a shift in activation state without detectable expression in non-immune cells. Thus, *irg1-KI:GFP* offers a stable, specific, and quantitative *in vivo* tool for tracking immune activation and metabolic state unmatched by existing reporters.”

Ultimately, *irg1-KI:GFP* fills a critical gap in the current toolkit: it is the first immune reporter line that faithfully tracks *irg1* transcription, a marker of metabolic activation, in a quantitative, genetically precise, and immune cell-type manner. This makes it an unmatched tool for dissecting immune cell states, particularly macrophage and neutrophil biology, *in vivo*.

Reviewer #2 (Remarks to the Author):

Chronic inflammation or immune disorders that impair tissue regeneration represent a significant clinical challenge, with the underlying mechanisms still not fully understood. Leveraging zebrafish, an excellent model organism for studying tissue regeneration, Spencer et al. employed an *irg1/acod1* reporter line to reveal significant heterogeneity in macrophage behavior during muscle regeneration in wild-type zebrafish. Interestingly, they found that in *nlrc3l* mutants, which the authors previously showed to induce chronic inflammation and macrophage developmental defect, phagocytic and muscle-encasing macrophages were absent. They further suggested that a deficiency in the mannose receptor contributes to macrophage dysfunction under chronic activation. This study provides a unique system for observing the dynamic *in vivo* behavior of macrophages in tissue regeneration. However, several critical issues must be addressed before the study is suitable for publication in Nature Communications.

1. The *nlrc3l* mutant is not an ideal model for studying the impact of chronic inflammation on tissue regeneration. As the authors previously demonstrated, it not only exhibits an automatic pro-inflammatory response but also causes defects in macrophage development. This raises the concern that the observed phenotypes are primarily due to developmental abnormalities in macrophages rather than chronic inflammation per se. To increase the generalizability and relevance of their findings, the authors should explore additional chronic inflammation models to demonstrate broader significance beyond the *nlrc3l* mutant.

*The reviewer raises a valid point that warrants clarification of our previous studies. In the original publication (Shiau et al., 2013, Cell Reports), the novel *nlrc3l* mutation was uncovered from a forward genetic screen in zebrafish led by the senior author of the current manuscript (Shiau). That study revealed the unexpected finding that macrophages failed to form microglia, because they did not migrate into the brain rather than having developmental defects (such as lack of maturation). This was attributed to inappropriate inflammatory activation of macrophages that diverted their normal migration: the macrophages behaved as if responding to an active infection, despite the absence of any immune challenge. The molecular mechanisms underlying this aberrant activation remained unknown at the time.*

*Our current study, along with recent work (Kwon et al., 2022, Communications Biology), builds on this foundation by showing that macrophages in *nlrc3l* mutants are functional and capable of responding to infection. However, they are constitutively activated at baseline, expressing pro-inflammatory cytokines and *irg1/acod1*. We therefore consider the *nlrc3l* mutant zebrafish a uniquely valuable genetic model to study inappropriate macrophages activation, providing a consistent and genetically controlled system for chronic inflammatory activation.*

Nonetheless, we agree that including an independent model strengthens the generality of our findings. We therefore developed an infection-based chronic inflammation model to examine how persistent macrophages activation affects acute muscle injury repair. Using a commonly employed *E. coli* lab strain (MG1655), we microinjected a high dose of bacteria into the brain ventricle of early embryos prior to injury and monitored both injury response and macrophages activation over time. We confirmed that our paradigm induced robust, systemic, and enduring macrophage activation (new data presented in Supplementary Fig. 17). Using this new model, we found that the muscle repair defects closely resemble those seen in *nlr3l* mutants. Furthermore, these defects were associated with the loss of the two key macrophages subtypes (clustering and muscle-encasing) and downregulation of macrophage *mrc1b* expression, all reminiscent of *nlr3l* mutants (Supplementary Fig. 17).

We have added this new data in Supplementary Fig. 17 and the following text to the Results section: “To further explore the generalizability to additional paradigms of chronic macrophage activation, we developed a model based on persistent systemic *E. coli* infection (Supplementary Fig. 17a-c), and showed that similar to *nlr3l* mutants, the chronic macrophage activation caused by infection disrupted normal skeletal muscle repair (Supplementary Fig. 17d-e), and was associated with a *mrc1b* downregulation and reduction of both clustering and encasing macrophage subtypes (Supplementary Fig. 17f-i).”

2. Observational data on macrophage behavior require deeper mechanistic insights.

The authors employed the *irg1* transgenic zebrafish to observe macrophage behavior in vivo during acute muscle injury, confirming that macrophages adopt distinct subtypes during muscle regeneration. While this behavioral data is valuable, it remains largely observational.

Regarding mechanistic exploration, the study links *nlr3* and *myd88* as potential pathways influencing chronic inflammation and muscle regeneration. However, given that *myd88* is a well-established master regulator of inflammation/infection, these findings are somewhat predictable. The authors should aim to identify more precise regulatory steps within the *nlr3*-*myd88* pathway.

*We appreciate the reviewer’s comment regarding the perceived predictability and novelty of the genetic interaction between *nlr3l* and *myd88*. However, we respectfully disagree that this interaction is expected or well-established. While zebrafish *nlr3l* shares some similarity with mammalian NOD-like receptors (e.g., NLRC3), it lacks the canonical leucine-rich repeat (LRR) domain that define this family, and as described in Shiao et al., 2013, it represents a non-canonical and teleost-specific NLR receptor without a clear mammalian NLR ortholog. Thus, its mechanism of action in vivo has remained undefined.*

*Previous in vitro studies suggested a potential link to inflammasome signaling via ASC (*pycard*), based on in vitro binding assays (Shiao et al., 2013), but without in vivo validation. Our current study through genetic experiments demonstrate that *asc* deletion does not modify *nlr3l* mutant phenotypes (Fig. 3). In contrast, deletion of *myd88* consistently rescues key chronic macrophage activation phenotypes in *nlr3l* mutants (Fig. 3 and 6), establishing a specific and functionally relevant genetic interaction. This is further supported by transcriptomic data (bulk and single-cell RNA-seq), which show that *myd88*- and NF- κ B-associated pathways are significantly dysregulated in *nlr3l* mutant macrophages (Fig. 2, 7 and Supplementary Figure 19).*

*Importantly, the identification of *nlr3l*-*myd88* connection was the result of testing several other candidate genes and pathways—experiments not shown here—but found that none modified *nlr3l* mutant phenotypes, highlighting the specificity and uniqueness of the *nlr3l*-*myd88* interaction.*

*Moreover, discovering this link was not only mechanistically informative but also critical to our overall study: it provided a genetic lever to reverse the chronic immune activation in *nlr3l* mutants, thereby*

*enabling us to assess how chronic immune dysregulation impairs tissue repair. While the mechanistic details of *nlrc3l*-*myd88* signaling would be interesting, dissecting this pathway in depth lies beyond the scope of the present study, which focuses on how chronic macrophage activation, rather than *nlrc3l* itself, affects muscle injury repair. We believe the discovery of the *nlrc3l*-*myd88* connection is both novel and foundational, and merits future study in its own right.*

Similarly, the study attempts to establish CD206 as a critical regulator of macrophages in muscle repair. However, since tissue repair involves extensive cell death and macrophage phagocytosis, it is expected that CD206, a key marker of phagocytosis, would be upregulated. A more in-depth investigation is needed to uncover why certain macrophage subtypes (subtypes 2 and 3, specifically those capable of phagocytosis) are absent in chronic inflammation. Sorting and analyzing these subtypes individually, or employing single-cell sequencing to identify differential pathways, followed by functional validation, could provide meaningful insights. The current comparison of injured versus uninjured macrophages is too simplistic to pinpoint these critical pathways.

*Thank you for your insightful comments. While the *Mrc1*/CD206 may be expected to influence phagocytosis, its specific role in macrophage-mediated skeletal muscle repair has not been established. A key finding of this study is the differential role of the *Mrc1*/CD206, particularly the downregulation of *mrc1b*, in chronically activated macrophages, a novel insight that may explain previously unrecognized deficiencies in macrophage function under chronic inflammation. Our data suggest that *mrc1b* downregulation contributes to the loss of pro-repair macrophage subtypes 2 and 3 in *nlrc3l* mutants. Notably, single-cell transcriptomic analysis across injury timepoints shows that *mrc1b* dynamically marks macrophage states, with unexpectedly high expression at baseline rather than during peak phagocytic activity, highlighting how its *in vivo* role remains poorly understood.*

*Furthermore, we agree that a more in-depth investigation would be important to further distinguish the macrophage subtypes during muscle repair. To address this, we performed a large single-cell RNA-seq experiment using our *irg1*-KI:GFP reporter line. We profiled macrophages and all *irg1*-expressing cells at 24 hours post-injury (hpa), 48 hpa, and in uninjured controls, using large numbers of zebrafish embryos (two independent samples per condition, 90 animals each). As shown in the newly added data in Figs. 7, 8 and Supplementary Figs. 18, 19, 20, the vast majority of the cells were macrophages with some neutrophil presence, from which we attempted to molecularly define the two pro-repair macrophage subtypes.*

*We summarize here the main points of the text we added to the Results and Discussion in relation to our scRNA-seq data. Clustering subtype 2, enriched in both 24 and 48 hpa controls, expressed *mrc1b* and showed upregulation of cholesterol efflux (*fabp4a*, *apoeb*, *apoc1*) and phagocytosis-related genes (*mertka*, *npc2.1*, *npc2.2*, *ctsk*, *ctsl.1*, *ifi30a*). Muscle-encasing subtype 3, observed only at 24 hpa, also expressed *mrc1b* and had significantly elevated expressions of genes involved in TGF- β signaling (*tgfbr1a*, *tgfbi*, *tgfbrap1*). In *nlrc3l* mutants, chronic macrophage activation reduced cellular heterogeneity, with macrophages clustering in a restricted region of UMAP space corresponding to macrophages. These mutant cells shared core pro-inflammatory signatures but still expressed some pro-repair genes, including markers of subtypes 2 and 3, but uniformly downregulating *mrc1b*. This resulted in hybrid M1/M2 states, co-expressing inflammatory and repair-associated markers, which were infrequent in controls (<10% at 24 hpa) but dominated in mutants (up to 50%). These findings suggest that chronic activation impairs repair not by enhancing inflammatory function, but by disrupting reparative programs, particularly through *mrc1b* downregulation and loss of key reparative macrophage subtypes. Chronic persistence of the pro-inflammatory program appears to ultimately constrain macrophage plasticity despite partial expression of repair pathways that we found to involve cholesterol processing, phagocytosis, and TGF- β signaling.*

Moreover, our scRNA-seq data uncovered cathepsin K (*ctsk*), a cysteine protease, as a marker of reparative macrophage subtype 2 (Figs. 7, 8). In vivo validation showed that cathepsin K protein accumulates in the cytosol not only in clustering (subtype 2) but also muscle-encasing (subtype 3) macrophages post-injury, suggesting intracellular collagen and protein degradation as a critical function for a subpopulation of both reparative subtypes (Fig. 8). These densely packed cathepsin K⁺ compartments are absent in chronically activated *nlr3l* mutant macrophages, which show reduced *ctsk* expression at both transcript and protein levels. Notably, cathepsin K⁺ cells co-express *tnfa*, supporting a hybrid inflammatory-reparative M1/M2 state. This identifies CTSK protein accumulation as a new functional marker of reparative macrophage identity and further clarifies how chronic inflammation disrupts macrophage function.

3. The authors state: "In *nlr3l* mutants, restoring macrophage *nlr3l* expression, deleting *myd88*, or both reversed the inappropriate macrophage activation, as evidenced by *irg1* and *tnfa* downregulation, with the combination fully restoring a normal macrophage state (Fig. 3 and Supplementary Figs. 4, 6). These results implicate both cell-autonomous and non-cell-autonomous mechanisms in the inappropriate immune activation."

However, the manuscript does not provide direct evidence for the involvement of non-cell-autonomous mechanisms. The authors need to clarify or provide experimental data to support this conclusion.

*We thank the reviewer for pointing out the confusion caused by the original placement of this statement. The sentence, "These results implicate both cell-autonomous and non-cell-autonomous mechanisms..." was previously included before presenting the germ-free data, which led to ambiguity. To address this, we have moved the statement to the end of the section, after presenting the data showing that commensal microbes contribute to inappropriate immune activation—evidenced by reduced *irg1* expression in germ-free *nlr3l* mutants compared to conventionally raised *nlr3l* mutants. The data supporting the non-cell-autonomous role of commensal microbes in aberrant macrophage activation are presented in Supplementary Fig. 7.*

*The revised text in the Results section now reads: "Given that MyD88 mediates TLR signaling responsible for bacterial recognition and response³⁵, we evaluated macrophage activation in *nlr3l* mutants under germ-free versus conventionally raised conditions (Supplementary Fig. 7). Our findings showed that while commensal microbes contributed, *nlr3l* mutant macrophages themselves retained substantial intrinsic activation (Fig. 3 and Supplementary Fig. 7). Consistent with this, full rescue of inappropriate macrophage activation in *nlr3l* mutants required both macrophage rescue and *myd88* deletion, as neither intervention alone was sufficient (Fig. 6). This indicates contributions from both cell-intrinsic defects and likely non-cell-autonomous signals, supported by the need to delete *myd88*, a gene mediating TLR signaling. Taken together, these findings highlight the intricate nature of the inappropriate macrophage activation in *nlr3l* mutants, where MyD88-dependent signaling, microbial cues, and internal cellular dysregulation collectively contribute to their chronic inflammatory activation state, in addition to the altered state of neutrophils. These results implicate both cell-autonomous and non-cell-autonomous mechanisms in the inappropriate immune activation."*

4. The data on *irf8* mutants following tissue regeneration does not offer significant insights. Any mutation affecting macrophage function or development would be expected to exhibit tissue regeneration defects. Further exploration is necessary to demonstrate the unique relevance of *irf8* in this context.

We thank the reviewer for this comment. We agree that this is important to clarify. Since the role of macrophages in skeletal muscle repair has not been definitively established in zebrafish, the animal model used in our study, it was essential for us to openly and rigorously investigate their involvement. This allowed us not only to validate whether macrophages contribute to the repair process in this

*species, but also to directly compare the response and repair outcomes between macrophage-deficient *irf8* mutants and that observed in our genetic *nlrc3l* mutant model for chronic inflammation.*

Reviewer #3 (Remarks to the Author):

Spencer et al. first produced *ifg1/acod1*-KI zebrafish for tracing activated macrophages. They compared the KI zebrafish with existing models and KO lines and demonstrated the effective and non-functional impairment of the newly established line. Using this line, next, the authors investigated the properties of activated macrophages in a chronic inflammatory state induced by the loss of *nlrc3*, which could be suppressed by the loss of *Myod88* and the restoration of *nlrc3* expression. Furthermore, using a muscle regeneration model, the authors verified the regeneration process including macrophage dynamics under control and *nlrc3*-deficient conditions, and demonstrated that regeneration does not occur when activated macrophages continue to exist, as evidenced by rescue experiments. Finally, the authors focused on the decreased expression of CD206 in the chronically activated macrophages and concluded that CD206 deficiency causes muscle regeneration failure and that CD206 in macrophages is responsible for regeneration failure under persistent inflammation. Overall, the experiments are well-conducted, and the data are convincing. However, this reviewer has some concerns about the model and results of CD206-KD. In particular, it is questionable whether the conceptual advance of this study meets the criteria of Nature Communications.

Major comments

1) The chronic model used here is an artificial model. It is unclear whether this model is applied to naturally occurring disease models. Based on the results found in this study, is it possible to verify the efficacy of the treatment by targeting a conceptually new molecule or pathway?

*Our current experimental paradigm represents a conceptually novel approach by targeting a novel NOD-like receptor, *nlrc3l*-like, in zebrafish to induce a consistent, highly reproducible, and broad-based inflammatory activation of macrophages. This strategy allows us to study the systemic impact of chronic inflammation on acute skeletal muscle injury within a vertebrate model using a robust genetic framework that had not been available previously. Unlike the transient and tightly regulated macrophage activation observed in wild-type animals, typically following a bell-shaped curve, our *nlrc3l* mutant model exhibits sustained and inappropriate macrophage activation throughout the injury and repair process (see Supplementary Fig. 12).*

*Importantly, this is a genetic model designed to uncover fundamental principles underlying macrophage mis-programming, rather than to mimic a specific disease state. We appreciate the reviewer's suggestion to explore the generalizability of our findings using an alternative paradigm. In response, we have developed an additional model based on a persistent systemic *E. coli* infection. This new model enables us to examine whether chronic macrophage activation similarly disrupts normal muscle repair and alters the functional subtypes of macrophages, particularly those associated with *Mrc1*/CD206 receptor downregulation. Using a commonly employed *E. coli* lab strain (MG1655), we microinjected a high dose of bacteria into the brain ventricle of early embryos prior to injury and monitored both injury response and macrophages activation over time. We confirmed that our paradigm induced robust, systemic, and enduring macrophage activation (new data presented in Supplementary Fig. 17). Using this new model, we found that the muscle repair defects closely resemble those seen in *nlrc3l* mutants. Furthermore, these defects were associated with the loss of the two key macrophages subtypes (clustering and muscle-encasing) and downregulation of macrophage *mrc1b* expression, all reminiscent of *nlrc3l* mutants (Supplementary Fig. 17).*

*We have added this new data in Supplementary Fig. 17 and the following text to the Results section: "To further explore the generalizability to additional paradigms of chronic macrophage activation, we developed a model based on persistent systemic *E. coli* infection (Supplementary Fig. 17a-c), and*

showed that similar to *nlr3l* mutants, the chronic macrophage activation caused by infection disrupted normal skeletal muscle repair (Supplementary Fig. 17d-e), and was associated with a *mrc1b* downregulation and reduction of both clustering and encasing macrophage subtypes (Supplementary Fig. 17f-i).”

Together, these complementary approaches strengthen the broader relevance of our findings and provide valuable insight into how chronic inflammatory conditions may derail tissue repair mechanisms through sustained macrophage activation.

2) Given the role of CD206 in macrophage-mediated phagocytosis, it is unsurprising that CD206 knockdown (CD206-KD) results in impaired muscle regeneration. However, the relationship between CD206-KD and MyD88-null experiments remains unclear. Does the loss of MyD88 in NLRC3-null conditions restore CD206 expression? If so, how is CD206 expression regulated under chronic conditions?

We appreciate the reviewer’s thoughtful comments. While the *Mrc1/CD206* may be expected to influence phagocytosis, its specific role in macrophage-mediated skeletal muscle repair has not been established. A key finding of this study is the differential role of the *Mrc1/CD206*, particularly the downregulation of *mrc1b*, in chronically activated macrophages in both genetic and persistent infection models, a novel insight that may explain previously unrecognized deficiencies in macrophage function under chronic inflammation. Furthermore, its essential role in acute muscle injury repair and ability to label macrophage subtypes involved in this process had not been previously demonstrated. Our study reveals that *mrc1b* downregulation contributes to the loss of pro-repair macrophage subtypes 2 and 3. Notably, single-cell transcriptomic analysis across injury timepoints shows that *mrc1b* dynamically marks macrophage states, with unexpectedly high expression at baseline rather than during peak phagocytic activity, highlighting further how its *in vivo* role remains poorly understood.

In response to the reviewer’s suggestion, we performed additional experiments to assess whether loss of MyD88 can restore *mrc1b* expression in *nlr3l* mutants. Our results show that MyD88 knockout in *nlr3l* mutants indeed leads to the recovery of *mrc1b* expression in macrophages. This finding supports the hypothesis that MyD88-dependent signaling contributes to the dysregulated macrophage state observed in *nlr3l* mutants that involves inappropriate repression of *mrc1b* transcription, and that its deletion helps restore a more homeostatic macrophage phenotype, including re-expression of *mrc1b* following muscle injury. This new data has been added to Fig. 6g,h.

While the exact molecular mechanism regulating *mrc1b* remains unclear, our findings provide a foundation for future studies into how chronic inflammatory activation alters *mrc1b* expression during injury and repair. Notably, depletion of MyD88 signaling partially restores *mrc1b* levels, suggesting that the NF- κ B pathway downstream of MyD88 may negatively regulate *mrc1b* transcription, or through NF- κ B target genes that repress *mrc1b* transcription. Fully elucidating this mechanism will require dedicated studies to identify key transcription factors or regulators, develop new reporters, and perform targeted functional assays, which are beyond the scope of this work. We view this as an exciting future direction and have added relevant discussion to the revised manuscript.

3) Lines 65-66 ; dynamic in vivo and longitudinal analyses of cellular interactions across distinct phases of the injury and repair process are lacking

Some studies conducted dynamic and longitudinal analyses using scRNA-seq. For instance, the following studies conducted dynamic, longitudinal, interactome analyses using scRNA-seq during muscle regeneration in detail. This reviewer recommends citing these papers and re-write more precisely.

iScience. 2020 Apr 24;23(4):100993. doi: 10.1016/j.isci.2020.100993.

Point well taken. We have revised the text to more clearly distinguish our study, which, to the best of our knowledge, is the first to capture real-time in vivo imaging of dynamic cellular interactions during skeletal muscle repair in intact, living animals. This was made possible through time-lapse imaging combined with a rigorous longitudinal microscopy approach. The recommended citations have also been added.

Revised text in manuscript states: Although prior studies have provided insights into the role of macrophages in responding to and facilitating acute muscle injury repair^{1,3,6,8}, including via single-cell transcriptomics and trajectory analysis to infer temporal changes and cellular interactions^{12,13}, direct real-time in vivo imaging of macrophage behaviors and interactions throughout the injury response remains lacking. Timelapse imaging as well as a rigorous longitudinal microscopy approach enabled us to track macrophages across distinct muscle injury and repair phases within the same subjects.

4) Lines 204; It is difficult to find Mrc1b/cd206 in the figure.

We have now made the mrc1b/cd206 text in red in Figure 2g, so that the data for this gene is easier to find in the heat map.

5) Line 244 ;

What is ASC? Please add a description of the relationship between ASC and nlr3l.

Thank for pointing out the lack of description on ASC. We have added and modified text to address this, which now states: "Furthermore, because several Nod-like receptors are known to interact with the adaptor protein called apoptosis-associated speck like protein containing a CARD (also known as ASC) to form cytosolic multiprotein complexes called inflammasomes, responsible for activation of inflammatory signaling, we explored whether macrophage activation was ASC-dependent. We generated nlr3l/asc double mutants that showed no change from the nlr3l mutant activation phenotype, thereby indicating nlr3l functioned in an inflammasome-independent manner in this context (Fig. 3)."

6) Lines 451-452; The precise role of macrophages in skeletal muscle injury and repair remains incompletely understood.

Please describe what the precise role is not understood. In addition, the authors need to underscore what the present study reveals in the role of macrophages in muscle regeneration. This reviewer thinks that mechanisms inducing chronic inflammatory conditions in muscle regeneration are unexplored. However, the disruption of muscle regeneration by chronic inflammation is not conceptually new.

Thank you for your helpful comments. We have revised the text to better highlight the knowledge gap we are addressing, now stating: "The impact of prolonged inflammatory activation on macrophage heterogeneity and function during muscle repair remains poorly defined."

To highlight the key advances of our study, we emphasize the work we have conducted to show that chronic inflammatory activation of macrophages reconfigures their states in a manner that impairs their ability to support muscle repair. Integrating in vivo dynamic imaging with single-cell RNA-seq, we identified two previously unknown reparative macrophage subtypes, apoeb+/ctsk+ clustering and tgfb1+/sb:cb81+ muscle-encasing, that are selectively lost in both genetic and persistent infection models of chronic inflammation (Fig. 7 and Supplementary Figs. 13, 17-19). Contrary to conventional discrete states, we show that chronic activation drives macrophages into a dysfunctional hybrid M1/M2 state, marked by sustained pro-inflammatory gene expression despite partial activation of reparative pathways. Mechanistically, chronic inflammatory activation of macrophages represses mrc1b via a MyD88-dependent pathway, leading to the loss of reparative subtypes enriched for genes involved in

cholesterol efflux, phagocytosis, and TGF- β signaling. scRNA-seq further identified cathepsin K (*ctsk*), a cysteine protease, as a novel marker of reparative subtype 2. With further *in vivo* validation, we show that dense CTSK protein accumulations are found intracellularly in a subpopulation of both reparative subtypes post-injury, an effect strikingly absent in chronically activated *Nlr3l* mutants (Fig. 8). Notably, CTSK⁺ macrophages co-express *tnfa*, consistent with a hybrid inflammatory-reparative phenotype (Fig. 8). These findings establish CTSK protein accumulation as a functional marker of reparative macrophage identity and provide new mechanistic insight into how chronic inflammation reshapes macrophage states, generally restricting functional plasticity, compromising their role in tissue repair. We have added text in the manuscript to better highlight our new findings and the significance of this study.

7) Lines 358-361, 461

The authors described that chronically activated macrophages inhibit satellite cell activation. There is no evidence that chronic macrophages directly inhibit satellite cell activation. Is it possible that chronically activated macrophages inhibit not the activation, but the proliferation? And is it possible that the chronic macrophages disrupt the balance of pro-inflammatory macrophages inducing satellite cell activation or proliferation?

We thank the reviewer for the thoughtful comment. We agree that it remains unclear whether the effects of chronically activated macrophages on stem cell activation are direct or indirect. We do not make this claim, but agree this represents an interesting avenue for future study to dissect the underlying signals.

Satellite cell activation has been defined in literature as the re-entry of quiescent cells into the cell cycle, leading to proliferation and relocalization of these stem cells from their typical location between the sarcolemma of the muscle fibers to the site of injury. Therefore, activation is coupled with proliferation. Both of these features—an increase in Pax7⁺ satellite cells and their enrichment at the injury site—are observed post-injury in wild-type and controls, but not in mutants, as shown in Fig. 12a. We therefore concluded that the chronically activated macrophages in the mutant "coincided with deficient activation" of stem cells (lines 358-361), and in the Discussion, in line 460, the data suggest stem cell activation is impaired in the mutant.

We have now added a citation to Fu et al., 2015 Cell Mol Life Sci. (doi: [10.1007/s00018-014-1819-5](https://doi.org/10.1007/s00018-014-1819-5)) and scRNA-seq study characterizing muscle stem activation (Micheli et al., 2020 Cell Reports <https://doi.org/10.1016/j.celrep.2020.02.067>) to provide further context on stem cell activation in skeletal muscle regeneration. We agree with the reviewer that the chronic activation of mutant macrophages is likely causing a major shift in macrophage states. We added text in the Discussion following line 460 that the impaired repair is "likely due to a significant imbalance in macrophage states, skewing heavily toward a chronically pro-inflammatory profile,".

Point-by-point response to the reviewers' comments is shown in *italicized* text.

Reviewer #1 (Remarks to the Author):

The authors have satisfactorily addressed all my concerns in the revised manuscript. Importantly, they have now included a comprehensive scRNA-seq analysis, which provides high-resolution validation of the bulk RNA-seq data presented in the original version. These new data not only strengthen the conclusions but also significantly expand the impact of the work.

The exhaustive characterization of macrophages and neutrophils in the context of chronic inflammation is exceptional and provides information of broad relevance to the field, beyond the scope of this article. Of particular interest is the identification of a hybrid M1/M2 macrophage state that predominates under chronic inflammation, as well as the delineation of numerous markers of macrophage polarization, such as *ctsk*.

Overall, the manuscript is now excellent in all respects. The study represents a major advance in our understanding of immune cell heterogeneity and plasticity during chronic inflammation and tissue repair.

I strongly recommend acceptance of this article.

We are grateful to Reviewer 1 for their recognition of the depth and significance of our work, and for their insightful and constructive feedback, which have substantially strengthened our study.

Reviewer #2 (Remarks to the Author):

The overall quality of the manuscript has improved significantly after revision, and some of my previous concerns have been addressed. However, several issues remain that the authors must further clarify and revise.

Major points

1. I acknowledge and agree that zebrafish *Nlrc3l* may represent a unique NLR without a clear mammalian ortholog. Nevertheless, its mechanism of action is expected to be evolutionarily conserved. Currently, there is insufficient evidence to support that zebrafish *Nlrc3l* functions through a completely distinct mechanism from mammals. As the authors noted, zebrafish *nlrc3l* lacks the LRR domain. In mammals, the only reported NLR lacking an LRR is NLRP10 (Kanneganti TD et al., *Immunity* 2007). Previous studies have demonstrated that NLRP10 can induce ASC speck formation and participate in anti-inflammatory regulation in human keratinocytes and mouse intestinal epithelial cells (Próchnicki T et al., *Nat Immunol* 2023; Zheng D et al., *Nat Immunol* 2023).

*We thank Reviewer 2 for the continued thoughtful feedback. While we understand the desire to draw parallels between *nlrc3l* and known mammalian NLRs such as NLRP10 because it also lacks LRR, it is important to underscore that *nlrc3l* lacks a mammalian ortholog, making it inherently difficult to predict its mechanism of action based on conserved domains or canonical pathways. Notably, NLRP10 has been shown to act through inflammasome and non-inflammasome pathways (Joly et al., 2013 *J. Immunol.* and reviewed in Zhong et al., 2013 *Front. Immunol.*). Several other NLRs (e.g., NOD1/2, NLRC5) are well-documented to also function through non-inflammasome pathways, activating transcriptional regulators like NF- κ B or MAPKs, and many mechanistic aspects of mammalian NLRs especially the less-studied members remain unresolved (reviewed in Platnich and Muruve 2023 *Arch Biochem and Biophys*).*

However, the authors' current conclusion that Nlrc3l is independent of Asc is based solely on low-resolution stereomicroscope imaging in Fig. 3C, which seems premature. Zebrafish are three-dimensional organisms, where signals from the opposite side are easily missed, and out-of-focus GFP signals are inherently attenuated.

Regarding the use of the low-resolution fluorescent stereomicroscope imaging, we would like to clarify that this setup was intentional for stringent screening of inflammatory macrophage activation, requiring a higher GFP detection threshold, to identify embryos with high GFP expression as a clear indicator for irg1/acod1 induction using the irg1-KI:GFP reporter. As shown in Fig. 3C, high GFP+ macrophages persist in nlrc3l/asc double mutants, comparable to nlrc3l single mutants, as opposed to wild-type and control siblings which have either undetectable or barely detectable GFP+ cells. Quantification of the data is adjacent to the images, and two biological replicates of single nlrc3l and double nlrc3l/asc mutants are shown to display the typical range of macrophage presence on yolk ball (Fig. 3C). There is no concern of missing GFP signals because our conclusion is based on finding high GFP expression in the nlrc3l/asc double mutants, and therefore supporting the conclusion that macrophage activation still persists despite asc deletion. Additional new data from qPCR in this submission as described next supports this finding (Supplementary Fig. 7).

More reliable approaches should include qRT-PCR or quantitative GFP Western blot. Moreover, the number of irg1⁺ cells shown in Fig. 3C appears to be slightly reduced in Nlrc3l/Asc double mutants. Even within zebrafish studies, evidence already suggests that Nlrc3l regulates microglial cell death in part through Asc (Chang MX et al., Dev Comp Immunol 2021; Wang T et al., J Genet Genomics 2019). Taken together with mammalian and zebrafish findings, as well as potential misinterpretation of the current data, it remains premature to exclude the possibility that Nlrc3l phenotypes are ASC-related. Additional experimental data are required.

To address the reviewer's concern with additional data, we performed additional experiments in this revision. Specifically, we added qPCR analysis of endogenous irg1/acod1 expression and a functional skeletal muscle repair assay to test whether asc deletion modifies the inflammatory macrophages or tissue repair defects of nlrc3l mutants. In both assays, asc deletion failed to rescue the nlrc3l phenotype, consistent with our prior imaging and genetic epistasis data (shown in Fig. 3). By contrast, macrophage-rescue of nlrc3l mutants was used as a positive control, which demonstrated rescue of the repair deficiency. These new findings are presented in Supplementary Fig. 7 and the corresponding figure legend text describing the results is copied below.

Relevant text to new data added in Supplementary Fig. 7: "b Quantification of endogenous irg1 (also known as acod1) transcript levels by qPCR demonstrates aberrantly elevated irg1 expression at 3 dpf in baseline nlrc3l mutants, as well as in nlrc3l/asc double mutants, indicating no evidence of a genetic interaction between nlrc3l and asc. c Functional assessment of the nlrc3l-asc interaction during the resolution phase of acute muscle injury response at 48 hours post-amputation (hpa) shows comparable persistence of unresolved muscle tissue (orange arrows) in nlrc3l single and nlrc3l;asc double mutants, whereas controls exhibit effective clearance of damaged muscle, further supporting an asc-independent role for nlrc3l. Macrophage-specific rescue of nlrc3l expression, which restores normal repair, was included as a positive control for the assay. d Quantification of phenotypic outcomes across genotypes is presented in the accompanying bar chart."

With regard to the citations on other studies of zebrafish nlrc3l, we are well aware of them, one being a review article Chang MX et al., Dev Comp Immunol 2021 that summarizes the primary research study by Wang et al., J. Genet. Genom. 2019. The rescue observed in a temperature-sensitive nlrc3l mutant by asc deletion (Wang et al.) is partial and reflects a different model and biological process (microglial maintenance and survival after microglia establishment where they were looking for recovery of microglial numbers), and uses a milder allele compared to the

nonsense mutation of *nlr3l* employed here. These differences in context and allele strength may explain the divergence in results regarding the role of *asc*.

In sum, across multiple independent approaches, our datasets consistently show that deletion of asc does not modify the nlr3l mutant phenotype, namely inappropriate macrophage activation and impaired tissue repair (see Supplementary Fig. 7). While nlr3l shares some structural features with other NLRs, its mechanism is yet unknown and requires future dedicated mechanistic investigation beyond the scope of this study. The additional transcriptional and functional data presented in this revision further reinforces a mechanism by which nlr3l negatively regulates macrophage inflammation independently of asc.

2. The authors attempt to explain the phenotype of Nlr3l deficiency via the “master regulator” Myd88, but this level of mechanistic interpretation is insufficient for a Nature Communications-level study. This is somewhat analogous to showing that an anti-inflammatory drug partially rescues the phenotype—an expected observation, but not true mechanistic insight.

We thank Reviewer 2 for their comments and the opportunity to clarify the scope and significance of our study.

*Regarding the Reviewer’s concern that the involvement of myd88 is not a “true mechanistic insight,” our genetic approach provides gene-specific analysis not achievable with pharmacological inhibition, which often produces complex off-target effects. While myd88 is a known adaptor in several inflammatory pathways pertaining to TLRs, its role in chronic macrophage activation, particularly in our *nlr3l* mutant model, was previously unknown. Moreover, myd88 does not mediate all inflammatory signaling, and its involvement in Nod-like receptor signaling remains poorly understood; importantly, myd88-independent pathways such as TRIF-dependent TLR signaling are also well established. The finding that myd88 knockout switches aberrantly activated *nlr3l* mutant macrophages to more normal states was a key finding for our study, and from another perspective, it underscores the relevance of our study to other chronic inflammatory conditions driven by excessive Myd88 activation.*

*The broader goal of our study is to address a key challenge in immunology, which is to understand the dynamic transitions and shifts in immune cell states by direct visualization of these processes in real time at single cellular resolution during a perturbation. This study is **not** focused on defining the molecular mechanism of *nlr3l*, but rather on using *nlr3l* mutants as a genetic tool to probe macrophage plasticity. By forcing all macrophages toward chronic pro-inflammatory (M1-like) states *in vivo* using the *nlr3l* mutants (as shown in Figure 2 and Supplementary Fig. 12), we can then directly test how adaptable and reversible macrophage states are upon an injury perturbation within the intact living organism, capturing its full physiological complexity, often in real time, in contrast to most studies that are limited to *in vitro* and *ex vivo* analyses.*

*Combining new imaging tools, including a GFP knock-in at the *irg1/acod1* locus, with single-cell analysis we identified dynamic hybrid M1/M2 macrophage states as well as novel reparative subtypes such as muscle-encasing macrophages, all previously not known, that coordinate early repair but become functionally impaired or lost under chronic inflammation. Notably, we found that chronic activation drives myd88-dependent drastic downregulation of the mannose receptor *mrc1b/cd206* in both the genetic and persistent infection model of chronic inflammation. This dynamic modulation of *mrc1b* links chronic inflammatory signaling to loss of macrophage reparative capacity (including expression of pro-repair genes such as accumulation of cathepsin K), providing a mechanistic bridge between inflammation and defective tissue repair.*

Furthermore, the *irg1* used in the Myd88-dependent assays is expressed only in myeloid cells, so the conclusion applies only to this lineage, not to the entire organism. Similarly,

the irg1 reporter line and mpeg1-nlrc3l rescue address immune cell-specific phenotypes, but cannot rule out contributions from non-immune cells.

We appreciate the reviewer's comment. To clarify, while the roles of non-immune cells may be interesting, they are beyond the scope of this study. Our work specifically focuses on how chronic activation affects macrophage cellular heterogeneity and plasticity on both a functional and molecular basis. The experimental design was intentionally tailored to target macrophages, and our conclusions are limited to the immune cells that were directly manipulated, analyzed, and characterized in this study.

In Fig. 3D, the difference between the last two groups exceeds threefold, yet the small sample size (n ≈ 5 per group) makes the lack of statistical significance unconvincing.

We interpret the Reviewer's comment as referring to the last two groups containing the macrophage-rescue (mpeg1-nlrc3l) construct: comparing nlrc3l mutants and nlrc3l/myd88 double mutants, both with the macrophage-rescue. In both groups, irg1 expression was generally very low or undetectable, indicating the restoration of normal homeostatic states in macrophages. Although the double mutants with the macrophage-rescue showed a trend toward even lower irg1 expression, suggesting a more complete rescue, this difference was not statistically significant. Since the macrophage-rescue restores both conditions to relatively more normal states, the potential statistical difference would not alter the overall conclusion. To provide more context to the experimental design, these experiments are technically challenging, as they require double heterozygous incrosses, which yield only 1 double mutant in every ~ 16 animals. To obtain the seven double knockouts we analyzed, we conducted multiple experiments totaling over 100 animals. All double mutants were analyzed, and a random subset of control animals without the nlrc3l deletion was included. Because all controls consistently exhibited low or undetectable irg1 expression, random sampling of controls provided an adequate comparison.

The subsequent germ-free model experiments further underscore the presence of non-cell-autonomous mechanisms, reinforcing this concern. In such models, commensal bacteria primarily interact with the skin and intestinal epithelium, where NLRP10 has been shown to be highly expressed. Given that the tissue injury model used by the authors inherently damages epithelial cells, a more plausible explanation is that Nlrc3l deficiency triggers commensal-driven mucosal systemic inflammation, which in turn disrupts macrophage function. This hypothesis may actually represent a more valuable research direction. I encourage the authors to integrate findings from NLRP10-related studies to explore this potential mechanistic conservation between zebrafish Nlrc3l and mammalian NLRP10.

The data do not indicate any major off-target effects in tissues or cell types beyond those analyzed. First, nlrc3l mutant macrophages exhibit inappropriate baseline activation even without injury or external challenge (Figs. 2, 3). Moreover, macrophage-specific rescue in nlrc3l mutants restored appropriate muscle injury repair (Fig. 5), indicating that the observed phenotypes related to injury response are macrophage-cell-autonomous. Furthermore, whole-body and in vivo imaging during muscle injury showed no abnormal immune cell coalescence or migration to the skin or other non-injured sites. Given that the imaging captures the entire body, including the skin, any off-target immune responses (particularly the skin) would have been detectable but were not observed.

3. The association of CD206 with tissue regeneration has been well documented (Shook B et al., J Invest Dermatol 2016; Honda A et al., EMBO Rep 2025). While not necessarily specific to skeletal muscle regeneration, the underlying logic remains the same. The current findings resemble more of a “me-too” type of study. The authors should sufficiently reference and acknowledge existing literature and refine the novelty of their study in light of prior research.

Thank you for the Reviewer's comment. We agree that expanding the Discussion to include additional studies identifying CD206 as a marker for reparative macrophages in other tissue injury models strengthens the broader relevance of our findings. Accordingly, we have cited Shook et al. (2016) and Honda et al. (2025), which demonstrated that CD206 marks major macrophage populations associated with skin wound repair, respectively, along with other relevant CD206 studies in the Discussion.

We also clarify in the revised Discussion how our work distinguishes from these prior studies. The novelty of our findings lies not in identifying Mrc1/CD206 as a reparative macrophage marker, but in uncovering its dynamic regulation during injury and chronic inflammation. Specifically, our new scRNA-seq analysis shows that Mrc1/CD206 expression is highest in steady-state macrophages, normally downregulated to enable reparative programs during the initial inflammatory phase of an injury response, but becomes pathologically suppressed under chronic inflammatory conditions (shown by several independent in vivo experiments- bulk and single-cell RNA-seq of sorted macrophages, HCR-based RNA in situ hybridization, and complementary infection model). We further demonstrate that this drastic suppression in chronically activated macrophages is MyD88-dependent and associated with loss of reparative features, including cytosolic cathepsin K accumulation. These insights reveal a previously unrecognized mechanism linking chronic inflammation to a severe loss of mannose receptor as a cause for reduced macrophage plasticity and impaired tissue repair.

Minor points

1. In Fig. S17, the bacterial injection site is not the brain ventricle.

Thank you for pointing out this issue that we missed. The cartoon has been corrected to show the needle pointing into the brain ventricle region.

Reviewer #3 (Remarks to the Author):

In the first round of review, this reviewer raised three major concerns: (1) the generalization of the findings in the Nlrc1 mutant model, (2) the expression in Nlrc3l/Myd88 double mutant mice, and (3) the conceptual advance of this study. The authors have sincerely addressed these concerns with new data and descriptions. In addition, the authors performed scRNA-seq analyses, which strengthen their findings on the alteration of macrophage subtypes in chronic environments. Please address the following comments to further improve the manuscript.

- **There are several instances in the main text where the specific figure panels are not indicated. Please provide this information consistently. In particular, the descriptions of Figures 3, 5, 8, and supplemental figures do not mention the corresponding panels.**

Thank you for the helpful feedback. We have added reference to specific figure panels to the main figures and supplemental figure where specific panels, rather than entire figures, are relevant.

- **Related to the above comments, it is unclear which figure corresponds to the description of the scRNA-seq results. The relationship between subtypes 1–3 and the scRNA-seq clusters represents a crucial result in this manuscript. The reviewer recommends that all genes used to analyze the relationship between imaging subtypes and scRNA-seq clusters be presented in the main figure, so that readers can readily recognize them.**

Thank you for the helpful comment. We have added additional references in the manuscript to specific scRNA-seq datasets corresponding to Figure 7 and Supplementary Figures 18-20, as we agree this better highlights the molecular profiling of the subtypes. We have also revised the main Figure 7 to include the key genes defining the three subtypes, which were originally shown in Supplementary Figure 19. Additional supporting data characterizing the macrophage-specific cells

will remain in Supplementary Figure 19 due to space constraints.

• Figure 6, Figure 4, Supplemental Figure 11, 15 contain errors in their preparation.

Thanks to Reviewer's careful review of these figures. We have corrected all errors we detected, including the image panels that were not aligned and missing annotations on plots or images.

•The meaning of the sample name in Supplemental Figure 19f is unclear. What does 3d_1 signify?

Thanks for pointing out the missing descriptions. The sample names indicate the condition based on genotype and injury-response timepoint from which the cells were derived. The suffix (e.g., "3d_1") denotes the zebrafish's stage (3 days post-fertilization) and the biological replicate (replicate 1 of 2). The bubble plot in Fig. 19f displays data for all samples, allowing comparison across all biological replicates and conditions. It highlights that the DEGs were highly robust and consistent across replicates. Unsupervised hierarchical clustering shows that samples grouped by condition, with mutant 24 hpa samples being the most distinct, followed by the control 24 hpa samples.

We have added the following text to the Figure legend to clarify the labels:

"The sample names below the bubble plot denote the genotype and either the injury-response time point (24 h or 48 h) or the uncut baseline, with the suffix indicating zebrafish stage (such as 3d for 3 dpf) and biological replicate (such as replicate 1 of 2)."